# An estrogen receptor/E2F1/CDKN3 axis protects from UV-induced skin cancers in females

Céline Lukowicz [1✉], Carine Winkler [1], Catherine Roger[1], Joanna C Fowler [2], Yi-Chien Tsai[3,4], Joachim Meuli [5], Stéphanie Claudinot [1], Yun-Tsan Chang [3,4], Christoph Iselin[3], Philip H Jones [2,6], Emmanuella Guenova[3,4,7], Paris Jafari[1] & Liliane Michalik [1✉]

## Abstract

**Men have a higher risk of developing cutaneous squamous cell carcinoma (SCC) compared with women, but models and comprehensive analyses of signaling pathways highlighting this sexual dimorphism are missing. Here, we use a UV-induced SCC model in hairless mice recapitulating this sex difference, with enhanced SCC development in males. While UV-induced DNA damage is similar between sexes, we uncover sex-specific responses in epidermal proliferation and differentiation. Global transcriptional profiling identifies E2F transcription factors as key sex-specific markers of the proliferative response to UV. E2F1/2 and their target gene CDKN3 are selectively downregulated in female mouse and human epidermis following UV exposure, and this is mediated by estrogen receptors. CDKN3 depletion impairs SCC cell progression into S-phase and reduces tumor growth in xenograft models. Consistently, low CDKN3 expression in head and neck SCC occurs exclusively in female patients and correlates with better prognosis. We thus reveal a mechanism protecting women from carcinogen-induced cancer formation, which could lead to better sex-targeted preventive and therapeutic strategies in SCC.**

**Keywords** UV-induced Squamous Cell Carcinoma; Skin; Proliferation; Estrogen Receptors; E2F Transcription Factors
**Subject Categories** Cancer; Cell Cycle; Chromatin, Transcription & Genomics

## Introduction

Differences in the incidence of at least 21 different cancers, excluding reproductive tissues, have been reported between men and women, with men having a greater risk of developing cancer than women (Clocchiatti et al, 2016; Jackson et al, 2022). This sex-difference also includes cancer mortality worldwide, with death rates 43% higher in men than in women (Sung et al, 2021).

Cutaneous Squamous Cell Carcinoma (SCC), the second most common keratinocyte skin cancer (Que et al, 2018), is also affected by this sexual dimorphism. Indeed, sex is recognized as a risk factor for SCC as important as fair skin or advanced age (Nagarajan et al, 2019), and a recent meta-analysis of cutaneous head and neck SCC reported that men are more commonly affected than women (Tan et al, 2022). This sex-disparity in SCC prevalence has been mainly attributed to differences in lifestyle (e.g. outdoor work (Szewczyk et al, 2018), behavior (e.g. motivation to sun protection measures and men shorter hair preferences (Roberts et al, 2021) and genetics (e.g. common men baldness (Jenkins et al, 2014) related to UV exposure. Beyond sex-differences in the immune system (Budden et al, 2021; Keller et al, 2010; Ramsay et al, 2003) and in oxidative stress in response to UV exposure (Thomas-Ahner et al, 2007), intrinsic factors and physiological differences remain largely underexplored.

In particular, the role of sex hormones in modulating vulnerability to SCC is barely understood. Nevertheless, an elevated risk of developing keratinocyte skin cancers was associated with the use of oral contraceptives and with hormone replacement therapy (Asgari et al, 2010; Li et al, 2023). Along the same lines, the androgen receptor seems to be a key determinant of melanoma development, another type of skin cancer (Ma et al, 2021). Besides multiple genes present on sex chromosomes have been proposed to also play a role in oncogenesis (Clocchiatti et al, 2016). Collectively, these data strongly suggest that sex hormones have a role in modulating SCC incidence in humans.

UV is the main carcinogenic factor for SCC (De Gruijl et al, 2001; Wei et al, 2021), but sex cellular and molecular responses of the skin to UV rays have not been described. UV provokes direct DNA damage which leads to mutations if unrepaired. UV exposure triggers a cascade of molecular events called the DNA damage response in exposed cells, aiming to repair DNA damage. Cell cycle arrest is required to allow a proper and timely regulated DNA repair and can switch to apoptosis if the DNA damage burden exceeds the repair capacity of the cell (Melnikova and Ananthaswamy, 2005). This initial response is followed by a mitogenic response, which leads to epidermal hyperplasia, a thickening which improves the protection of the basal layer of the epidermis against

[1]Center for Integrative Genomics, University of Lausanne, Lausanne, Switzerland. [2]Wellcome Sanger Institute, Hinxton, UK. [3]Department of Dermatology, Lausanne University Hospital (CHUV) and Faculty of Biology and Medicine, University of Lausanne, Lausanne, Switzerland. [4]Clinical Research Institute for Inflammation Medicine, Medical Faculty, Johannes Kepler University, Linz, Austria. [5]Department of Plastic and Hand Surgery, Lausanne University Hospital (CHUV), University of Lausanne, Lausanne, Switzerland. [6]Department of Oncology, University of Cambridge, Cambridge, UK. [7]Clinical Department of Immunodermatology, Kepler University Clinic and Medical Faculty, Johannes Kepler University, Linz, Austria. ✉E-mail: celine.lukowicz@unil.ch; liliane.michalik@unil.ch

further penetration of UV rays (Berton et al, 2001; El-Abaseri et al, 2006). The cutaneous response to UV also includes the production of various inflammatory cytokines such as Interleukin 1 (IL-1), inflammatory metabolites, arachidonic acid and mast cell-derived mediators leading to sun burn reaction (D'Orazio et al, 2013; Lisi Hruza and Pentland, 1993).

Understanding the role of sex hormones and their receptors, as well as sex-disparities in UV responses, is crucial to understand why women are better protected than men from the risk of developing SCC. Here, we provide cellular and molecular evidence underlying differences in SCC development in males and females using a relevant mouse model for UV-induced skin cancers. Specifically, we demonstrate that UV exposure triggers a sex-dependent response through the modulation of the E2F1/CDKN3 axis mediated by estrogen receptors. Finally, we prove the relevance of our findings to humans, using ex vivo skin explant cultures from women, in vivo xenograft models of human SCC, and squamous cell carcinoma biopsies from patients.

## Results

### Chronic UV exposure enhances squamous cell carcinoma development in males compared to females

Chronic UV exposure is a major risk factor for skin carcinogenesis. Men are more likely to develop skin cancer and the reasons for that are still poorly understood. We used a mouse model to document such sex-disparities, comparing the skin responses of male and female mice to chronic UV exposure. B6.129-SKH1-Hr$^{hr}$ mice were exposed to UV three times per week for 6 months (UV 70 mJ/cm$^2$: a sub-erythematous dose), which provoked the formation of various lesions on their dorsal skin. Macroscopic observations revealed that the dorsal skin of males was far more affected than females: on average, 65% of the dorsal skin surface of males showed UV-induced lesions, compared to only 27% of the dorsal skin surface of females (Fig. 1A).

The severity of these lesions was further characterized by a dermatopathologist in a blinded manner. Chronic UV exposure of the skin is well-known to lead to the development of precursor tumoral lesions named actinic keratosis (AK) and of cancers of epithelial origin named cutaneous squamous cell carcinoma (SCC). Female mice presented on average, higher number of benign AK lesions, and tended to have fewer malignant SCC lesions compared to male mice (Fig. EV1A). Each collected AK and SCC was then further characterized according to its tumor grade, based on the degree of keratinization and of keratinocyte differentiation. This characterization revealed that tumors were more severe in males compared to females (Fig. 1B). Indeed, the commonest tumors collected in males were SCC grade II, while in females the commonest of the collected tumors were AK grade III (Fig. 1B). Regarding the most severe tumors, 22.5% versus 14% of the collected lesions were SCC III in males and females, respectively. As AK (grade III) and SCC (grade II) are the predominant tumor types in females and males, respectively, we next investigated their proliferation rates by Ki67 staining. No difference between sexes was observed (Fig. EV1B,C), and tumor depth increased with the severity of SCC grade, only significantly

in males (Fig. EV1D). In summary, male mice bear a greater overall tumor burden compared to females, with a predominance of more advanced lesions upon chronic exposure. This mouse model of photo-carcinogenesis therefore recapitulates epidemiological data in humans that cannot be attributed to lifestyle or behavior, making it an excellent model for studying the cellular and molecular basis of sex-disparity in the development of UV-induced SCC.

While chronic exposure to UV rays causes AK and SCC, acute UV exposure (or single UV exposure, 120 mJ/cm$^2$) induces transient damages resulting in reversible cellular atypia, which we then characterized in the epidermis of male and female mice. Atypia was graded by a dermatopathologist in a blinded manner 24 h and 72 h after UV exposure, based on disorganization of epidermal layers and apparent phenotypical changes in histological sections (Fig. 1C). All epidermis samples showed some degree of atypia 24 h after UV exposure, which diminished in severity after 72 h, demonstrating skin recovery over time. Atypia severity was strikingly different between the sexes, affecting more females than males. Specifically, 66.7% of females versus only 6.7% of males presented a severe atypia (grade 3) 24 h after UV exposure (Fig. 1D). At the same time point, most (80%) of males showed intermediate atypia, whereas 33.3% of females showed intermediate atypia. Differences between sexes remained 72 h after UV exposure, with 25% of females still presenting severe atypia, a condition no longer present in males (Fig. 1D). Epidermal thickness increased in both sexes in response to acute UV (Fig. 1E). However, 72 h after UV exposure, males presented greater epidermal thickening than females, with an average thickness of 72 μm for males versus 57 μm for females (Fig. 1E).

Collectively, these data show that both acute and chronic UV exposure affect the epidermis in a sex-specific way. On the long run, male mice develop more severe skin lesions caused by chronic UV exposure whereas, in the short term, female mice are more affected by a single dose of UV exposure.

### UV exposure induces similar DNA damages and mutations patterns in male and female epidermis

We next addressed the mechanisms underlying such sex disparities in the skin responses to UV. We first assessed if these differences arise from variations in direct DNA damage provoked by UV in male and female epidermal cells. Cyclobutane pyrimidine dimers (CPD) are the major type of direct DNA damage caused by UV exposure, representing 75–90% of the total UV lesions (Friedberg et al, 1995). We observed similar amounts of CPD one hour after a single UV exposure (120 mJ/cm$^2$) in male and female epidermis, which were being efficiently removed over time by the DNA repair process underway in both sexes (Fig. 2A). Among the numerous proteins involved in DNA damage response, the histone variant H2AX plays an important role. Upon DNA damage, H2AX is phosphorylated at Ser-139 (Ivashkevich et al, 2012) (γH2AX), which is the first step for localizing DNA damage and recruiting repair proteins. Using immunofluorescent staining of γH2AX as a biomarker for DNA damage, we observed that 24 h after acute UV exposure, male and female murine epidermis exhibited a similar proportion γH2AX-positive keratinocytes presenting a foci staining pattern (Fig. 2B).

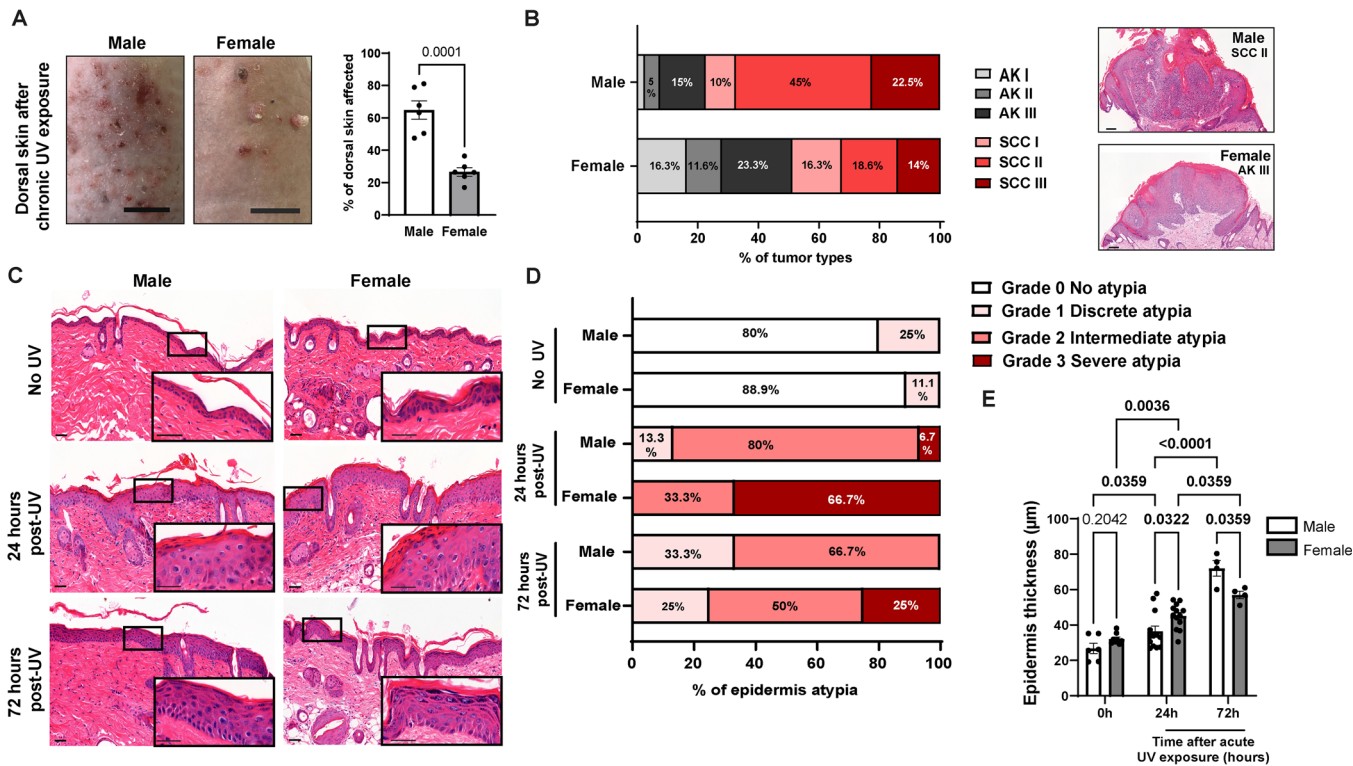

**Figure 1. Chronic and acute UV exposure induce distinct, sex-specific skin responses in mice.**

(A) Left: Representative images of the dorsal skin in male and female mice following chronic UV exposure (3 times per week for 6 months; scale bars: 1 cm). Right: Percentage of dorsal skin with UV-induced lesions in male and female following chronic UV exposure. $n = 6$ mice, mean ± SEM, unpaired $t$ test. (B) Left: Percentage of actinic keratosis (AK) and squamous cell carcinoma (SCC) lesions graded, based on lesions collected from male ($n = 43$) and female ($n = 40$) mice after chronic UV exposure. Right: Representative H&E-stained micrographs of the predominant tumor types for each group. Scale bars: 100 μm. (C) Representative H&E-stained images of dorsal skin in male and female mice 24 h or 72 h after a single dose of UV exposure (120 mJ/cm²), compared to control mice not exposed to UV (No UV). Scale bars: 50 μm and for the magnified image, 10 μm. (D) Histogram showing the percentage of epidermal atypia grading in male and female mice following a single dose of acute UV exposure (120 mJ/cm²) at 24 h and 72 h time points, compared to control mice (No UV). No UV: males ($n = 5$), females ($n = 9$); 24 h post-UV: males ($n = 15$), females ($n = 18$); 72 h post-UV: males ($n = 6$), females ($n = 4$). (E) Epidermal thickness measurements in male and female mice following a single dose of acute UV exposure (120 mJ/cm²) at 24 h and 72 h time-points, compared to control mice (No UV). $n = 4$–12 mice, mean ± SEM, two-way ANOVA followed by Holm–Šidák multiple comparisons test. $P$ value < 0.001 = 0.000000004. Source data are available online for this figure.

When DNA damages are not repaired, mutations are acquired in response to external agents such as UV (De Gruijl et al, 2001). We quantified the number of mutations present in 2 mm² areas of the epidermis of male and female mice exposed to UV for 2, 8 or 12 weeks followed by ageing, by ultradeep (average 1361× target coverage) targeted sequencing of 74 genes commonly mutated in SCC and other cancers (Martincorena et al, 2015b, 2018a) (Fig. 2C). As expected, we observed that the total number of mutations per 2 mm² of epidermis increased with the duration of UV exposure, with a sevenfold increase between the groups exposed for 2 compared to 12 weeks (Fig. 2C, left). Along the same lines, the epidermal mutation burden, quantified as the number of synonymous substitutions per megabase, increased 30-fold between the groups exposed for 2 versus 12 weeks (Fig. 2C, middle). However, we observed no difference either in the total number of mutations or the mutation burden between male and female epidermis at any time point except for the total number of mutations 12 weeks after repeated UV exposure (Fig. 2C, left and middle). Despite a higher number of mutations observed in the female epidermis compared with males 12 weeks after repeated UV exposure, neither the mutational burden nor the proportion of CC > TT UV-signature mutations was increased, suggesting that the additional mutations are attributable to

the accumulation of low-level background mutations rather than to enhanced UV-induced mutagenesis. We compared the mutational spectrum of our data to the mutational signatures SBS7a and SBS7b associated with exposure to UV light (COSMIC Mutational Signatures, 2024). These signatures are characterized by cytosine substitution by thymidine (C > T) at TCN (SBS7a) or CCN/ TCN (SBS7b) trinucleotide sites, that may reflect different pyrimidine-dimer photoproducts induced by UV exposure (Appendix Fig. S1A). In analyzing these signatures, we observed no difference in the proportion of mutations assigned to UV associated signatures between the sexes (Appendix Fig. S1B). To collect more detailed information, we next quantified the CC to TT double base substitutions which are strongly associated with exposure to UV light. The number of CC to TT mutations increased with the duration of UV exposure at a similar rate between males and females (Fig. 2C, right). We also examined selection in this mutated landscape of synonymous, missense, nonsense and indel mutations, using dN/dS. Mutations in genes that confer a fitness advantage have a dN/dS ratio greater than 1 while those that are detrimental to cell fitness have a dN/dS ratio of less than 1. Examining the entire dataset showed that several mutant genes were under positive selection (Appendix Fig. S1C), however comparing the strength of selection

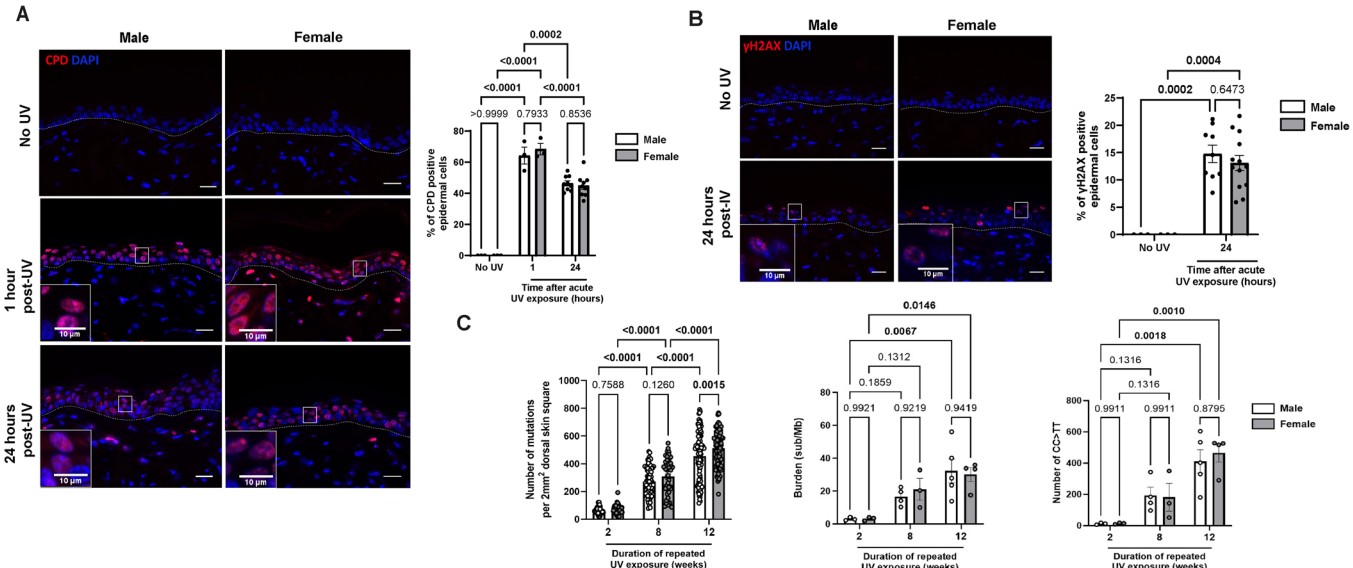

**Figure 2. UV exposure induces similar level of epidermal DNA damage and UV-signature mutations in male and female mice epidermis.**

(A) Left: CPD (red) immunofluorescence staining in male and female dorsal skin collected 1 h and 24 h after a single dose of acute UV exposure (120 mJ/cm²), compared to control skin (No UV). DAPI was used as counterstaining (blue). The dotted line separates the epidermis from the dermis. Scale bars: 20 μm. Right: Percentage of CPD positive epidermal cells. n(fields) = 4 per mouse, n = 3–9 mice, mean ± SEM, two-way ANOVA with Holm–Šidák post hoc test. (B) Left: γH2AX (red) immunofluorescence staining in male and female dorsal skin collected 24 h after a single dose of acute UV exposure (120mJ/cm²), compared to control skin (No UV). DAPI was used as counterstaining (blue). The dotted line separates the epidermis from the dermis. Scale bars: 20 μm. Right: Percentage of γH2AX foci staining pattern positive epidermal cells. n(fields) = 4 per mouse, n = 3–9 mice, mean ± SEM, two-way ANOVA with Holm–Šidák post hoc test. (C) Left: Number of mutations detected per 2 mm² in male and female dorsal epidermis exposed to UV (70 mJ/cm²) for 2, 8, or 12 weeks, n(square) = 19–22 per mouse, n = 3–5 mice, mean ± SEM, two-way ANOVA with Holm–Šidák post hoc test. Middle: Mutation burden in male and female dorsal epidermis collected from mice exposed to UV (70 mJ/cm²) for 2, 8, or 12 weeks, n = 3–5 mice, mean ± SEM, two-way ANOVA with Holm–Šidák post hoc test. Right: Total counts of CC > TT mutations in male and female dorsal epidermis collected 2, 8, or 12 weeks after multiple doses of UV exposure (70 mJ/cm²), n = 3–5 mice, mean ± SEM, two-way ANOVA with Holm–Šidák post hoc test. Source data are available online for this figure.

between sexes at the different time points showed no significant differences (Appendix Fig. S1D).

Overall, these data show that the differences in atypia, tumor development and tumor severity observed in males and females after UV exposure cannot be explained by differences in UV-induced DNA lesions or mutation patterns, nor by the nature of the mutant genes that are selected.

## UV exposure affects epidermal proliferation in males, while it affects differentiation in females

It has been shown that exposure of the skin to UV rays affects the rate of epidermal keratinocyte proliferation and the proportion of dividing cells (El-Abaseri et al, 2006), a response which must always be balanced with differentiation to maintain epidermal homeostasis (de Pedro et al, 2018). We then asked if an imbalance between proliferation and differentiation could be involved in the contrasting UV response in male and female epidermis. After a single acute UV exposure (UV 120 mJ/cm²), we quantified epidermal proliferation in UV-exposed male and female epidermis, using Ki67 labeling of skin sections (Fig. 3A). Examination of control samples showed that background cell proliferation was similar in male and female epidermis, with 24.8% and 27.4% of Ki67-positive epidermal cells, respectively. We observed a proliferative response of epidermal layers in two steps. First, an early response 24 h after UV exposure, when we quantified a significant decrease in epidermal cell

proliferation, which was similar in male and female epidermis (Fig. EV2A). Second, a later response 72 h after UV exposure, when we observed a strong increase in epidermal cell proliferation in male epidermis only, while in females, the proliferation rate returned to the unexposed control (Fig. 3A). Of note, while the proliferation rate had returned to control level 72 h after UV exposure in female epidermis, males did not turn off proliferation in epidermal suprabasal layers (spinous layer specifically) at that time point (Fig. 3A). This proliferation of keratinocytes in the layers above the basal layer is a characteristic feature of AK (Roewert-Huber et al, 2007). At the mRNA level, these results were confirmed by an increase in Mki67 mRNA expression only in male epidermis (Fig. EV2B). Five days after a single UV exposure, epidermal thickness (Appendix Fig. S2A) and epidermal proliferation (Appendix Fig. S2B) returned to baseline in both sexes, with no difference observed between sexes. These findings indicate that acute UV exposure induces transient sex-dependent differences detectable at day 2 and 3, reflecting a difference in response amplitude rather than in response kinetics, based on the time points we analyzed.

In healthy skin, epidermal homeostasis is supported through a well-balanced equilibrium between proliferation and differentiation. Keratinocytes proliferate in the basal layer of the epidermis, then they exit cell cycle and undergo differentiation while migrating up into the suprabasal layers (Blanpain and Fuchs, 2006; Jones et al, 2007). We next studied the proliferation/differentiation equilibrium

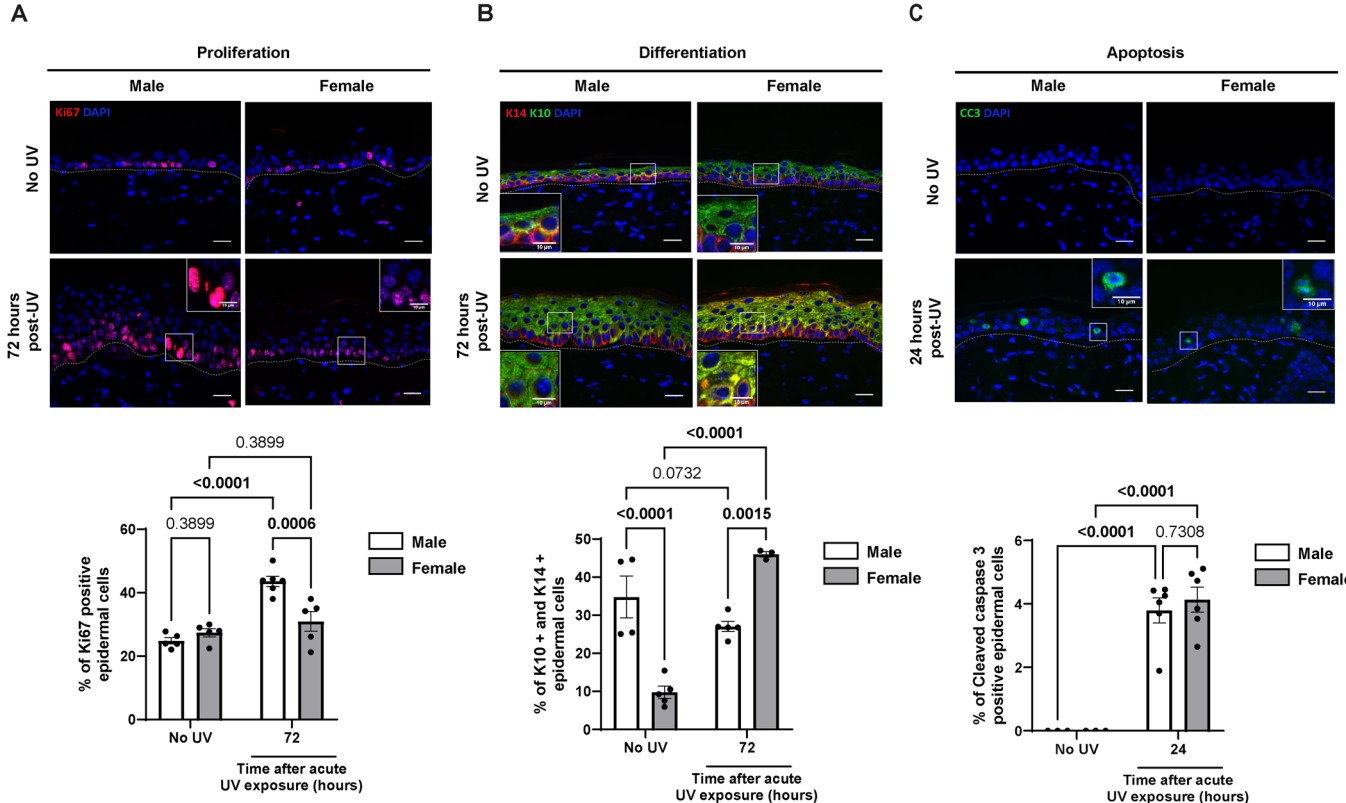

Figure 3. UV exposure affects epidermal proliferation in males and differentiation in female mice.

(A) Top: Ki67 (red) immunofluorescence staining in male and female dorsal skin collected 72 h after a single dose of acute UV exposure (120 mJ/cm²), compared to control skin (No UV). DAPI was used as counterstaining (blue). The dotted line separates the epidermis from the dermis. Scale bars: 20 μm. Bottom: Percentage of Ki67-positive keratinocytes. n(fields) = 4 per mouse, n = 5 mice, mean ± SEM, two-way ANOVA with Holm–Šidák post hoc test. (B) Top: Keratin 14 (K14; red) and Keratin 10 (K10; green) immunofluorescences staining in male and female dorsal skin collected 72 h after a single dose of acute UV exposure (120 mJ/cm²), compared to control skin (No UV). DAPI was used as counterstaining (blue). The dotted line separates the epidermis from the dermis. Scale bars: 20 μm. Bottom: Percentage of double K10-positive and K14-positive epidermal keratinocytes. n(fields) = 4 per mouse, n = 3 to 6 mice, mean ± SEM, two-way ANOVA with Holm–Šidák post hoc test. (C) Top: Cleaved caspase 3 (CC3; green) immunofluorescence staining in male and female dorsal skin collected 24 h after a single dose of acute UV exposure (120 mJ/cm²), compared to control skin (No UV). DAPI was used as counterstaining (blue). The dotted line separates the epidermis from the dermis. Scale bars: 20 μm. Bottom: Percentage of CC3-positive keratinocytes and quantification. n(fields) = 4 per mouse, n = 3–6 mice, mean ± SEM, two-way ANOVA with Holm–Šidák post hoc test. Source data are available online for this figure.

in response to UV exposure, using Keratin 14 (K14) and Keratin 10 (K10) as prototypic markers to label basal/dividing and suprabasal/non-dividing layers of keratinocytes, respectively (Fig. 3B). As suggested by both morphology and the percentage of cells expressing only K14 (K10 negative and K14-positive), neither UV exposure nor sex seems to affect the keratinocytes of basal layer (Fig. EV2D). The co-expression of K14 and K10 is usually considered as abnormal and the expansion of the expression of the basal keratinocyte marker K14 to suprabasal layers suggests a failure of keratinocytes to shut down K14 expression upon differentiation. In absence of UV, we saw that male epidermis presented a greater proportion of keratinocytes expressing both K14 and K10 compared to female epidermis (Fig. 3B). In response to UV, we observed an increase of K10 and K14 marker co-expression in the suprabasal layers both in male and female epidermis (Fig. EV2C). Importantly, the abnormal co-expression of K10 and K14 persisted and increased 72 h after UV exposure only in female epidermis (Fig. 3B), which suggests that the return to a

normal differentiation process differs in female epidermis. In contrast, increased co-expression of K14 and K10 in response to UV was transient in male epidermis (compare Fig. 3B and EV2C). In agreement with this observation, the percentage of K10-positive and K14-negative epidermal cells was decreased after UV exposure only in females (Fig. EV2D).

UV exposure is also well-known to provoke apoptosis (Lee et al, 2013), which is also involved in the maintenance of epidermal homeostasis. To quantify apoptosis in UV-exposed epidermis, we performed Cleaved caspase 3 (CC3) staining on histological sections of male and female epidermal skin 24 h after a single UV exposure. We observed no difference in the number or localization of apoptotic epidermal cells (CC3-positive) between male and female epidermis (Fig. 3C).

In brief, we discovered that two major cellular responses to UV exposure are differentially influenced by sex. In males, UV exposure primarily impacts proliferation, whereas in females, it predominantly affects epidermal differentiation.

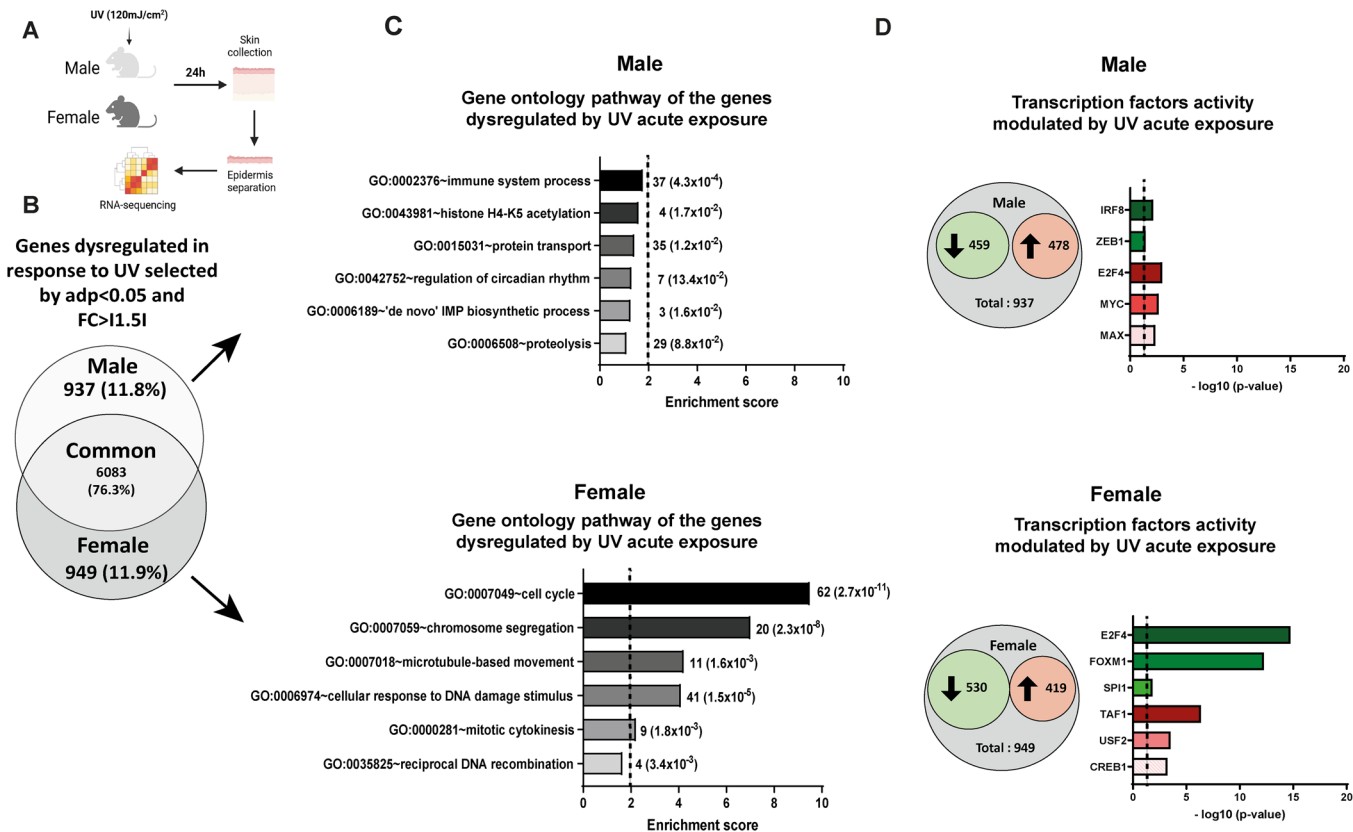

**Figure 4. Acute UV exposure triggers female-specific downregulation of cell cycle pathway and E2F in the epidermis.**

(A) Experimental flowchart. (B) Venn diagram illustrating the number of genes with significant changes in expression (adjusted P value < 0.05 and FC > |1.5|) in male and female mouse epidermis 24 h post-UV exposure (single dose; 120 mJ/cm²). (C) Gene ontology pathway analysis of the genes significantly dysregulated in response to acute UV exposure specifically in male, 937 genes (top) and in female, 949 genes (bottom) samples. Histograms show the enrichment score for each identified pathway. Gene number and the corresponding P value are indicated to the right of the histograms. (D) Histograms showing the number of downregulated genes—459 in male and 530 in female—and upregulated genes—478 in male and 419 in female—specifically in male (top) or female (bottom) samples, accompanied by transcription factor enrichment analysis of the dysregulated genes in each sex. Source data are available online for this figure.

## Sex-specific modulation of epidermal cell cycle and E2F transcription factors in response to UV

Our analyses so far show intrinsic differences in the epidermal response to acute and chronic UV exposure between male and female mice in terms of tumor incidence and progression, as well as proliferation and differentiation. To investigate the molecular mechanisms involved in this sex-specific response to UV, we analyzed the transcriptome of male and female mouse epidermis exposed to UV using RNA sequencing (four epidermal samples per sex, non-UV exposed controls and 24 h after UV exposure) (Fig. 4A). The 24 h post UV exposure time point was chosen based on (i) the peak of atypia in both males and females (Fig. 1) and (ii) an increase of *Il1b* and *Tgfb-1* expression, two cytokines known to be induced in response to UV exposure and used here as a marker of effective epidermal response (Ravindran et al, 2014) (Appendix Fig. S3). Using hierarchical clustering, differentially expressed genes in male (Appendix Fig. S4) and female (Appendix Fig. S5) epidermis were mapped. 9658 and 9703 genes were identified as differentially regulated in response to a single dose of UV in male and female epidermis, respectively (adjusted P value < 0.05).

Differentially regulated genes were grouped into three clusters in male epidermis, with clusters I and II including downregulated genes and cluster III upregulated genes in response to UV (Appendix Fig. S4). Four clusters of differentially regulated genes were identified in female epidermis: clusters I and II with downregulated genes and clusters III and IV with upregulated genes in response to UV (Appendix Fig. S5). Reactome Pathway Enrichment analysis revealed that most of the genes with UV-induced upregulated expression were similar in male and female epidermis and were enriched in "ribonucleoprotein complex biogenesis", "RNA splicing", "mitochondrial gene expression", "regulation of apoptotic signaling" and "keratinocyte differentiation" pathways (Appendix Figs. S4 and S5). Exceptions were the "response to oxidative stress" and "positive regulation of defense response" pathways that were upregulated only in male or only in female epidermis, respectively, in response to UV. In contrast, the genes whose expression was reduced by UV exposure were mainly sex-specific. "Mitotic cell cycle phase transition", "positive regulation of cell cycle", "double strand break repair" and "cell cycle checkpoint signaling" were the most significantly downregulated pathways only in the epidermis of female mice (Appendix Fig. S5).

In male epidermis, the two most downregulated pathways were "lymphocyte differentiation" and "regulation of T cell activation" (Appendix Fig. S4).

A Venn diagram of the differentially expressed genes (DEGs; adjusted $P$ value < 0.05; Fold change IFCI > 1.5), illustrates that the majority of DEGs were common for both male and female epidermis (6083, 76.3%) (Fig. 4B), while approximately 12% of the genes exhibited a sex-specific response to UV (937 and 949 in male and female epidermis, respectively). Gene ontology analysis of these sex-specific DEGs revealed that a majority belonged to "immune system process" (37), "protein transport" (35) and "proteolysis" (29) in male epidermis (Fig. 4C, top) and to "cell cycle" (62) and "cellular response to DNA damage stimulus" (41) in female epidermis (Fig. 4C, bottom). We then focused on the most significant groups of genes, with the highest enrichment score and showing a sex-specific regulation, that is the group of 62 genes dysregulated upon UV exposure in female epidermis only, belonging to "cell cycle" (GO:0007049; enrichment score of 9.5). To identify transcription factors associated with this group of DEGs, we analyzed the up- and downregulated sex specific genes separately referring to ENCODE (Dunham et al, 2012) and ChEA (Lachmann et al, 2010) Consensus TFs from ChIP-X category in Enrichr (Chen et al, 2013) (Fig. 4D). In male epidermis exposed to UV, we found an enrichment of upregulated genes associated with three transcription factors namely E2F4, the proto-oncogene MYC and MAX, while downregulated genes were associated with the two transcription factors IRF8 and ZEB1 (Fig. 4D, top). In female epidermis exposed to UV, upregulated genes were associated with the three transcription factors TAF1, USF2 and CREB1, while downregulated genes were associated with the three transcription factors E2F4, FOXM1 and SPI1 (Fig. 4D, bottom). The most significant enrichment was observed for E2F4 where 60 out of its 710 target genes where among the downregulated genes in UV exposed female epidermis. The list of target genes identified for each transcription factor—downregulated (Tables EV1 and EV2) and upregulated (Tables EV3 and EV4) in response to acute UV exposure—is presented separately for each sex. As shown in Table EV2, the transcription factor E2F4 exhibits the highest number of identified deregulated target genes ($n = 60$), with the strongest statistical significance (adjusted $P$ value = $1.82 \times 10^{-13}$). Notably, many of these target genes overlap with those of other E2F family members: among the 60 E2F4 targets, 43 are shared with E2F7 and E2F8 (71.7%), 39 with E2F1 (65%), 38 with E2F2 (63.3%), and 22 with E2F3 (36.6%) (Keenan et al, 2019).

Overall, we identified that 12% of the total genes deregulated in response to UV are sex-specific. Importantly, we identified the transcription factor E2F, a major regulator of the cell cycle whose activity is repressed in response to UV specifically in female epidermis.

## Estrogen receptors-mediate UV-induced modulation of CDKN3 in the epidermis of female mice and in squamous cell carcinoma

The E2F family of transcription factors particularly caught our attention as key regulators of the cell cycle, for which sex-specific regulation could explain the sex-specific proliferation and differentiation responses that we observed following UV exposure (Fig. 3). There are seven subgroups in the E2F family. E2F1-3 are required for normal cell cycle progression, while in contrast E2F4

and 5 are mainly involved in cell cycle exit and differentiation. The roles of E2F6 and 7 are still unclear, but they are believed to act as transcriptional repressors of the cell cycle (Attwooll et al, 2004). In quantifying the expression of the transcription factors themselves, we found that $E2f1$ and $E2f2$ gene expression was decreased in response to UV only in female epidermis (Figs. 5A and EV3A), while the expression of $E2f3$, $E2f6$ and $E2f7$ genes was not affected by sex (Fig. EV3A). As for $E2f4$ and $E2f5$ expression, we observed that it was significantly increased in UV-treated females compared to males, but no difference was observed in their basal levels. In line with these observations, the expression of the E2F target genes, the proto-oncogene $Cdkn3$ and $Cdc25c$, which are mainly involved in cell cycle progression, were also downregulated in response to UV only in female epidermis (Figs. 5A and EV3B).

To test whether the female-specific downregulation of $E2f1$, $E2f2$, $Cdkn3$, and $Cdc25c$ genes in response to acute UV exposure was dependent on estrogen receptors, female mice were treated with fulvestrant, a pharmacological Estrogen Receptors antagonist, 24 h prior to UV exposure. The effectiveness of Estrogen Receptors antagonism was demonstrated by the expected absence of expression of Estrogen Receptor α in the epidermis of fulvestrant-treated female mice on the one hand (Fig. EV4A), and by a reduction in uterine weight in these mice on the other hand (Fig. EV4B). The effectiveness of UV exposure was confirmed by an increase in epidermal $Il1b$ gene expression (Ravindran et al, 2014) (Fig. EV4C).

In line with Figs. 5A and EV3A,B, we found that UV exposure downregulated the expression of $E2f1$, $E2f2$, $Cdkn3$, and $Cdc25c$ in the solvent control condition (Figs. 5B and EV4D). Significantly, when Estrogen Receptor α were degraded following fulvestrant treatment, this downregulation was abolished for $E2f1$ and $Cdkn3$, the typical UV-induced decrease was entirely lost, demonstrating that the UV-induced reduction in the expression of these genes is mediated by Estrogen Receptors (Fig. 5B). In order to test whether the regulation of the expression of these genes by estrogen is direct or not, we treated human immortalized keratinocyte cell lines (HaCaT) with increasing concentrations of β-estradiol (E2). Our results show that the only lowest concentration of estradiol (0.1 μM) we used induced the expression of a well-known Estrogen Receptors target gene ($TFF1$), as well as the transcription factor $E2F1$ (Appendix Fig. S6). These findings indicate that the lowest levels of estradiol we used selectively activate the $E2F1$ pathway, suggesting a subsequent indirect regulation of $CDKN3$ in human keratinocyte cells.

To assess whether the acute effects of UV exposure on CDKN3 expression persist after chronic exposure, we measured CDKN3 expression in chronically UV-exposed, non-pathological mouse skin in both males and females. We observed an increase in epidermal CDKN3 protein expression only in males in chronically irradiated skin compared to females (Fig. 5C). We further investigated these observations in pre-lesional actinic keratosis (AK) tissues and squamous cell carcinoma (SCC) tumors derived from male and female mice chronically exposed to UV. In SCC samples, we found significantly higher numbers of CDKN3-positive cells in males compared to females, with males exhibiting 15 times more CDKN3-positive cells than females (Fig. 5D). In AK biopsies, a slight increase in CDKN3 expression was detected in males compared to females (Fig. 5D).

As key findings, we identified that the expression of E2F1 and CDKN3— key regulators of the cell cycle, with E2F1 functioning as

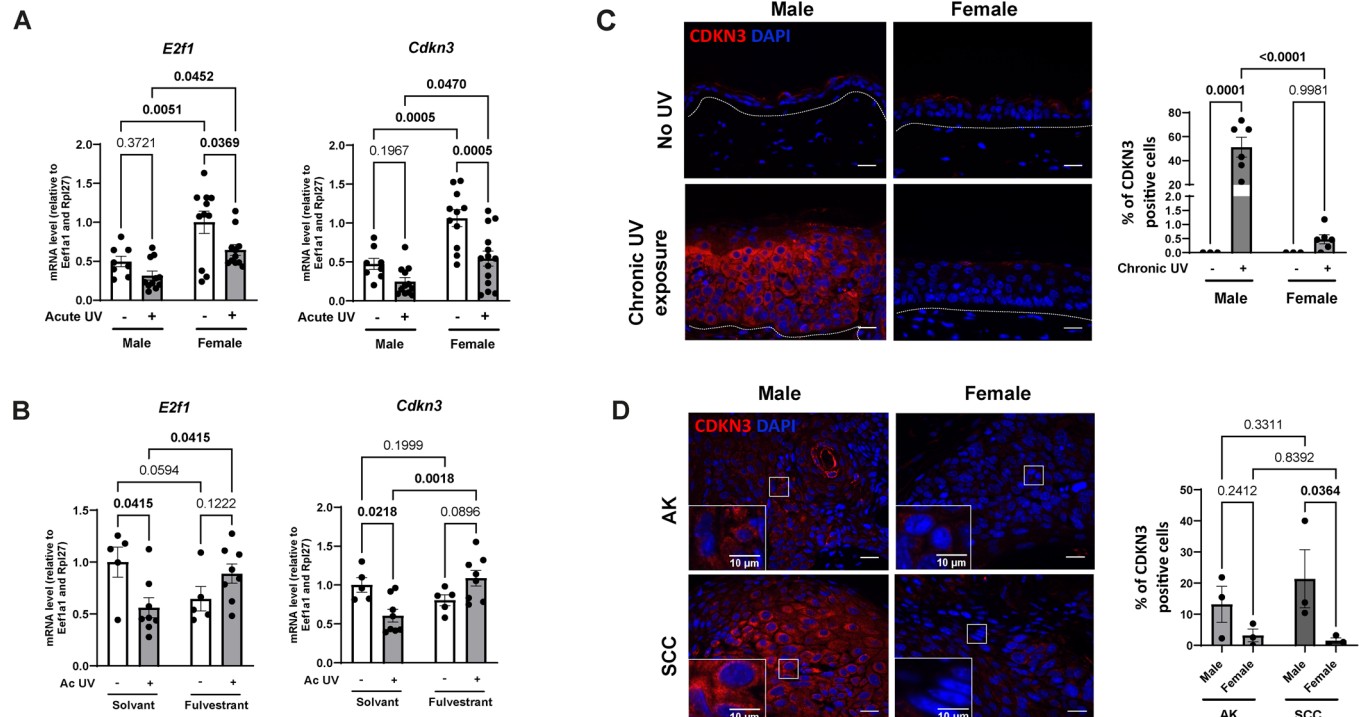

**Figure 5. Estrogen receptor-mediated UV-induced downregulation of the proto-oncogene CDKN3 in the female epidermis and in squamous cell carcinoma.**

(A) Quantification of the relative gene expression of *E2f1, Cdkn3* at mRNA level by RT-qPCR in epidermal samples from male and female mice. n = 8–14 mice per sex, mean ± SEM, two-way ANOVA with Holm–Šidák post hoc test. (B) Quantification of the relative gene expression of *E2f1* and *Cdkn3* at the mRNA level by RT-qPCR in epidermal samples from female mice treated with solvent or fulvestrant (150 mg/kg) for 48 h prior to acute UV exposure. Data represent the mean values for n = 5 mice (solvent) and n = 8 mice (fulvestrant), mean ± SEM, two-way ANOVA with Holm–Šidák post hoc test. (C) Left: CDKN3 (red) immunofluorescence staining in male and female dorsal skin following chronic UV exposure (3 times per week for 6 months). DAPI was used as counterstaining (blue). The dotted line separates the epidermis from the dermis. Scale bars: 20 μm. Right: Percentage of CDKN3-positive epidermal keratinocytes. n(fields) = 4 per mouse, n = 3–6 mice, mean ± SEM, two-way ANOVA with Holm–Šidák post hoc test. (D) Left: CDKN3 (red) immunofluorescence staining in actinic keratosis (AK) and squamous cell carcinoma (SCC) biopsies collected from male or female mice following chronic UV exposure. DAPI was used as counterstaining (blue). Scale bars: 20 μm. Right: Percentage of CDKN3-positive cells. n(fields) = 4 per mouse, n = 3 tumors per sex, mean ± SEM, two-way ANOVA with uncorrected Fisher's LSD post hoc test. Source data are available online for this figure.

---

a major transcription factor and CDKN3 as a proto-oncogene—was significantly reduced in response to UV exposure, only in females. Mechanistically, this female-specific UV-induced downregulation of *E2f1* and *Cdkn3* is mediated by estrogen receptors. Notably, in SCC tissues, we observed a persistent sex-specific difference in CDKN3 protein expression, with significantly higher levels in males compared to females.

## Low CDKN3 expression is associated with women skin response to UV, women SCC and improved prognosis in women HNSC patients

The above findings suggest that differences in susceptibility of men and women to UV-induced skin malignancies is routed in biological and physiological differences, rather than behavioral differences. To go beyond our mouse model and to test this hypothesis in human tumors and skin, we performed a series of experiments on ex vivo human skin explants, and we analyzed malignant skin lesions from men and women patients. First, we assessed the responses of skin explants with phototypes I and II (Fitzpatrick scale) harvested from healthy women having abdominoplasty procedures, collected 24 h and 72 h after single increasing

doses of UV (either 240, 480, or 720 mJ/cm²). UV exposure led to a decrease in proliferation 24 h and 72 h after UV exposure (Fig. 6A), in line with the reduction in epidermal proliferation we also observed in female mouse epidermis 24 h after UV exposure (Fig. EV2A). Unlike what we observed in male mice, but consistent with our observation in female mice, there was no increase in proliferation in women human skin 72 h after UV exposure (Fig. 3A). The expression of *E2F1, E2F2* and *CDKN3* mRNAs was also significantly reduced, and *CDC25C* mRNA showed a tendency to decrease in a UV dose-dependent manner, with *E2F2* showing a 2.7-fold decrease in response to the highest dose of UV (Figs. 6B and EV5B). The effectiveness of UV exposure on the different ex vivo explants was confirmed by an increase in epidermal Il1b gene expression, used as a marker of UV-induced inflammation (Ravindran et al, 2014) (Fig. EV5A).

We then proceeded with a retrospective analysis of the genomic data in two independent sets of human samples. The GSE66412 (Farshchian et al, 2015; Nissinen et al, 2017) set of data includes 8 cutaneous SCC cell lines and 5 normal human epidermal keratinocytes (NHEK) cell lines; the GSE84293 (Chitsazzadeh et al, 2016a) patient cohort includes Normal Skin (NS), Actinic Keratosis (AK) and Squamous cell carcinoma (SCC). In line with

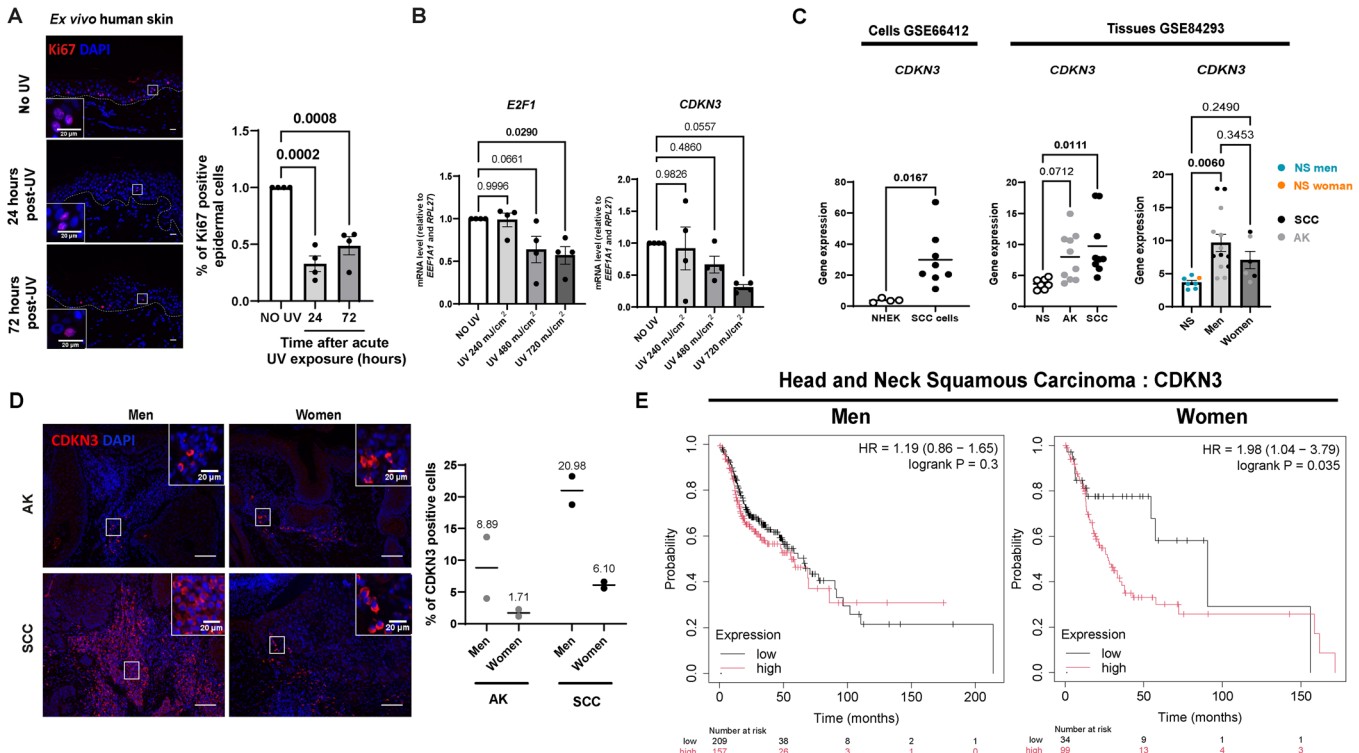

**Figure 6. Downregulation of CDKN3 proto-oncogene expression in the skin of healthy women and in squamous cell carcinoma.**

(A) Left: Ki67 (red) immunofluorescence staining in human ex vivo skin explant cultures from woman subjects, 24 h and 72 h after exposure to a single dose of UV (720 mJ/cm²), compared to non-UV-exposed control explants. DAPI was used as counterstaining (blue). The dotted line separates the epidermis from the dermis. Scale bars: 50 μm. Right: Percentage of Ki67-positive epidermal cells. n(fields) = 4 per subject, n = 4 subjects, mean ± SEM, two-way ANOVA with Tukey's post hoc test. (B) Relative mRNA expression of E2F1 and CDKN3 in ex vivo human skin explant cultures from woman subjects, 24 h after exposure to UV with increasing intensities (240, 480 and 720 mJ/cm²), compared to non-UV-exposed control explants, quantified by RT-qPCR. n = 4 subjects, mean ± SEM, two-way ANOVA with Tukey's post hoc test. (C) Left: CDKN3 transcript level in normal human epithelial keratinocytes (NHEK) and human Squamous Cell Carcinoma (SCC) from public datasets (GSE66412). NHEK: n = 4, SCC: n = 8. Mean ± SEM, unpaired t test. Right: CDKN3 transcript level in normal human skin (NS), actinic keratosis (AK) and squamous cell carcinoma (SCC) lesions from men and women in public datasets (GSE84293). NS: n = 7 (NS; 6 men and 1 woman), AK: n = 10 (AK; 7 men and 3 women), SCC n = 9 (SCC; 5 men and 3 women). The right graph shows CDKN3 expression separated by sex for AK and SCC lesions. Mean ± SEM, one-way ANOVA with Tukey's post hoc test for non-sex-separated graph, two-way ANOVA with Holm–Šidák post hoc test for sex-separated graphs. (D) Left: CDKN3 (red) immunofluorescence in AK and SCC biopsies collected from men or women patients. DAPI was used as counterstaining (blue). Scale bars: 100 μm. Right: Percentage of CDKN3-positive cells. n = 2 tumors per sex, for each 3 fields were counted. Means are shown. (E) Kaplan–Meier survival curves for men (left) or women (right) patients with Head and Neck Squamous Cell Carcinoma (HNSC), based on CDKN3 gene expression. Source data are available online for this figure.

the experimental data we collected, SCC cell lines exhibited overexpression of *CDKN3* and *CDC25C* mRNA compared to healthy NHEK cells (Figs. 6C and EV5C). In the second dataset, we observed a similar significant increase in the expression of *CDKN3* and *CDC25C* mRNA in SCC compared to healthy skin. While there was an increasing trend in the expression of *CDKN3* and *CDC25C* mRNA in AK compared to healthy skin, too, this was however not significant. Of much interest, the increased expression of *CDKN3* and *CDC25C*, was significant in men AK and SCC lesions compared to healthy tissue (Figs. 6C and EV5D), which is consistent with what we previously observed in AK and SCC mice.

We next deepened these observations by analyzing the expression of CDKN3 protein in additional AK and SCC lesions derived from men and women patients. CDKN3 staining of tissue sections of men and women AK (n = 2) and SCC (n = 2) biopsies revealed that men lesions had 3-times more CDKN3-positive cells than women lesions (Fig. 6D). Moreover, the increase in CDKN3-

positive cells in SCC compared to AK was more prominent in men that in women tissues sections.

Finally, we investigated whether the expression level of the E2F target genes *CDKN3* and *CDC25C* was correlated with patient survival rates. To do so, because cutaneous SCC has a very high survival rate (Alam et al, 2018), survival analysis is not a relevant indicator for this type of cancer. We therefore turned to another form of carcinoma, highly aggressive, head and neck squamous cell carcinomas (HNSC), the SCC subtype with the closest molecular profile to cutaneous SCC (Chitsazzadeh et al, 2016b), as the most relevant comparable epithelial cancer model. Available public databases, show that the expression of *CDKN3* and *CDC25C* genes in patients with HNSC, is negatively correlated with patient survival rate in women, and not in men (Figs. 6E and EV5E) (Nagy et al, 2021).

Overall, our data suggest that CDKN3 and CDC25C play a role in the difference in susceptibility between men and women to UV-induced skin cancers. Moreover, their lower expression in women,

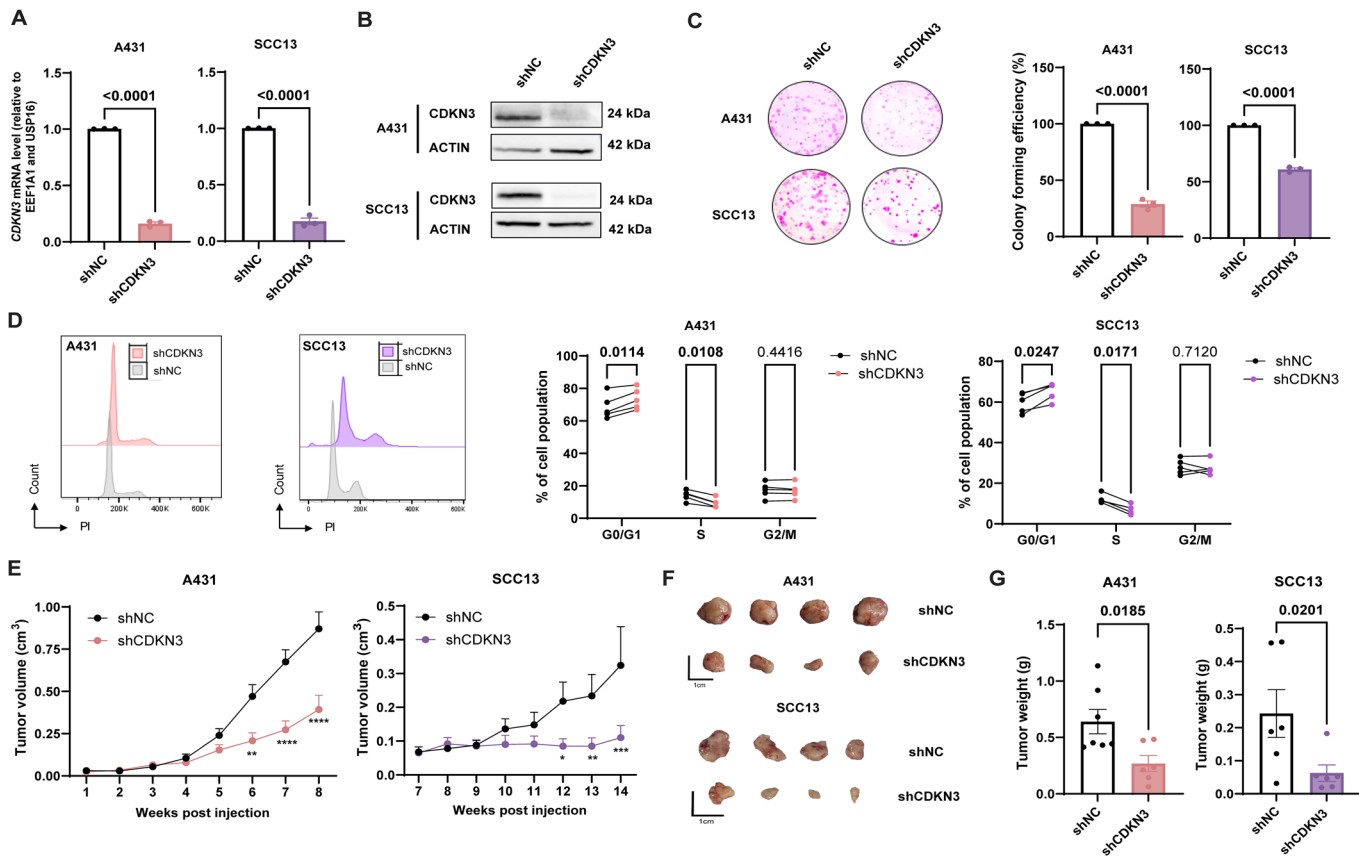

**Figure 7. CDKN3 depletion blocks cell cycle progression and reduces tumor growth in A431 and SCC13 cells.**

(A) Relative *CDKN3* mRNA expression in shRNA-mediated knockdown (shCDKN3) versus negative control (shNC) cells, assessed by qPCR. $n = 3$ independent experiments, mean ± SEM, unpaired *t* test. A431: *P* value < 0.0001 = 0.000000691, SCC13: *P* value < 0.0001 = 0.000009170. (B) CDKN3 protein levels in A431 and SCC13 cells after shCDKN3 or shNC transduction, analyzed by immunoblotting. (C) Left: Representative pictures of colony formation assay following CDKN3 knockdown in A431 and SCC13 human cells. Right: Percentage of colony-forming efficiency after CDKN3 depletion (shCDKN3) and control (shNC). $n = 3$ independent experiments, mean ± SEM, unpaired *t* test. A43: *P* value < 0.0001 = 0.000013176, SCC13: *P* value < 0.0001 = 0.000008998. (D) Left: Representative histograms showing cell cycle phase distribution of A431 cells and SCC13 cells after CDKN3 depletion (shCDKN3) and control (shNC). Right: Average percentages of cell cycle phases. $n = 5$ independent experiments, multiple paired *t* tests. (E) Growth curves of subcutaneous xenografts derived from human A431 and SCC13 cells after CDKN3 depletion (shCDKN3) or control (shNC). $n = 7$ mice per group, mean ± SEM, two-way ANOVA followed by Sidak's multiple comparisons test. *P* value < 0.05, **P* value < 0.01, ***P* value < 0.001, ****P* value < 0.0001. (F) Representative images of tumors collected 8 weeks after injection of human CDKN3-depleted (shCDKN3) A431 cells (top panel) and 14 weeks after injection of human CDKN3-depleted (shCDKN3) SCC13 cells (bottom panel), compared with respective controls (shNC). (G) Tumor weights collected 8 weeks after injection of human CDKN3-depleted (shCDKN3) A431 cells (left panel) or 14 weeks after injection of human CDKN3-depleted (shCDKN3) SCC13 cells (right panel), compared with their respective control (shNC). $n = 7$ mice per group, mean ± SEM, unpaired *t* test. Source data are available online for this figure.

might have a protective effect against the development of skin SCC.

## CDKN3 depletion blocks cell cycle progression and in vivo SCC tumor growth

We therefore evaluated the impact of CDKN3 depletion, which is recognized as proto-oncogene in some cancers (Gao et al, 2023) on the tumorigenicity of two human squamous carcinoma cell lines, A431 and SCC13, derived from female and male skin carcinoma, respectively. shRNA-mediated knockdown (shCDKN3) reduced CDKN3 mRNA levels by approximatively 83% in SCC13 cells and by 84% in A431 cells compared with scramble negative control (shNC) (Fig. 7A). This reduction in mRNA expression results in decreased protein levels as observed by western blotting (Fig. 7B). CDKN3-depleted A431 and SCC13 cells showed a strong reduction

in colony-forming potential (Fig. 7C), as well as impaired progression into the cell-cycle S phase (Fig. 7D) compared to control cells (shNC), indicating that lower CDKN3 expression reduces their proliferative capacity. Using an in vivo xenograft model of A431 and SCC13 cells in nude female mice, we showed that stable shRNA-mediated depletion of CDKN3 significantly inhibited tumor growth of both cell lines (Fig. 7E–G). Indeed, eight and fourteen weeks after cell injection, tumors volume were reduced by 60.9% and 66.1% in tumors from A431 and SCC13 CDKN3-depleted cells, respectively, while size of these tumors was reduced by 0.53 cm³ and 0.21 cm³, compared with tumor from control cells (shNC). The stable reduction in CDKN3 expression in A431 and SCC13 tumor cells following tumor resection at the end of the experiment was confirmed by qPCR (Appendix Fig. S7).

Collectively, these data indicate that decreased CDKN3 levels have a protective effect against SCC development, most likely

through blocking G1/S phase transition and thus blocking cell cycle progression (Li et al, 2014; Wang et al, 2017).

# Discussion

Our study provides new insights into intrinsic differences between male and female epidermal responses to both acute and chronic UV exposure, at the tissue, cellular and molecular levels. Our data emphasize that extrinsic factors, such as behavior (Roberts et al, 2021), are not the sole contributors to the sex-disparity in SCC incidence. Indeed, chronic UV exposure leads to the development of more numerous and aggressive tumors in male compared to female mice, in line with findings from epidemiological studies in human (Nagarajan et al, 2019).

Epidermal proliferation is the most parameter we found differentially regulated with UV between male and female murine skins. Indeed, proliferation is greater in males in response to a single dose of UV, consistent with previous similar observations made with another carcinogenic factor (DMBA/TPA; although less relevant for human cancers) (Budden et al, 2021). Prolonged epidermal proliferation and hyperplasia are sufficient to promote the formation of early premalignant actinic keratosis (AK) lesions (Azazmeh et al, 2020). Along the same lines, UV is known to activate epidermal proliferation and hyperplasia through activation of the epidermal growth factor receptor (EGFR) (El-Abaseri et al, 2006). EGFR inhibition thus efficiently suppressed UV-induced epidermal proliferation, ultimately decreasing tumorigenesis (El-Abaseri et al, 2005). These data collectively show that the regulation of epidermal proliferation in response to UV is a critical factor of UV-induced cancer development. Our study highlights that, in the long term, sex-dependent proliferative response to UV is likely crucial for sex-specific SCC development.

We observed more severe UV-induced epidermal atypia in female mice compared to males. This atypia was associated with altered epidermal differentiation, consistent with previous findings showing that defects in the differentiation process leads to epidermal disorganization and atypical architecture (Li et al, 2020). In males, we observed that this effect was transient, as their differentiation capacity was restored 72 h after UV exposure. In contrast, the prolonged disruption of the differentiation process in females likely impairs epidermal stratification, thereby contributing to the more pronounced epidermal atypia observed in female epidermis. Moreover, the higher proliferation rate observed in male epidermis following acute UV exposure likely supports effective epidermal stratification and, consequently, reduced atypia. Indeed, a previous study demonstrated that during epidermal development, higher proliferation rates, particularly in basal and early suprabasal layers are associated with the formation of a properly stratified epidermis (Damen et al, 2021). The combination of impaired epidermal differentiation in females and increased epidermal proliferation in males could therefore explain the sex-dependent differences in epidermal atypia observed in our study.

At the molecular level, we further discover that sex-specific difference in epidermal proliferation is underpinned by an estrogen receptor-dependent and sex-specific regulation of the activity of the E2F1 transcription factor. The crucial role of E2F1 in controlling keratinocyte growth is well-illustrated by data showing that E2F1$^{-/-}$ mouse primary keratinocytes cannot be maintained in culture

(D'Souza et al, 2002; Thurlings et al, 2016). In contrast, high expression and activity of E2F transcription factors were observed in many tumors and are usually associated with oncogenesis and poor prognosis (Hanahan and Weinberg, 2000; Kent et al, 2017; Kent and Leone, 2019). In the skin, E2F was identified as a key transcriptional driver of squamous cell carcinoma development, whose activity is increased during the transition from healthy skin to actinic keratosis (Chitsazzadeh et al, 2016a). Meanwhile, in skin cutaneous melanoma, low levels of E2F1 and E2F2 were associated with better prognosis (Tan and Lu, 2023). E2F1 may thus be considered as a potential target for the treatment of squamous cell carcinoma (Dicker et al, 2000). In support of the female-specific regulation of E2F1, existing research indicates that estrogen receptor alpha promotes E2F1 expression (Louie et al, 2010; Stender et al, 2007) and estradiol regulates the expression of several E2F family members in breast cancer cell line MCF7 (Bourdeau et al, 2008). Our work reveals that E2F1 exhibits decreased expression in response to UV only in the epidermis of female mice, identifying this transcription factor as a major player in the sex-specific difference of SCC development. Moreover, we discover that lower expression and activity of E2F are associated with the development of less numerous and less aggressive SCCs in female mouse epidermis, suggesting a protective role of low E2F activity against SCC development in females.

Among the target genes of E2F transcription factors, we identify CDKN3 as a marker exhibiting sex-specific regulation in response to UV in both healthy epidermis and SCC. In response to acute UV exposure, we observed a downregulation in CDKN3 expression in the epidermis of female mice only, with CDKN3 mRNA levels remaining higher than in male epidermis. We believe that the key difference between males and females lies in the ability to dynamically regulate the expression of CDKN3, rather than in its level of expression, thereby limiting its overexpression during repeated (chronic) UV exposures. The pronounced reduction of CDKN3 expression after chronic UV exposure, observed only in female epidermis, further supports the hypothesis that the ability to downregulate CDKN3 under repeated UV exposure provides a protective effect. CDKN3 encodes a cell cycle regulator that facilitates mitosis and G1/S transition. It is frequently overexpressed in various cancers and has been recognized as a tumor promoter in ovarian, breast, lung, gastric and hepatocellular cancers (Chen et al, 2024; Fan et al, 2015; Gao et al, 2023; Wang et al, 2021). Its high expression is associated with poor overall survival across eight types of cancers (Zhang et al, 2024). However, CDKN3 also exhibits distinct functions that confer a context-dependent role. Indeed, CDKN3 has also been described as a tumor suppressor in other cancers, such as glioblastoma and myelogenous leukemia (Gao et al, 2023).

In the context of skin cancers, CDKN3 has been identified as a marker of progenitor cells associated with the suppression of epidermal differentiation and progression of SCC. Its overexpression was associated with poorly differentiated patient SCC and worse prognosis (Bailey et al, 2023). Here, we demonstrate that UV-induced downregulation of CDKN3 occurs specifically in the female epidermis of both mice and humans, and that this effect is mechanistically mediated by estrogen receptors Using functional and in vivo assays, we further show that CDKN3 experimental depletion in two human SCC cell lines impairs their progression to the cell-cycle S-phase and reduces tumor growth in xenograft models (Hanxu et al, 2019; Wang et al, 2019).

Together, our findings demonstrate that epidermal CDKN3 is regulated in a sex-specific manner in response to UV exposure,

likely contributing to the differential progression and outcomes of SCC between males and females, in mice and human. By identifying E2F1 and CDKN3 as key mediators of the female-specific protection against SCC development in both mice and humans, we provide new insights into molecular basis of sex dimorphism in skin cancer susceptibility. These results highlight the importance of considering sex as a biological variable in skin cancer research and therapy development.

# Methods

### Reagents and tools table

| Reagent/resource | Reference or source | Identifier or catalog number |
|---|---|---|
| **Experimental models** | | |
| Mouse: B6.129-SKH1-Hr[hr] | Montagner et al, 2014 | |
| Mouse: Rj: NMRI Foxn1 nu/nu | Janvier Labs | Rj:NMRI-Foxn1nu/nu |
| A431 | ECACC; 85090402 | RRID:CVCL_0037 |
| SCC13 | Rheinwald et al, 1981 | RRID:CVCL_4029 |
| HaCaT | Cell Line Service; 300493 | RRID:CVCL_0038 |
| **Recombinant DNA** | | |
| pLV[shRNA]-Puro-U6 > hCDKN3 | VectorBuilder | VB250210-1177vhe |
| pLV[shRNA]-EGFP/Puro-U6>Scramble | VectorBuilder | VB010000-9526zpu |
| **Antibodies** | | |
| Rabbit Polyclonal CDKN3 antibody | Abcam | Cat#ab175393 |
| GAPDH (14C10) Rabbit mAb | Cell Signaling Technologies | Cat#2118 |
| Anti-CPD | Cosmo Bio | Cat#NMDND004 |
| Anti-Keratine14 | Covance | Cat#PRB155P |
| Anti-cleaved caspase-3 | Cell Signaling Technologies | Cat#9664 |
| Anti-γH2AX | EMD Millipore | Cat#05-636-25UG |
| Anti-Keratine10 | Progen | Cat#GP-K10 |
| Goat anti-rat A568 | Thermo Fischer Scientific | Cat#A11077 |
| Goat anti-guinea pig A488 | Thermo Fischer Scientific | Cat#A11073 |
| Goat anti-rabbit A568 | Thermo Fischer Scientific | Cat#A11036 |
| Goat anti-mouse A568 | Thermo Fischer Scientific | Cat#A21235 |
| Anti-rabbit IgG, HRP conjugate | Promega | Cat#W4011 |
| **Oligonucleotides and other sequence-based reagents** | | |
| Mouse qPCR Primers | | |
| *Mki67 forward (5′-3′): ACC ATC ATT GAC CGC TCC TT* | This study | N/A |
| *Mki67 reverse (5′-3′): TTG ACC TTC CCC ATC AGG GT* | This study | N/A |
| *Il1b forward (5′-3′): TGC CAC CTT TTG ACA GTG ATG AG* | This study | N/A |
| *Il1b reverse (5′-3′): TCA TCT TTT GGG GTC CGT CAA C* | This study | N/A |
| *Tgfb1 forward (5′-3′): GCA GTG GCT GAA CCA AGG A* | This study | N/A |
| *Tgfb1 reverse (5′-3′): AGA GCA GTG AGC GCT GAA TC* | This study | N/A |
| *E2f1 forward (5′-3′): GGGGGATAAGCAAAGGGTGG* | This study | N/A |
| *E2f1 reverse (5′-3′): ATCCTGCCCTTGCTTCAGAG* | This study | N/A |
| *E2f2 forward (5′-3′): AGG GCC AAA TTG TGC GAT GT* | This study | N/A |
| *E2f2 reverse (5′-3′): CCT CCA GGT CCA ACT TCC TTT* | This study | N/A |
| *E2f3 forward (5′-3′): GTC TAA AGA CCC CCA GGG GC* | This study | N/A |
| *E2f3 reverse (5′-3′): TGA GGG AGA TTT TGG AGT TTT GG* | This study | N/A |
| *E2f4 forward (5′-3′): GCG TTC TGG ATC TCC CCA AA* | This study | N/A |
| *E2f4 reverse (5′-3′): AGC TCA TGC ACT CTC TCG TG* | This study | N/A |
| *E2f5 forward (5′-3′): TGG GCT TGC TTA CCA CCA AA* | This study | N/A |
| *E2f5 reverse (5′-3′): GGT ATC TGC AGC CGC TTT GA* | This study | N/A |
| *E2f6 forward (5′-3′): GTG GAA AAC CTA CTG CCA TCA AA* | This study | N/A |
| *E2f6 reverse (5′-3′): AGT GAC ACA TCA AAC CGG GG* | This study | N/A |
| *E2f7 forward (5′-3′): TGA AGA TGC AGA GAA CGC ACA AA* | This study | N/A |
| *E2f7 reverse (5′-3′): GCT CGT TCT CA TCG GTG TCT TT* | This study | N/A |
| *Cdkn3 forward (5′-3′): CAG ACG AAG AAC CTG TTG ATG AAG* | This study | N/A |
| *Cdkn3 reverse (5′-3′): AAT TCA CTC GCG ACA GAG GTA G* | This study | N/A |
| *Cdc25c forward (5′-3′): TGC CTG ACG TCT ATA GCC CC* | This study | N/A |
| *Cdc25c reverse (5′-3′): TGC AGG TGG GAT AGG TCC TG* | This study | N/A |
| *Eefa1a forward (5′-3′): TCA TGT CAC GAA CAG CAA AGC* | This study | N/A |
| *Eefa1a reverse (5′-3′): CCT GGC AAG CCC ATG TGT* | This study | N/A |
| *Rpl27 forward (5′-3′): CTG GCC TTG CGC TTC AA* | This study | N/A |
| *Rpl27 reverse (5′-3′): TCA TGC CAC CAA GGT ACT CTG T* | This study | N/A |
| Human qPCR Primers | | |

| Reagent/resource | Reference or source | Identifier or catalog number |
| --- | --- | --- |
| *IL1B forward (5'-3'): GCACGATGCACCTGTACGA* | This study | N/A |
| *IL1B reverse (5'-3'): AGAACACCACTTGTTGCTCCATATC* | This study | N/A |
| *E2F1 forward (5'-3'): CCG CCA TCC AGG AAA AGG TGT GAA* | This study | N/A |
| *E2F1 reverse (5'-3'): AGG TCG ACG ACA CCG TCA GC* | This study | N/A |
| *E2F2 forward (5'-3'): AAC AAC ATC CAG TGG GTA GGC* | This study | N/A |
| *E2F2 reverse (5'-3'): GGT CTG GTG GGG TCT TCA AA* | This study | N/A |
| *CDKN3 forward (5'-3'): AGC CGC CCA GTT CAA TAC AA* | This study | N/A |
| *CDKN3 reverse (5'-3'): ACT CGT GAC AAA GAT AGC CAT GA* | This study | N/A |
| *CDC25C forward (5'-3'): ACCTAGGAGAAGACCAGGCA* | This study | N/A |
| *CDC25C reverse (5'-3'): GGCCACTTCTGCTCACCTTT* | This study | N/A |
| *TFF1 forward (5'-3'): GTT GGG AGC TAG GAT GGT CA* | This study | N/A |
| *TFF1 reverse (5'-3'): AGT GAG TGG CGG ATT TGA AC* | This study | N/A |
| *EEF1a1 forward (5'-3'): GCC GTG TGG CAA TCC AAT* | This study | N/A |
| *EEF1a1 reverse (5'-3'): AGC GCC GGC TAT GCC CCT G* | This study | N/A |
| *RPL27 forward (5'-3'): GTGAAAGTGTATAACTACAATCACC* | This study | N/A |
| *RPL27 reverse (5'-3'): TCAAACTTGACCTTGGCCT* | This study | N/A |
| *USP16 forward (5'-3'): TGGGCTCTGTCGCCGTGGATTG* | This study | N/A |
| *USP16 reverse (5'-3'): TGTCCGTTTCTTTCCCATGTTGGCAC* | This study | N/A |
| **Chemicals, enzymes and other reagents** | | |
| Fulvestrant | Sigma-Aldrich | Cat#I4409 |
| β-Estradiol | Sigma-Aldrich | Cat#E8875 |
| Puromycin | InvivoGen | Cat#ant-pr-1 |
| DMEM medium | Thermo Fisher | Cat#31966047 |
| DMEM no glutamine no phenol red | Gibco | Cat# 31053028 |
| DMEM /F12 with Glutamax-1 | Gibco | Cat#31331028 |
| Trypsin EDTA 0.25% | Corning | Cat#25-053-CI |
| Fetal Bovine Serum | Gibco | Cat#10270106 |
| Fetal Bovine Serum charcoal stripped | Gibco | Cat#12676029 |
| Penicillin-Streptomycin | Gibco | Cat#15140-122 |
| MEM vitamin solution | Gibco | Cat#1120052 |
| Rhodamine B | Sigma-Aldrich | Cat#R6626 |
| Matrigel | BD bioscience | Cat#356237 |
| Protease/Phosphatase Inhibitor Cocktail | Cell Signaling Technologies | Cat#5872S |

| Reagent/resource | Reference or source | Identifier or catalog number |
| --- | --- | --- |
| Tween® 20 | Applichem | Cat#A4974 |
| Anti-Rabbit IgG (H + L), HRP Conjugate | Promega | Cat# W401B |
| DAPI | Sigma-Aldrich | Cat#D9542 |
| Normal goat serum | Abcam | Cat#ab7481 |
| Trizol LS | Thermo Fischer Scientific | Cat#10296028 |
| Iodure de propidium | Sigma-Aldrich | Cat#81845 |
| PureLink™ RNase A | Thermo Fischer Scientific | Cat#12091021 |
| Paraformaldehyde | AppliChem | Cat#A3813 |
| Mowiol | Fluka | Cat#81381 |
| Ammonium thiocyanate | Roth | Cat#4477.3 |
| PBS, Phos-Buff Saline, 1x pH 7.4 | Invitrogen | Cat#10010015 |
| **Commercial kits** | | |
| Pierce™ BCA Protein Assay Kit | Life Technologies | Cat#23225 |
| Advansta ECL WesternBright Quantum | Advansta | Cat#K-12042-D20 |
| Amersham™ ECL Select™ Western Blotting Detection Reagent | Amersham | Cat#RPN2235 |
| QIAGEN DNA micro kit | QIAGEN | Cat#56204 |
| RNeasy mini kit | QIAGEN | Cat#74104 |
| GoScript™ Reverse Transcriptase | Promega | Cat# A5001 |
| GoTaq® qPCR Master Mix | Promega | Cat#A6001 |
| **Software** | | |
| GraphPad Prism v7 | https://www.graphpad.com | |
| FlowJo™ v10.7.1 | BD bioscience | RRID:SCR_008520 |
| ImageJ | https://imagej.net/ij/ | |
| AxioVision | Carl Zeiss | |
| Qupath | https://qupath.github.io/ | RRID:SCR_018257 |
| R version 4.1.0 | | |
| Bcle2fastq2 Conversion software version2.20 | Illumina | |
| **Other** | | |
| BD Accuri C6 Plus | BD bioscience | RRID:SCR_014422 |
| Fusion FX | Vilber | |
| AxioImager M1 microscope | Carl Zeiss | |
| Illumina Novaseq 6000 | Illumina | |
| Fragment Analyzer | Agilent Technologies | |

## Animal experiments

All experiments involving animals were approved by the Veterinary Office of the Canton Vaud (Switzerland) in accordance with the

Federal Swiss Veterinary Office Guidelines and conform to the Commission Directive 2010/63/EU.

SKH-1 hairless mice (Crl: SKH1-Hr$^{hr}$, Charles River) were crossed with Sv129/C57BL6/J male and female mice to obtain B6.129-SKH1-Hr$^{hr}$ mice. This model is well established in the laboratory as part of our research into the role of PPARβ in the development of skin cancers and is therefore used for all projects (Montagner et al, 2014). Mice were raised, housed, and exposed to experimental conditions as described below in the conventional animal facility of the Center for Integrative Genomics, at the University of Lausanne. They were kept on IVC cages in a standard colony (2–5 animals per cage), in a light-controlled environment (12/12-h light/dark cycle, artificial light with daylight spectrum at an average intensity of 100 lux), with a hygrometry between 45% and 65% and a temperature of 22 °C ( +/− 2 °C). They were housed on Aspen bedding (Safe select) and fed ad libitum with Sp-150 irradiated pellets (Safe) and filtered water.

For acute and chronic UV exposure, mice were UV irradiated on their backs with a GL40E 40 W tube (SNEE), which emits most of its energy within the UVB range (90%; emission spectrum 280–370; 10% UVA). Doses of UVB (312 nm) and UVA (370 nm) were monitored using an appropriate radiometer. The exposures were carried out in a (home-made) box where the mice were separated individually to avoid bias due to grouping, and the experiment took place in the animal facility.

### Acute UV exposure

Females and males B6.129-SKH1-Hr$^{hr}$ aged 10–14 weeks old were placed in individual room and irradiated on their backs with a UV lamp. UV radiation was monitored using a radiometer until a dose of 120 mJ/cm$^2$ was reached in approximately 3 min. This is a sub-erythematous dose. Control mice were sham manipulated but not exposed to UV. Mice were euthanized 1, 24, 72 or 120 h after UV exposure by lethal intraperitoneal injection of pentobarbital (300 mg/kg) followed by cervical dislocation. Female mice were treated with fulvestrant (Sigma-Aldrich I4409) (150 mg/kg, i.p.) or its solvent (95% corn oil, 5% DMSO) 24 h prior to acute UV exposure.

### Chronic UV exposure

Females and males B6.129-SKH1-Hr$^{hr}$ aged 10–14 weeks old were exposed on their backs three times per week with a dose of 70 mJ/cm$^2$ delivered approximately in 1 min and 45 s for 21 weeks. Each mouse received a total number of 74 UV doses and were then euthanized by lethal intraperitoneal injection of pentobarbital (300 mg/kg) followed by cervical dislocation. Mice were euthanized before the tumor reached 1 cm$^3$. Tumors' size, appearance and number were monitored twice a week. Control mice were sham manipulated. Tumors were collected as follows: well-defined lesions were excised entirely, whereas for poorly defined lesions, which were observed only in males, representative punch biopsies of the affected areas were sampled. Tumors, as well as proximal (1 cm away from tumor) and distal (2 cm away from tumor) skin from tumors were collected.

### Multiple UV exposure for mutations detection

Females and males B6.129-SKH1-Hr$^h$ aged 10–14 weeks old were exposed on their backs three times per week with a dose of 70 mJ/cm$^2$ delivered approximately in 1 min and 45 s for 2 weeks, 8 weeks or 12 weeks and aged for 18 weeks before being euthanized.

## Tumor grading and epidermal atypia grading

Mouse skin tumors derived from chronic or acute UV exposure were fixed in 4% paraformaldehyde for 24 h and then embedded in paraffin. Tissue sections (4 μm) were stained with hematoxylin and eosin. Histological analysis of actinic keratosis and tumor classification were performed blindly by a dermatopathologist. SCCs were classified according to the Broders' classification based on the degree of SCC keratinization and of keratinocyte differentiation (Broders, 1921). The classification includes SCC Grade I ( > 75% well-differentiated keratinocytes), SCC Grade II (25%–75% well-differentiated keratinocytes), SCC Grade III ( < 25% well-differentiated keratinocytes). Actinic keratosis was histologically defined and classified as grade I, II or III based on the degree of cytological atypia of epidermal keratinocytes and involvement of adnexal structures according to the criteria proposed by Rowert-Huber et al (Röwert-Huber et al, 2007).

Epidermal atypia was graded based on cytological features, including keratinocyte nuclei larger than the average size within each specific epidermal layer, abnormal nuclear shape and texture, nucleolar prominence in more than 10% of keratinocytes, as well as architectural changes such as acanthosis, hyperkeratosis, altered epidermal thickness, and disorganization of cell alignment (Katayama et al, 2022; Pambuccian, 2015; Pellacani et al, 2015; Sanfrancesco et al, 2013; Smoller, 2006; Tucker et al, 2002). The degree of atypia was based on the number of epidermal layers affected: Grade 0, no atypia; Grade 1, discrete atypia; Grade 2, intermediate atypia; and Grade 3, severe atypia. Histological evaluation was performed blindly by a dermatopathologist.

## Whole-mount preparation of mice dorsal epidermis

The dorsal skin was cut into 24 pieces of 4 × 5 mm per mice and incubated in PBS containing 5 mM EDTA at 37 °C for 2.5 h. Samples were transferred into PBS and the epidermis was carefully scraped off using curved scalpel while holding one corner of the skin with forceps. The epidermal whole mounts were fixed in 4% paraformaldehyde in PBS for 30 min and then stored in PBS at 4 °C.

## Ultra-deep targeted sequencing epidermal whole mounts

Wholemount epidermis was micro-dissected into a gridded array of 2 mm$^2$ samples. DNA was extracted using QIAGEN DNA micro kit (QIAGEN) by digesting overnight and following manufacturer's instructions. DNA was eluted using pre-warmed AE buffer and was passed through the column a total of three times. To analyze germline mutations, DNA was extracted from liver tissue for each animal.

In total, 200 ng of genomic DNA was fragmented to give an average size distribution of ~150 bp (LE220, Covaris Inc). Samples were purified, libraries prepared (NEBNext Ultra II DNA Library prep Kit, New England Biolabs), and index tags applied (Sanger 168 tag set). Index tagged samples were amplified (6 cycles of PCR,

KAPA HiFi kit, KAPA Biosystems), quantified (Accuclear dsDNA Quantitation Solution, Biotium), then pooled in equimolar ratios. 500 ng of pooled material was taken forward for hybridization, capture and enrichment (SureSelect Target enrichment system, Agilent technologies). A target bait panel of 74 genes was used. Genes were selected to cover those frequently mutated in squamous cell carcinoma.

The gene sequenced were:

*Aff3, Ajuba, Arid1a, Arid2, Arid5b, Atm, Atp2a2, Bcl11b, Braf, Cacna1d, Card11, Casp8, Ccnd1, Cdkn2a, Cobll1, Crebbp, Ctcf,, Ctnnb1, Dclk1, Dclre1a, Dnmt3a, Ddr2, Egfr, Eif2d, Ep300, Erbb2, Erbb3, Erbb4, Ezh2, Fat1, Fat2, Fat3, Fat4, Fbxw7, Fbxo21, Fgfr3, Flt3, Grin2a, Hras, Kdm6a, Kdr, Kit, Kmt2c, Kmt2d, Kras, Lrp1b, Mtor, Nf1, Nf2, Notch1, Notch2, Notch3, Notch4, Nras, Pik3ca, Ptch1, Pten, Rb1, Ros1, Smad4, Smarca4, Smo, Sox2, Stat5b, Tert, Tet2, Tgfbr1, Tgfbr2, Trp53, Tsc1, Vhl, Zfp750, Nrf2, Keap1.*

Post-clean up samples were normalized to approximately 6 nM and submitted to cluster formation for sequencing on Novaseq 6000 (Illumina) to generate 100 bp paired end reads.

BAM files were mapped to the GRCh37d5 reference genome using BWA-mem (version 0.7.17) (Li, 2013) Duplicate reads were marked using SAMtools (v1.11) (Danecek et al, 2021). Depth of coverage was also calculated using SAMTools to exclude reads which were: unmapped, not in the primary alignment, failing platform/vendor quality checks or were PCR/Optical duplicates. BEDTools (version 2.23.0) coverage program was then used to calculate the depth of coverage per base across samples (Quinlan and Hall, 2010).

Mutation variant calling was made using the deepSNV R package (also commonly referred to as ShearwaterML), version 1.21.3, available at https://github.com/gerstung-lab/deepSNV, used in conjunction with R version 3.3.0 (2016-05-03) (Martincorena et al, 2018a). Variants were annotated using VAGrENT (Menzies et al, 2015).

Mutations called by ShearwaterML were filtered using the following criteria:

- Positions of called SNVs must have a coverage of at least 100 reads.
- Germline variants called from the same individual were removed from the list of called variants.
- Adjustment for FDR and mutations demanded support from at least one read from both strands for the mutations identified.
- Pairs of SNVs on adjacent nucleotides within the same sample are merged into a dinucleotide variant if at least 90% of the mapped DNA reads containing at least one of the SNV pair, contained both SNVs.
- Identical mutations found in multiple contiguous tissue biopsies are merged and considered as a single clone to prevent duplicate clone counting.

ShearwaterML was run with a normal panel of approximately 45k reads.

COSMIC signature profiles (v3.2) (https://cancer.sanger.ac.uk/signatures) were used for Single Base Substitutions (SBS), and Double base Substitutions (DBS) signature classification using the SigProfiler packages as follows: MatrixGenerator (v1.2.12), Extractor (v1.1.12), Assignment (v0.0.13), Plotting (1.2.2). Frequency of mutations within each trinucleotide context was calculated using SigProfiler within the SBS288 context.

Comparison of differential selection was calculated as described (Martincorena et al, 2015a).

## Human skin biopsies

Patient-derived formalin-fixed paraffin-embedded blocks with actinic keratosis and cutaneous SCC were obtained from the VITA certified Dermatology Biobank (CHUV_2103_12) of the Lausanne University Hospital (CHUV), Switzerland. Informed consent was obtained from all enrolled patients.

## Ex vivo human skin culture

The tissue sources were registered within the department biobank validated by the Operational Committee of Biobanks and Official Registers of the Lausanne University Hospital CHUV (Biobank N#BB_029_DAL). Skin explants were obtained from women adult healthy patients undergoing abdominoplasty (phototype 1 or 2, Fitzpatrick scale). The study was approved by the local ethics committee. Written informed consent was obtained from all subjects. Immediately after the post-excision, the human skin was washed in sterile phosphate buffered saline (PBS) and subcutaneous fat was removed with a scalpel. Next, the skin was cut into $1 \times 1$ cm square. Skin pieces were placed into 6 cm dishes containing DMEM semi-solid medium (DMEM /F12 with Glutamax-1 (Gibco 31331-028) containing 10% of FBS, 1% of P/S and 1% of multivitamins and 2.5% of agarose). The epidermis was exposed to air whereas the dermis was embedded in the semi-solid medium. Pieces of skin were incubated at 37 °C in a humidified atmosphere containing 5% $CO_2$, and the medium was changed every 3 days. To mimic human UV exposure, human skin explants were exposed to 10% of UVA (365 nm) and 90% of UVB (312 nm) at three different doses: 240 mJ/cm² (UVA: 24 mJ/cm²; UVB: 216 mJ/cm²), 480 mJcm² (UVA: 48 mJ/cm²; UVB: 432 mJ/cm²) or 720 mJ/cm² (UVA: 77 mJ/cm²; UVB: 648 mJ/cm²). Immediately after UV exposure skin was placed in fresh medium and incubator. Control skin not exposed to UV was sham manipulated.

## Cell lines, cell cultures and treatment

The human cervical squamous cell carcinoma line A431, derived from a cutaneous epidermoid carcinoma in an 85-year-old female, was purchased from the European Collection of Authenticated Cell Cultures (ECACC; 85090402). The human skin squamous cell carcinoma line SCC13, derived from cutaneous SCC in a 56-year-old male (RRID:CVCL_4029), was kindly provided by Prof. Sabine Werner (Institute of Molecular Health Sciences, Zurich, Switzerland). All cells were cultured in DMEM (Dulbecco's Modified Eagle's Medium; Gibco) supplemented with 4500 mg/L glucose,10% fetal bovine serum and 1% penicillin-streptomycin at 37 °C in a humidified atmosphere containing 5% $CO_2$. The immortalized human keratinocytes cell line HaCaT (Cell Line Service; Germany) was cultured under the same conditions. For β-estradiol treatment, cells were maintained in red-phenol-free DMEM (Dulbecco's Modified Eagle's Medium; Gibco) supplemented with 4500 mg/L glucose, 10% of charcoal-stripped fetal bovine serum (Gibco) and 1% penicillin-streptomycin. Cells were stimulated with β-Estradiol (Sigma-Aldrich; 0.1 μM, 1 μM, and 10 μM) or vehicle (0.1% ethanol) and incubated for 24 h.

## Lentiviral knockdown of CDKN3

CDKN3 was silenced in A431 and SCC13 cells by lentiviral transduction (VectorBuilder). Vectors used were pLV[shRNA]-EGFP/Puro-U6>Scramble (shNC, negative control; VB010000-9526zpu; target sequence: CCTAAGGTTAAGTCGCCCTCG) and pLV[shRNA]-Puro-U6 > hCDKN3 (shCDKN3; VB250210-1177vhe; target sequence: GTGCAGATATTCCTAAAGTTT). A431 and SCC13 cell lines were incubated for 16 h in culture medium containing lentivirus at a multiplicity of infection (MOI) of 5 in the presence of 8 μg/mL polybrene. Puromycin (3 μg/mL) was applied to select stably transduced cells, and knockdown efficiency was assessed by qPCR and western blotting.

## Colony formation assay

Negative Control (shNC) and shCDKN3 cells (500 cells/well) were seeded in 6-well plates and cultured for 2 weeks until the colonies were formed. Formed colonies were washed twice with 1× PBS, fixed by 4% paraformaldehyde for 15 min, and stained with 1% Rhodamine B. The colonies were photographed, and they were counted using ImageJ software.

## Cell cycle analysis

Cells were fixed with 70% cold ethanol at 4 °C for a period of 1 h. Following fixation, DNA was stained by incubating saponin-based permeabilization and wash buffer containing 0.2 mg/mL RNase and 5 μg/mL Propidium Iodide. Cells were then acquired using Accuri C6 flow cytometer (BD Accuri C6 Plus, RRID:SCR_014422) and analyzed using FlowJo™ v10.7.1 Software (FlowJo, RRID:SCR_008520). Single cells are gated based on their FSCH and FSCA profiles.

## Nude mice xenograft experiment

Nude mice (Rj: NMRI Foxn1 nu/nu, RRID:IMSR_RJ:NMRI-NUDE) were purchased from Janvier Labs (France). A total of $3 \times 10^6$ A431 cells in 50 μl PBS, and $5 \times 10^6$ SCC13 cells in 200 μl PBS:Matrigel (1:1), stably expression shNC or shCDKN3 were subcutaneously injected into the right flank of 10-week-olds female mice Mouse behavior and tumor growth were monitored twice weekly. The maximum tumor volume authorized by the authorities was fixed at 1 cm³, as estimated once per week in live animals using the following formula: $(4/3) \times 3{,}14159 \times (Length/2) \times (Width/2)^2$. Mice were euthanized when tumors measured on the live animal reached this size, and tumors were recovered for further analysis.

## Western blotting

Cells were lyzed in RIPA buffer (containing 50 mM Tris [pH 7.4], 150 mM NaCl, 0.1% SDS, 1% NP-40, 0.5% sodium deoxycholate), supplemented with Protease/Phosphatase Inhibitor Cocktail 1× (5872S, Cell Signaling). Protein amount was quantified using Pierce™ BCA Protein Assay Kit (23225, Life Technologies). After quantification, proteins (15 μg per well) were separated by SDS-PAGE and subjected to immunoblotting. Primary antibodies: CDKN3 (Abcam ab175393), GAPDH (Cell Signaling Technology Cat# 2118) were incubated overnight in 1×Tris-buffered saline plus 0.1% Tween-20 and 5% nonfatty milk at a dilution recommended

by the manufacturer, followed by peroxidase conjugated goat anti-rabbit secondary antibodies (Promega W401B). Immunoreactivity was detected using Advansta ECL WesternBright Quantum (ref K-12042-D20) or Amersham™ ECL Select™ Western Blotting Detection Reagent (ref RPN2235). Images were acquired using a Fusion FX (Vilber).

## Immunofluorescence staining

Tumors and skin were fixed in 4% paraformaldehyde solution for 24 h, then PBS washed, and paraffin-embedded. Staining was performed on 4 μm skin sections.

Antigen retrieval using Citrate buffer 0.01 M pH6 was performed. Sections were blocked in NGS 5% for 45–90 min, incubated with the following antibodies: anti-CPD (Cosmo Bio, NMDND004, 1/1000), anti-γH2AX (EMD Millipore, 05-636-25UG, 1/500), anti-Ki67 (Invitrogen, 41-5698-80, 1/60), anti-Keratine10 (Progen, GP-K10, 1/100), anti-Keratine14 (Covance, PRB155P, 1/500) anti-cleaved caspase-3 (Cell Signaling Technology, 9664, 1/100), anti-CDKN3 (Abcam, ab175393, 1/500) in NGS 5% O/N at 4 °C; washed in PBS; incubated with a fluorescent antibody (Molecular Probes, 1/500 in NSG 5%) for 30 min at room temperature: goat anti-mouse A568 (for CPD and γH2AX), goat anti-rat A568 (for KI67), Goat anti-guinea pig A488 (for K10), goat anti-rabbit A568 (for K14 and CDKN3), goat anti-rabbit A568 (for cleaved caspase-3), washed in PBS, counterstained with DAPI, and embedded in MOWIOL. Pictures were taken with a motorized AxioImager M1 microscope and the AxioVision software (Carl Zeiss), using a magnification of ×10, ×20 or ×40. After defining epidermis region, automatic cell detection of staining was performed using Qupath software (Bankhead et al, 2017) in a blinded way.

## Epidermis-dermis separation for RNA extraction

Epidermis and dermis were prepared from whole dorsal skin of mice according to the protocol developed by Clemmensen et al (Clemmensen et al, 2009). After dorsal skin harvesting, the skin was cut into thin strips (2 mm wide) and immediately incubated for 20 min at RT in ammonium thiocyanate (3.8% in 1×PBS). Before mechanical separation with forceps and scalpel, skin was washed once with 4 °C PBS.

## RNA isolation and RT-qPCR analysis

Total RNA was isolated from human and mouse epidermis using TRIzol LS (Life Technologies, Thermo Fisher) and purified with the RNeasy kit (Qiagen) according to the manufacturer's instructions. RNA quality was verified by microfluidic (Agilent 2100 Bioanalyzer) and concentration determined with a NanoDrop spectrophotometer (Wilmington). Total RNA (500 ng–1 μg) was reverse transcribed using an RT Go script Kit (Promega) according to the manufacturer's instructions. cDNA quantifications were performed with GoTaq qPCR Master Mix (Promega). No template control and no reverse transcriptase enzyme sample were used as negative controls. Gene expressions were normalized to the meaning of two housekeeping gene expressions (*Eef1a1* and *Rpl27* genes or *EEF1a1* and *USP16* genes). Primer sequences are accessible in the Reagents and tools table.

## RNA-seq gene expression studies

RNA-seq experiment was conducted at the Lausanne Genomic Technologies Facility (Lausanne, Switzerland) according to an in-house pipeline. Briefly, mouse RNA samples were extracted from four epidermal samples of each sex, in control epidermis and 24 h after single UV exposure (120 mJ/cm²). RNA quality was assessed on a Fragment Analyzer (Agilent Technologies) and all RNAs had a RQN from 8 to 10. RNA-seq libraries were prepared from 500 ng of total RNA with the Illumina TruSeq Stranded mRNA reagents (Illumina) using a unique dual indexing strategy and following the official protocol automated on a Sciclone liquid handling robot (PerkinElmer). Libraries were quantified by a fluorimetric method (QubIT, Life Technologies) and their quality assessed on a Fragment Analyzer (Agilent Technologies). Sequencing was performed on an Illumina NovaSeq 6000 for 100 cycles single read. Sequencing data were demultiplexed using the bcl2fastq2 Conversion Software (version 2.20, Illumina).

### RNA-seq data processing

Data cleaning. Purity-filtered reads were adapted and quality trimmed with Cutadapt (v. 1.8, Martin 2011). Reads matching ribosomal RNA sequences were removed with fastq_screen (v. 0.11.1). Remaining reads were further filtered for low complexity with reaper (v. 15-065).

Reads were aligned against the GRCh38.102 genome using STAR (v. 2.5.3a). The number of read counts per gene locus was summarized with htseq-count (v. 0.9.1) using GRCh38.102 gene annotation. Quality of the RNA-seq data alignment was assessed using RSeQC (v. 2.3.7).

Statistical analysis was performed in R (R version 4.1.0). Genes with low counts were filtered out according to the following rule: at least 1 sample had to have at least 1 cpm (1 count per million) reads to keep the gene in the dataset. Library sizes were then scaled using TMM normalization. Subsequently, the normalized counts were transformed to cpm values and a log2 transformation was applied by means of the function cpm with the parameter setting prior.counts = 1 (EdgeR v 3.34.1).

### Differential expression

Differential expressions were computed with the R Bioconductor package limma by fitting data to a linear model. The approach limma-trend was used. Results from contrasts of interest and interactions were extracted. Moderated $F$ tests were performed for groups of contrasts and groups of interactions. The resulting $P$ values were adjusted for multiple testing by the Benjamini-Hochberg method, which controls the false discovery rate (FDR). This adjustment was performed for each $F$ test. A post-hoc test was performed, using the function decide Tests with parameter method=nestedF.

Functional annotation clustering of GO Biological Process was performed using DAVID Bioinformatics resource system (Sherman et al, 2022).

## Publicly available datasets

Public datasets were found on the Gene Expression Omnibus data repository; two independent gene expression datasets were used: GSE66412 and GSE84293. GSE66412 access available in August 2015,

compared to the RNAseq whole transcriptome of Normal human epithelial keratinocytes (NHEKs) and Squamous cell carcinoma cell lines (SCC). GSE84293 access available on February 2017 through next generation sequencing compared transcriptome of normal skin (NS, peritumoral), actinic keratosis (AK) and cutaneous SCC. Values for CDKN3 and CDC25C gene expression from RNA sequencing data were calculated using GREIN (Mahi et al, 2019).

## Kaplan–Meier plotter database analysis

The correlation between CDKN3 expression and CDC25C expression and survival in head and neck squamous cells carcinoma patients ($n = 500$) was analyzed using Kaplan–Meier plotter (http://kmplot.com/analysis), the hazard ratio (HR) with 95% confidence intervals and log-rank $P$ value was also determined (Győrffy, 2024).

## Statistical analysis

To compare two independent variables in more than two groups, we used two-way ANOVA after verifying normality and homoscedasticity of the data. Post-hoc tests were selected based on the experimental design: Tukey's or Dunnett's for balanced pairwise comparisons, Holm–Šidák for multiple related comparisons with unbalanced samples, and uncorrected Fisher's LSD for a small number of samples, interpreted cautiously due to the lack of correction for multiple comparisons. Statistics on the survival curves were performed using a log-rank $t$ test. All statistical analyses were performed using Prism GraphPad (v7).

# Data availability

The sequencing dataset generated during this study is available at the European Nucleotide Archive (ENA) with the following dataset accession number: ERP159842 (https://www.ebi.ac.uk/ena/browser/view/PRJEB75253). The data generated during this study have been deposited in the GEO database under accession code GSE266221.

The source data of this paper are collected in the following database record: biostudies:S-SCDT-10_1038-S44319-026-00743-2.

# Peer review information

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

## Acknowledgements

We thank the Animal facility (Center of Integrative Genomics, University of Lausanne, Switzerland) for expert technical assistance and the Genomic Technologies Facility (University of Lausanne, Switzerland) for performing RNA-seq analyses. We thank Fabienne Lammers (Center for Integrative Genomics, University of Lausanne, Switzerland) for mouse genotyping and Nathalie Hirt (Lausanne University Hospital, University of Lausanne, Switzerland) for valuable support in getting human skin samples from the Lausanne University Hospital biobank. We are grateful to Prof. Hon. Béatrice Desvergne (University of Lausanne, Switzerland) and Hervé Guillou, PhD, Laurence Payrastre, PhD and Sandrine Ellero-Simatos, PhD (Toxalim, INRAE, ENVT, INP- PURPAN, UMR 1331, UPS, University of Toulouse, France) and Alexandra Montagner, PhD (I2MC, UMR1297, INSERM/UPS), Dorian Ziegler, PhD and Sarah Geller, PhD (Center of Integrative Genomics, University of Lausanne, Switzerland) for constructive scientific discussion. This work was supported by the Swiss National Science Foundation (IZLIZ3_200253/1 to EG), the Foundation Recherche Cancer ISREC (CCP 10-3224-9 to EG), and the Etat de Vaud (University of Lausanne; SKINTEGRITY.CH collaborative research to EG and LM).

## Author contributions

**Céline Lukowicz**: Conceptualization; Formal analysis; Supervision; Validation; Investigation; Visualization; Methodology; Writing—original draft; Writing—review and editing. **Carine Winkler**: Formal analysis; Validation; Investigation; Methodology; Writing—review and editing. **Catherine Roger**: Investigation; Visualization; Methodology. **Joanna C Fowler**: Data curation; Formal analysis; Investigation; Methodology; Writing—review and editing. **Yi-Chien Tsai**: Formal analysis; Visualization; Methodology. **Joachim Meuli**: Resources; Methodology; Writing—review and editing. **Stéphanie Claudinot**: Investigation; Methodology; Writing—review and editing. **Yun-Tsan Chang**: Methodology. **Christoph Iselin**: Resources; Methodology. **Philip H Jones**: Resources; Formal analysis; Writing—review and editing. **Emmanuella Guenova**: Resources; Formal analysis; Visualization; Writing—review and editing. **Paris Jafari**: Conceptualization; Formal analysis; Methodology; Writing—review and editing. **Liliane Michalik**: Conceptualization; Resources; Supervision; Validation; Writing—original draft; Project administration; Writing—review and editing.

Source data underlying figure panels in this paper may have individual authorship assigned. Where available, figure panel/source data authorship is listed in the following database record: biostudies:S-SCDT-10_1038-S44319-026-00743-2.

## Disclosure and competing interests statement

The authors declare no competing interests.

# Expanded View Figures

**Figure EV1. Characterization of dorsal lesions following chronic UV exposure in male and in female mice.**

(**A**) Mean number of actinic keratosis (AK) and squamous cell carcinoma (SCC) lesions collected per mouse. $n = 6$ for each male and female mice group, mean ± SEM, Unpaired $t$ test. (**B, C**) Left: Ki67 (red) immunofluorescence staining in AK (**B**) or SCC (**C**) lesions from male and female mice chronically exposed to UV. DAPI was used as counterstaining (blue). Scale bars: 50 μm. Right: Quantification of the percentage of Ki67-positive keratinocytes. $n$(fields) = 3 per mouse, mean ± SEM, Unpaired $t$ test. (**D**) Tumor depth from indicated SCC stages coming from male (white) or female (grey) mice chronically exposed to UV. Each dot represents a tumor, $n = 4$–16 per group. Mean ± SEM, two-way ANOVA with Holm–Šidák post hoc test.

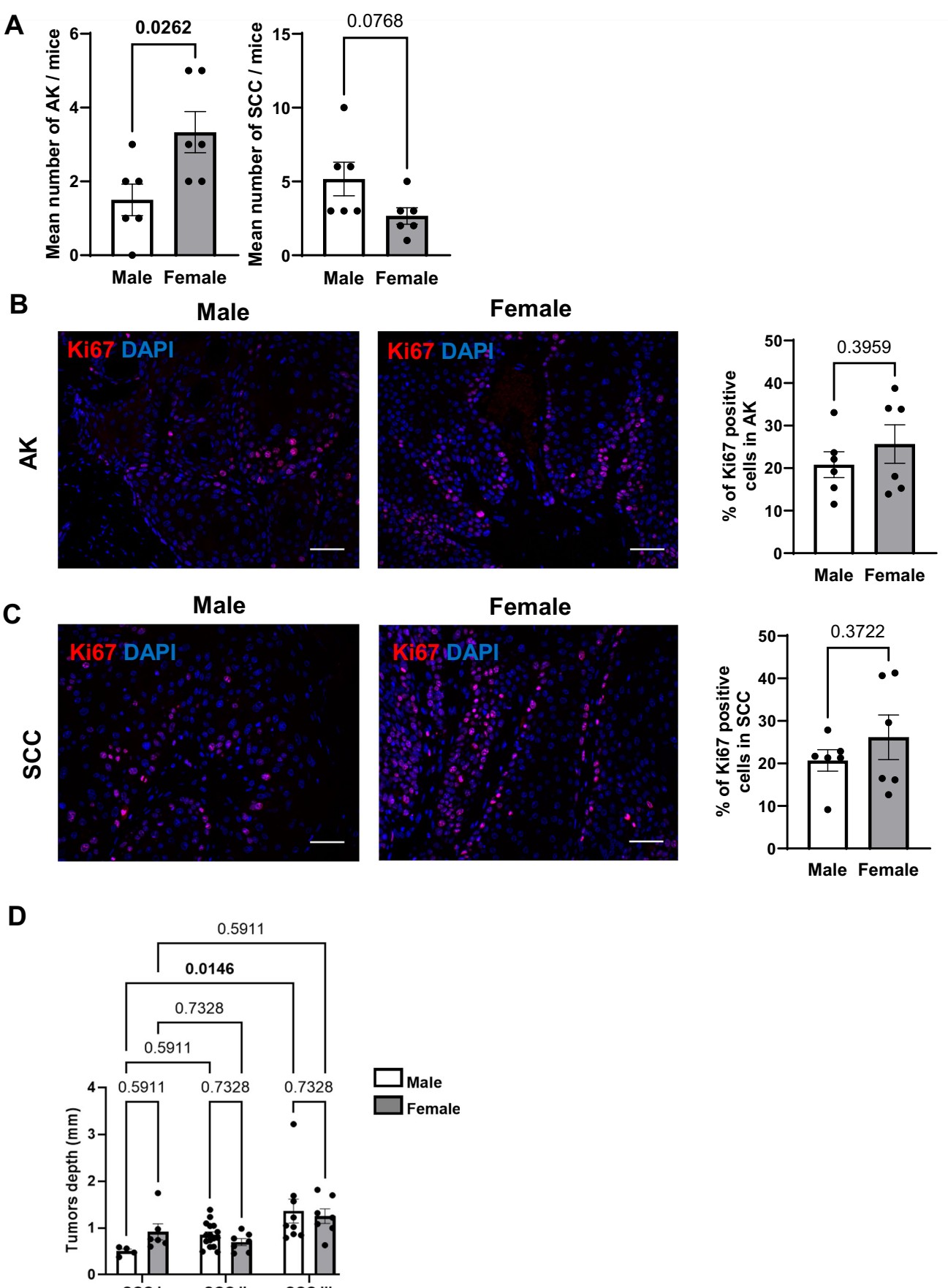

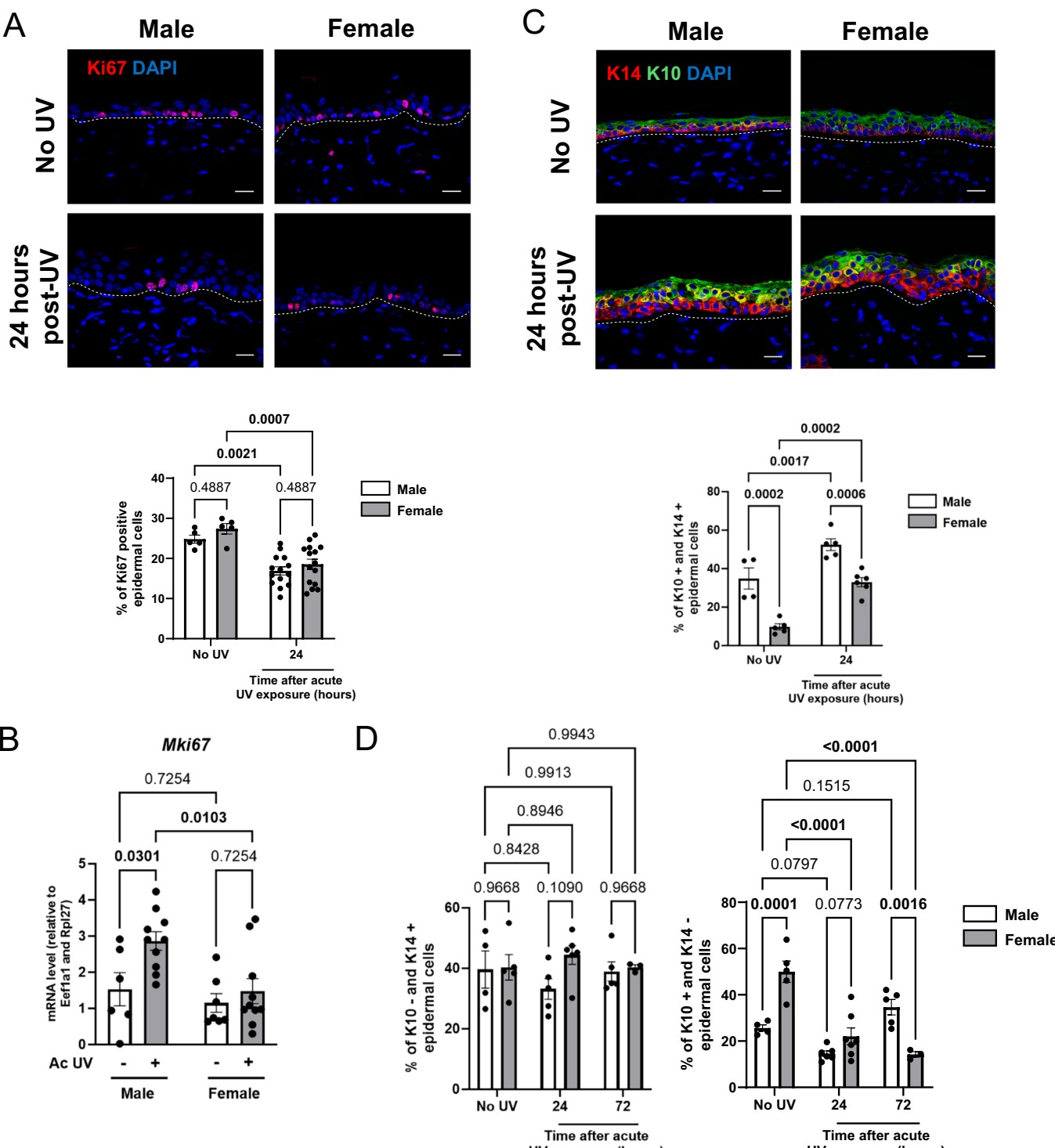

**Figure EV2.   Epidermal proliferation and differentiation following acute UV exposure in male and female mice.**

(A) Top: Ki67 (red) immunofluorescence staining in male and female dorsal skin collected 24 h after a single dose of acute UV exposure (120 mJ/cm²), compared to control skin (No UV). DAPI was used as counterstaining (blue). The dotted line separates the epidermis from the dermis. Scale bars: 20 μm. Bottom: Percentage of Ki67-positive keratinocytes. $n$(fields) = 4 per mouse, mean ± SEM, two-way ANOVA with Holm–Šidák post hoc test. (B) RT-qPCR analysis of Mki67 mRNA expression levels in epidermal samples from male and female mice collected 24 h after acute UV exposure. $n$ = 5–10 mice, mean ± SEM, two-way ANOVA with Holm–Šidák post hoc test. (C) Top: Keratin 14 (K14; red) and Keratin 10 (K10: green) immunofluorescences staining in male and female dorsal skin collected 24 h after a single dose of acute UV exposure (120 mJ/cm²), compared to control skin (No UV). DAPI was used as counterstaining (blue). The dotted line separates the epidermis from the dermis. Scale bars: 20  μm. Bottom: Percentage of K10-positive and K14-positive epidermal cells. $n$(fields) = 4 per mouse, mean ± SEM, two-way ANOVA with Holm–Šidák post hoc test. (D) Percentage of K10 negative and K14-positive (left), and K10-positive and K14-negative (right) epidermal cells. $n$(fields) = 4 per mouse, $n$ = 3– 6 mice, mean ± SEM, two-way ANOVA with Holm–Šidák post hoc test. Female No UV versus Female 24 h post-UV: *P* value < 0.0001 = 0.000004734, Female No UV versus Female 72 h post-UV: *P* value < 0.0001 = 0.000003746.

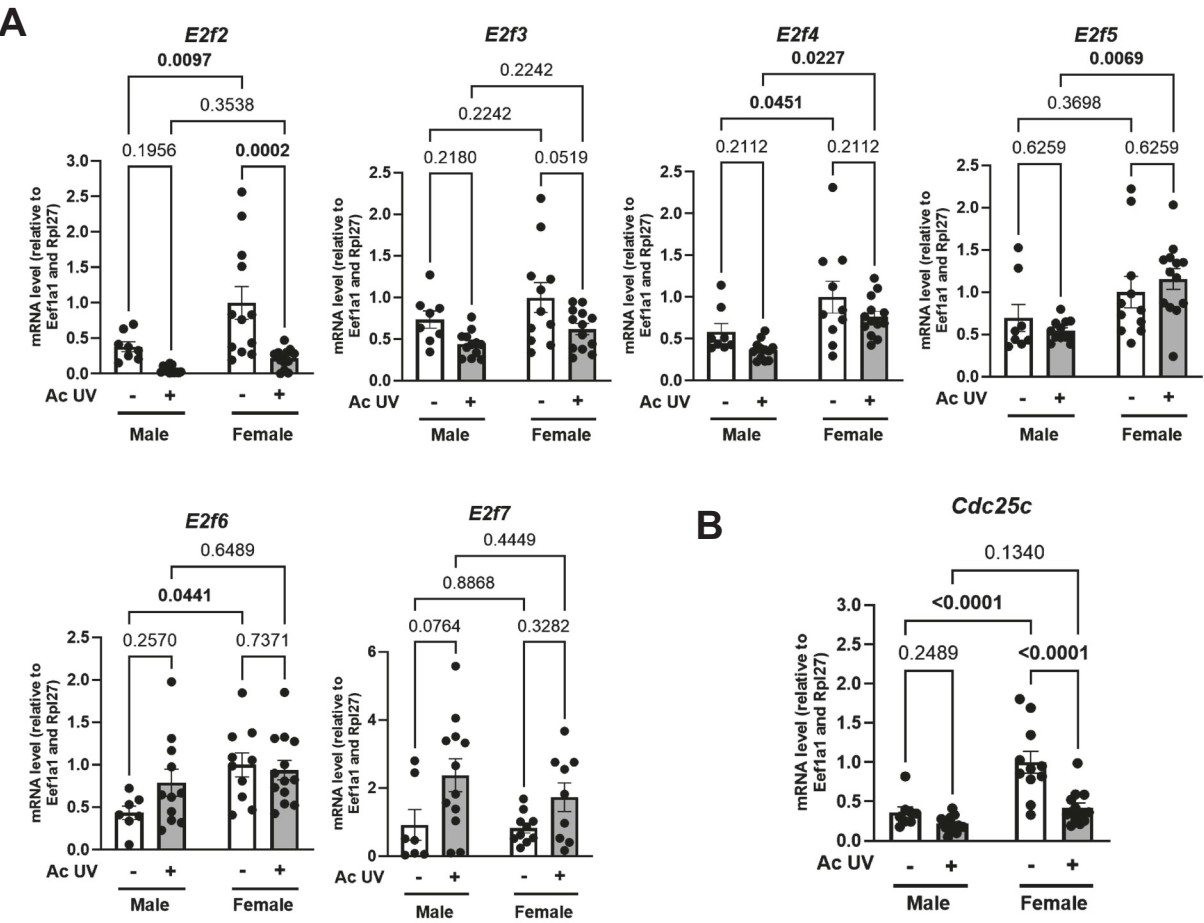

**Figure EV3. Relative gene expression in male and female mice epidermis following acute UV exposure.**

(A) Quantification of the relative *E2f2, E2f3, E2f4, E2f5, E2f6* and *E2f7* transcripts by RT-qPCR in male and female mice epidermal samples collected 24 h after a single dose of acute UV exposure (120 mJ/cm²), compared to control skin (No UV). $n = 8$–14 mice per sex, mean ± SEM, two-way ANOVA with Holm–Šidák post hoc test. (B) Quantification of the relative *Cdc25c* transcript level by RT-qPCR in male and female mice epidermal samples collected 24 h after a single dose of acute UV exposure (120 mJ/cm²), compared to control skin (No UV). $n = 8$–14 mice per sex, mean ± SEM, two-way ANOVA with Holm–Šidák post hoc test. Male No UV versus Female No UV $P$ value $< 0.0001 = 0.000051655$, Female No UV versus Female UV $P$ value $< 0.0001 = 0.000040308$.

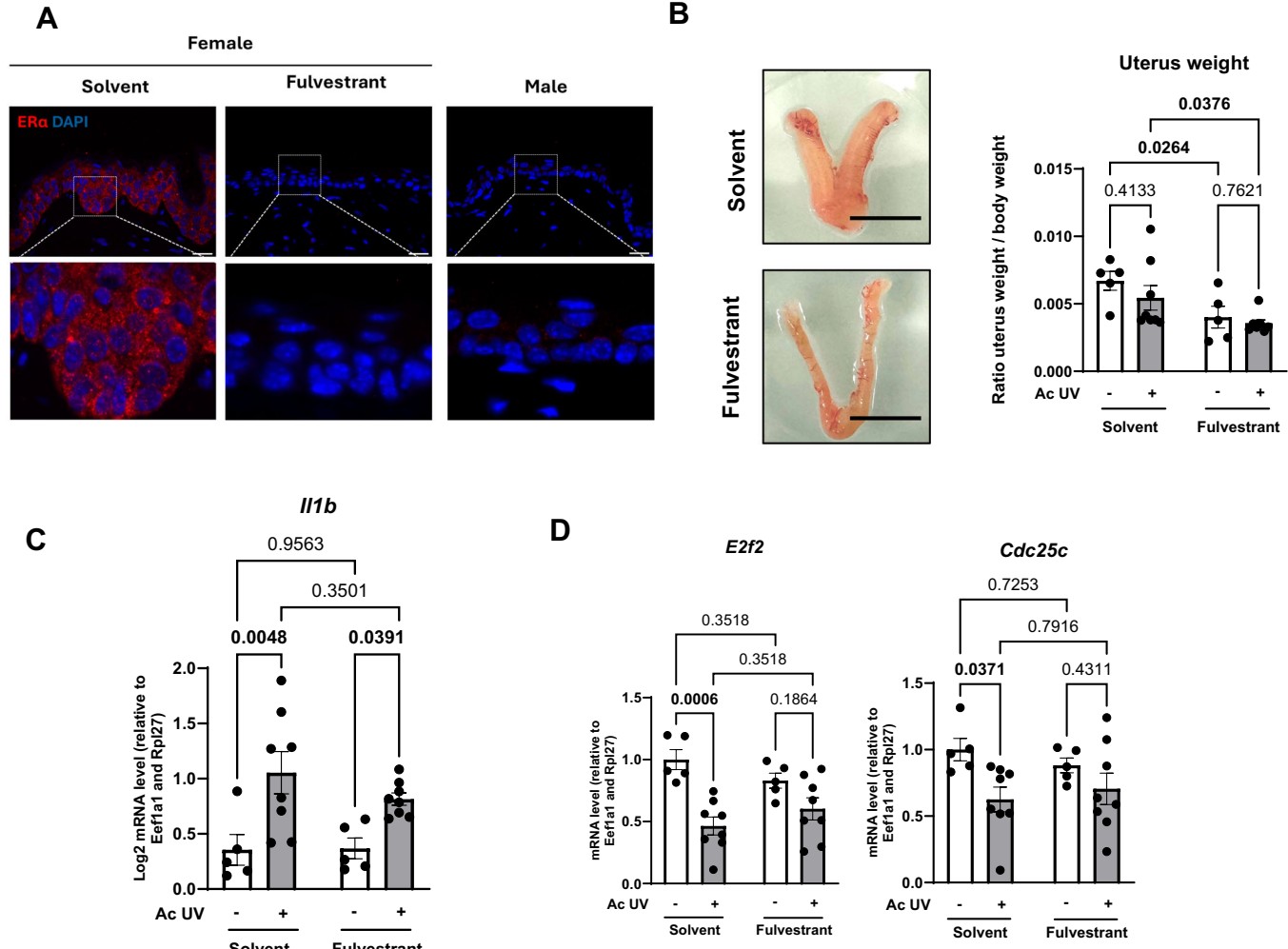

**Figure EV4. Effect of fulvestrant treatment on female mice exposed to acute UV.**

(A) Estrogen Receptors α (red) immunofluorescence staining in female dorsal skin mice following treatment with either solvent or fulvestrant (150 mg/kg) for 48 h. Dorsal skin of male mice was used as negative control. DAPI was used as counterstaining (blue). Scale bars: 20 μm. (B) Left: Representative image of the uterus following treatment with either solvent or fulvestrant. Scale bars: 1 cm. Right: Relative uterus weight after treatment with fulvestrant (150 mg/kg) for 48 h, or with solvent (control). $n = 5$–8 female mice, mean ± SEM, two-way ANOVA with Holm–Šidák post hoc test. (C) Quantification of the relative *Il1b* transcripts by RT-qPCR in female mice epidermal samples. $n = 5$–8 mice, mean ± SEM, two-way ANOVA with Holm–Šidák post hoc test. (D) Quantification of the relative *E2f2* and *Cdc25c* transcripts by RT-qPCR in female mice epidermal samples. $n = 5$–8 mice, mean ± SEM, two-way ANOVA with Holm–Šidák post hoc test.

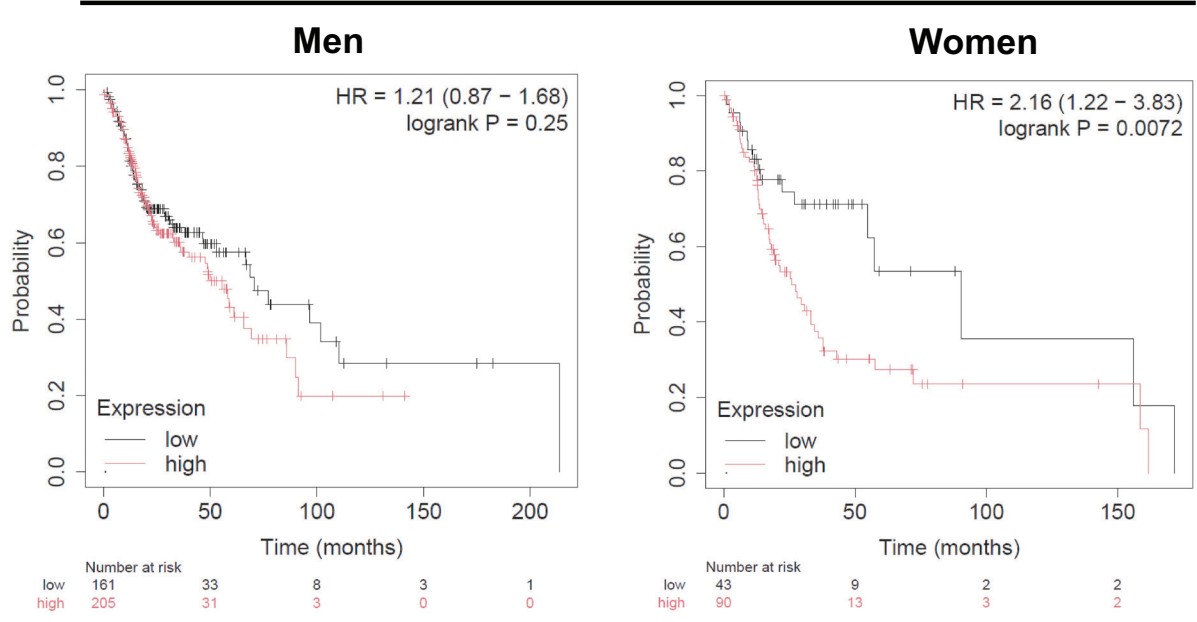

**Figure EV5. Downregulation of CDC25C expression in the skin of healthy women and in squamous cell carcinoma.**

(A, B) Relative *IL1b* (A), *E2F2* and *CDC25C* (B) transcript levels in ex vivo skin explant cultures from women subjects collected 24 h after exposure to a single dose of UV with increasing intensities (240, 480 and 720 mJ/cm$^2$) compared to non-UV-exposed control explants, quantify by RT-qPCR. $n = 4$ subjects. Mean ± SEM, two-way ANOVA with Tukey's post hoc test. (C) *CDC25C* transcript level in normal human epithelial keratinocytes (NHEK) and human Squamous Cell Carcinoma (SCC). Data from available public datasets (GSE66412). NHEK: $n = 4$, SCC: $n = 8$. Mean ± SEM, unpaired *t* test. (D) *CDC25C* transcript level in men and women human Normal Skin (NS), Actinic Keratosis (AK) and Squamous Cell Carcinoma (SCC) lesions. Data from available public datasets (GSE84293). NS: $n = 7$ (NS; 6 men and 1 woman), AK: $n = 10$ (AK; 7 men and 3 women), SCC: $n = 9$ (SCC; 5 men and 3 women). The right graph represents gene expression with sex separation for AK and SCC lesions. Mean ± SEM, one-way ANOVA with Tukey's post hoc test for non-sex separated graph, two-way ANOVA with Holm–Šidák post hoc test for sex-separated graphs. (E) Kaplan–Meier survival curves for men (left) or women (right) patients with Head and Neck Squamous Cell Carcinoma (HNSC) based on CDC25C gene expression by Kaplan–Meier plotter.

