## [Peer Review File · EMBO Reports]

An Estrogen Receptor/E2F1/CDKN3 Axis Protects from UV-induced Skin Cancers in Females

Celine Lukowicz, Carine Winkler, Catherine Roger, Joanna Fowler, Yi-Chien Tsai, Joachim Meuli, Stéphanie Claudinot, Yun-Tsan Chang, Christoph Iselin, Philip Jones, Emmanuella Guenova, Paris Jafari, and Liliane Michalik

Corresponding author(s): Celine Lukowicz (celine.lukowicz@unil.ch) , Liliane Michalik (liliane.michalik@unil.ch)

Review Timeline:

Submission Date:	27th Jan 25
Editorial Decision:	31st Jan 25
Appeal Received:	2nd Sep 25
Editorial Decision:	2nd Oct 25
Revision Received:	5th Dec 25
Editorial Decision:	9th Jan 26
Revision Received:	6th Feb 26
Accepted:	27th Feb 26

Editor: Esther Schnapp

Transaction Report:

30th Jan 2025

Dear Dr. Lukowicz,

Thank you for the submission of your manuscript to EMBO reports. I have now read and discussed it with my colleagues here, and I am sorry to say that we all agree that it is not well suited for us.

We note that your study reports that the transcription factor E2F1 and its target gene CDKN3 exhibit a differential response to UV exposure in the skin of male and female mice. You show that males develop more severe skin lesions than females in response to UV exposure, that UV exposure induces a similar DNA damage and mutation pattern in male and female epidermis, that it affects epidermal proliferation in males while it affects differentiation in females, that 12% of the genes deregulated in response to UV are sex-specific including E2F target genes being downregulated in female epidermis specifically with E2f1 and Cdkn3 downregulation depending on ER, that ex vivo female human skin explants also reduce E2F1 expression after UV exposure, that increased expression of CDKN3 and CDC25C is more prominent in men SCC lesions compared with women, and that the expression of CDKN3 and CDC25C genes in HNSC patients negatively correlates with patient survival exclusively in women. We recognize that this will be of interest to researchers in the field.

However, we also note that the data remain correlative, and that the correlation between CDKN3 and CDC25C expression and patient survival only applies to females. While we acknowledge that this is an intriguing observation, it remains to be demonstrated whether the sex-specific gene expressions you identify contribute to SCC susceptibility. We think that the manuscript does not provide the kind of advance we are looking for in our EMBO reports papers, and we have therefore decided not to proceed with in-depth review.

That said, we think that your work is an excellent candidate for our partner journal Life Science Alliance (<http://www.life-science-alliance.org/>; our broad scope Open Access journal published in partnership between the EMBO-, Rockefeller University-, and Cold Spring Harbor Laboratory Presses). Eric Sawey, Executive Editor of Life Science Alliance (e.sawey@life-science-alliance.org) would be pleased to send your manuscript for in-depth peer review; no reformatting is required. We very much hope that you will be interested in this option: please follow the link below for transfer.

For EMBO reports, I am sorry that I cannot be more positive this time, and I thank you once more for your interest in our journal.

Yours sincerely,

** As a service to authors, EMBO Press provides authors with the ability to transfer a manuscript that one journal cannot offer to publish to another journal, without the author having to upload the manuscript data again. To transfer your manuscript to another EMBO Press journal using this service, please click on Link Not Available

UNIL | Université de Lausanne

Pr. Liliane Michalik
Center for Integrative Genomics (CIG)
Genopode Building CH-1015
Switzerland

Lausanne, 2nd September 2025

Dear Dr. Esther Schnapp,

We are pleased to resubmit our original research article entitled “**Estrogen Receptors/E2F1/CDKN3 Axis Protects from UV-induced Skin Cancers in Females**” for consideration in *EMBO reports*.

Our study uncovers a **novel mechanism explaining the sexual dimorphism in susceptibility and outcome of UV-induced skin cancer**.

Sex-based differences in cancer incidence and prognosis are of major interest, both from a mechanistic point of view and in translational aspects such as personalized medicine. However, the molecular mechanisms underlying such dimorphism remain poorly understood.

Using UV-induced skin cancer as benchmark, we show for the first time, that E2F1 and its target gene, CDKN3 respond differently to UV in male and female skin. In females, estrogen receptors suppress E2F1 activity, leading to reduced CDKN3 expression. This **female-specific mechanism blocks the transition from cell cycle G1 to S phase and protects against the progression of cutaneous squamous cell carcinoma (SCC)**, as demonstrated *in vivo* using xenografts of two **CDKN3-depleted human SCC cells**.

These findings are particularly relevant in the context of aggressive cancer: we show that in head and neck squamous cell carcinoma, low CDKN3 expression is specific to female patients and correlates with better prognosis.

Our study identifies the **ER/E2F1/CDKN3 axis** as a key mediator of sexual dimorphism in cancer, highlights molecular mechanisms that protect women from tumor progression, and suggests new avenues for sex-specific cancer prevention and therapeutic strategies.

We strongly believe that our research will appeal to the broad readership of the *EMBO reports* journal including **researchers and clinicians in oncology, endocrinology, cancer prevention, and precision medicine**.

We thank you for considering our manuscript for publication in *EMBO reports*. We look forward to the opportunity to share our findings with your readers and are happy to provide any additional information or address questions during the review process.

Sincerely,

Prof. Liliane Michalik and Dr Céline Lukowicz, PhD

Faculty of Biology and Medicine

Center for Integrative Genomics (CIG)

Dear Dr. Lukowicz,

Thank you for the submission of your manuscript to EMBO reports. We have now received the enclosed reports from 2 referees, and given that they are in fair agreement that you should be given a chance to revise your study, I am making a decision now in the interest of time.

As you will see, both referees acknowledge that the findings are interesting. However, they also have several suggestions for how the data could be strengthened and the study improved. I think all suggestions are good and should be addressed, except for referee 1 point 8; the mechanistic insight this referee is asking for does not need to be provided. All missing statistical analyses should be added. Please let me know if you have any comments or questions regarding the revisions and we can discuss this further, also in a video chat, if you like.

I would thus like to invite you to revise your manuscript with the understanding that the referee concerns must be fully addressed and their suggestions taken on board. Please address all referee concerns in a complete point-by-point response. Acceptance of the manuscript will depend on a positive outcome of a second round of review. It is EMBO reports policy to allow a single round of major revision only and acceptance or rejection of the manuscript will therefore depend on the completeness of your responses included in the next, final version of the manuscript.

We realize that it is difficult to revise to a specific deadline. In the interest of protecting the conceptual advance provided by the work, we recommend a revision within 3 months (2nd Jan 2026). Please discuss the revision progress ahead of this time with the editor if you require more time to complete the revisions.

- 1) A data availability section providing access to data deposited in public databases is missing. If you have not deposited any data, please add a sentence to the data availability section that explains that.
- 2) Your manuscript contains statistics and error bars based on $n=2$. Please use scatter blots in these cases. No statistics should be calculated if $n=2$.

3) We replaced Supplementary Information with Expanded View (EV) Figures and Tables that are collapsible/expandable online. A maximum of 5 EV Figures can be typeset. EV Figures should be cited as 'Figure EV1, Figure EV2' etc... in the text and their respective legends should be included in the main text after the legends of regular figures.

5) a complete author checklist, which you can download from our author guidelines <https://www.embopress.org/page/journal/14693178/authorguide>. Please insert information in the checklist that is also reflected in the manuscript. The completed author checklist will also be part of the RPF.

6) Please note that all corresponding authors are required to supply an ORCID ID for their name upon submission of a revised manuscript (<https://orcid.org/>). Please find instructions on how to link your ORCID ID to your account in our manuscript tracking system in our Author guidelines

<<https://www.embopress.org/page/journal/14693178/authorguide#authorshippinguidelines>>

12) All Materials and Methods need to be described in the main text using our 'Structured Methods' format, which is required for all research articles. According to this format, the Methods section includes a separate Reagents and Tools Table file (listing key reagents, experimental models, software and relevant equipment and including their sources and relevant identifiers) and a Methods and Protocols section describing the methods using a step-by-step protocol format. The aim is to facilitate adoption of the methodologies across labs. More information on how to adhere to this format as well as a downloadable template (.docx) for the Reagents and Tools Table can be found in our author guidelines: <https://www.embopress.org/page/journal/14693178/authorguide#structuredmethods>.

An example of a Method paper with Structured Methods can be found here: <https://www.embopress.org/doi/full/10.1038/s44320-024-00037-6#sec-4>

I look forward to seeing a revised form of your manuscript when it is ready.

Referee #1:

The study by Lukowicz et al investigates the signaling pathways underlying the sexual dimorphism in cutaneous squamous cell carcinoma (SCC) development. To this end, they analyzed acute and chronic UV-induced defects in hairless mice, acute UV effects on ex vivo skin explants, the growth of human squamous carcinoma cell lines, and publicly available datasets. They report that UV-induced SCC development is enhanced in male mice compared to females, even though acute UV leads to similar DNA damage in mice of both sexes and repeated UV doses induce similar mutation patterns.

In addition, they show that acute UV increases epidermal proliferation in males but not in females, whereas differentiation is selectively enhanced in females.

Moreover, they provide evidence that E2F1/2 and their target gene CDKN3 are down regulated in acute UV-treated epidermis of female mice by estrogen receptors, and that chronic UV treatments enhances the number of CDKN3 expressing cells in SCC of males.

To extend their findings to humans, they analyzed human skin explants and publicly available data sets of normal skin, actinic keratosis (AK) and SCC which revealed increased CDKN3 and CDC25C in SCC of men. Moreover, CDKN3 depletion in human squamous carcinoma cell lines impaired their progression into the S phase of the cell cycle and colony-forming potential, as well as tumor growth in xenografted mice. Finally, they report that low CDKN3 and CDC25C levels are associated with longer survival of women, but not men, with head and neck squamous carcinoma.

The authors conclude that they unraveled a new mechanism protecting women from carcinogen-induced cancers, which could lead to better sex-targeted preventive and therapeutic strategies in SCC and other malignancies.

The manuscript is of interest, but additional experiments and proper statistical analyses should be performed to strengthen the conclusions.

General comments

1. Page 5: "Macroscopic observations revealed that the dorsal skin of males was far more affected than females: on average, 65% of the dorsal skin surface of males showed UV-induced lesions, compared to only 27% of the dorsal skin surface of females (Fig. 1a)". However, in Fig.S1, the difference in the number of lesions is lower. Indeed, 6.8 lesions are reported in males (1.8 AK and 5 SCC), and 5.5 in females (>3 AK and 2.5 SCC). The authors should comment this discrepancy.

2. Page 6: "tumor depth increased with the severity of SCC grade, similarly in both sexes (Fig. S1d)". As no statistical analysis was performed for females, this conclusion cannot be drawn.

3. In figure 1d, 4-12 mice were analyzed by condition. With such a variability, the authors should specify the number of mice analyzed for each group to ensure the validity of the results.

4. "Page 6: "Epidermal thickness increased in both sexes in response to acute UV (Fig. 1e)". To make this statement, statistical tests should be performed between times after UV exposure but not between sexes.

5. Page 7: The authors conclude from the first set of data that female mice develop more severe atypia than males after a single dose of UV exposure, whereas males develop more severe skin lesions by chronic UV exposure. Further characterization of the impact of acute UV exposure on skin revealed that epidermal proliferation and differentiation are mainly affected in males and females respectively, but the relationship with atypia is not discussed.

6. Page 12: "we identified the transcription factor E2F, a major regulator of the cell cycle whose activity is repressed in response to UV specifically in female epidermis". It is not clear how the authors come to this conclusion. Indeed, they only provide a correlation between downregulated genes and binding of transcription factors (i.e. E2F4).

7. Page 13: "we found that E2f1 and E2f2 gene expression was decreased in response to UV only in female epidermis (Fig. 5a and Fig. S7a), while the expression of E2f3, E2f4, E2f5, E2f6, E2f7 genes was not affected by sex (Fig. S7a)". As no statistical analysis for E2F1 levels is provided in males (Fig 5a), as E2F7 levels are statistically increased by UV in males ($p= 0.0390$; Fig. S7a), but not in females, and as no statistical analysis was performed between males and females, this conclusion cannot be drawn.
8. Page 13: "... when estrogen receptors were degraded following fulvestrant treatment". No evidence is provided that the receptors are degraded in the experimental setting. Moreover, the authors show that inhibition of estrogen receptors by fulvestrant impairs UV-induced downregulation of E2f1 and Cdkn3. However, they do not provide any mechanistic insight into how these receptors downregulate these genes, which of the two estrogen receptors is involved, and whether another anti-estrogen, Tamoxifen, which is used to treat breast cancer has similar effects. Such results would strengthen the clinical implications of the study.
9. Page 14: The authors analyze CDKN3 levels after chronic UV exposure and conclude that they increase only in males. However, according to Fig. 5c, the number of positive cells might also be increased, even though at lower levels. Statistical analyses between UV treated and non-treated females should be performed, and not between UV-treated males and females. Proper statistical analyses should also be performed in Fig. 5d, and E2F1, E2F2 and Cdc25c expression determined.
10. Page 15: The impact of UV exposure is analyzed in skin explants from healthy women. The authors conclude from their experiments "UV exposure led to a decrease in proliferation 24 hours and 72 hours after UV exposure (Fig. 6a), in line with the reduction in epidermal proliferation we also observed in female mouse epidermis (Fig. 3a)". However, in Fig.3a, the number of positive Ki67 cells does not seem to be decreased 72 hrs, after UV exposure, whereas it is increased in males. In female mice, it is only decreased at 24 hrs (Fig. S3a). Moreover, proliferation is not determined in epidermis of UV treated skin explants of healthy men.
11. Page 15: "the increased expression of CDKN3 and CDC25C, was more prominent in men SCC lesions compared to women (Fig.6c and Fig. S9d)". As proper statistical analyses were not performed, this conclusion cannot be drawn.
12. Page 16: "the increase in CDKN3-positive cells in SCC compared to AK was more prominent in men than in women tissues sections". Same comment as above.
13. Page 16: The relevance of analyzing head and neck squamous cell carcinomas (HNSC) is not clear, as it is not induced by UV.
14. Page 16: It should be mentioned in the text that A431 and SCC13 are derived from skin carcinoma of a female and male, respectively, and that nude females mice were xenografted.
15. Page 16: "while weight of these tumors was reduced by 0.53cm³ and 0.21cm³" should be: "while size of these tumors was reduced by 0.53 cm³ and 0.21 cm³
16. Page 17: The conclusion of the result section "Collectively, these data indicate that decreased CDKN3 levels have a protective effect against SCC development and mechanistically, that CDKN3 promotes SCC progression by regulating G1/S phase transition of the cell cycle" is not very strong.

Minor points:

Page 10, line 13: "Fig. S3c" should be "compare Fig. S3c and 3b", and proper statistical analyses should be performed.

Page 20: Antibodies table : Anti-CDKN3 is indicated twice, but K10 and gH2AX antibodies are not listed. Their references are given page 29.

Chemicals table : Trizol LS (Page 20) and TRIzol (Page 21) are listed, but I did not find the use of TRIzol in the methods section.

Page 21: "B6.129-SKH1-Hrhr mouse" should be "B6.129-SKH1-Hrhr mice".

Page 22: "Mice were raised, housed, and experienced" should be "Mice were raised, housed, and subjected to experimental protocols/characterized..., or : and exposed to experimental conditions as described below...

Page 24: The sentence "DNA was extracted in the same manner from liver samples from the same animal to act as a germline control." is not clear.

"pooled together in an equimolar fashion" should be : "Pooled in equimolar ratios".

Page 28: "50ul PBS, and 5x106 SCC13 cells in 200ul..." should be : "50 µl PBS, and 5x106 SCC13 cells in 200 µl..."

"The maximum volume authorized by the authorities was fixed at 1 cm³, therefore mice were euthanized when tumors exceeded this size, and tumors were recovered for further analysis"... If this was the case, the authors did not comply with the guidelines.

Page 30: "Primer sequences are accessible in the supplemental Table 1".
This table was not present in the supplementary data file.

Figures and Figure legends

- as mentioned above, proper statistical analyses should be performed. Moreover, as the number of samples is relatively low, statistical analyses should be rounded at best to three decimal places.

Figure 2: "a. Left: CPD (red) immunofluorescence in male and female dorsal epidermis collected..." should be : "a. Left: CPD (red) immunofluorescence in male and female dorsal skin collected...". Same comment for 2b and Fig. 3; Fig. S3.

Figure 6c: "Expression of CDKN3 mRNA expression" should be "CDKN3 transcript level"...idem for CDC25C in Fig. S9c and d.

Fig 6c: "The left graph shows CDKN3 expression with sex separation" should be "The right graph shows CDKN3 expression with sex separation.."

Fig 6d: "mmunofluorescence" should be "immunofluorescence".

Fig S1: "b-c. Left: Ki67 (red) immunofluorescence in AK (panel c) or SCC (panel d) lesions" should be "b-c. Left: Ki67 (red) immunofluorescence in AK (b) or SCC (c) lesions.

"d. Tumor depth from different stages SCC" should be : "d. Tumor depth from indicated SCC stages".

Fig S2: Please clarify: in a "mice exposed to UV for 8 weeks and aged for 14 weeks".

In d: "Differential selection between male and female mice".

Supplementary Fig. S4. "Cytokine production in male and female mice following acute UV exposure" should be : "Cytokine transcript levels in male and female mice following acute UV exposure.

Fig S5: "n= 4 mice per sex" should be : " n= 4 male mice"

Fig S6: "n= 4 mice per sex" should be : " n= 4 female mice"

Fig S7 : "a. Quantification of the relative gene expression of E2f2, E2F3, E2f4, E2f5, E2f6 and E2f7 at mRNA level by RT-qPCR" should be " a. Quantification of the relative E2f2, E2F3, E2f4, E2f5, E2f6 and E2f7 transcripts by RT-qPCR".

"b. Quantification of the relative gene expression of E2f2 and Cdc25c at mRNA level by RT-qPCR..." should be : "b. Quantification of the relative Cdc25c transcript level by RT-qPCR..."

Fig S8: "a. Left: Representative image of the female uterus following treatment with either solvent or fulvestrant. Right: Relative uterus weight after treatment with fulvestrant (150 mg/kg) for 48 hours, or with solvent (control). n = 5-8 mice",
Should be : "a. Left: Representative image of the uterus following treatment with either solvent or fulvestrant. Right: Relative uterus weight after treatment with fulvestrant (150 mg/kg) for 48 hours, or with solvent (control). n = 5-8 female mice,..."

Fig. S9: "a-b Relative mRNA expression of IL1b, E2F2 and CDC25C..." should be :

"a-b Relative IL1b (a), and E2F2 and CDC25C (b) transcript levels..."

"The two right graphs represent gene expression with sex separation" should be "The right graphs represent gene expression with sex separation".

Fig. S10: "Quantification of the relative gene expression of CDKN3at mRNA level by RT-qPCR" should be : "Quantification of the relative CDKN3 transcripts by RT-qPCR".

Referee #2:

It has long been known that the incidence of cutaneous squamous cell carcinoma (SCC) is significantly higher in male compared to female individuals, but the underlying mechanisms remained speculative. Here, the authors recapitulate this finding in mice. Most importantly, they show that the effect is dependent on estrogen receptors and involve an E2F1-CDKN3 signaling axis. These data are highly relevant and important for the field. The results paper is very well written, and the data are nicely presented. Overall, this is an important study from a highly experienced team of investigators.

I have a few suggestions and comments:

- 1.) Fig. 1c shows epidermal atypia after acute UV irradiation. It would be useful to show the individual parameters (nuclear size, shape, texture, nucleolar prominence) separately. It is not fully clear how the overall score was calculated.
- 2.) The result in Fig. 1e suggests that the response of keratinocytes to UV irradiation may simply be faster in female compared male mice (rather than generally stronger). A 5-day time point would be useful to check if there are still differences between females and males at a later stage.
- 3.) Please mention the time point in the legend to Fig. S3b.
- 4.) Fig. 5a nicely shows that E2f1 and Cdkn3 expression are only significantly down-regulated by UV irradiation in female mice but not in male mice. However, the absolute expression of both genes is still higher in female than in male mice - prior to and after UVB irradiation (at least at the mRNA level). Therefore, it is unlikely that the altered (early) down-regulation of these genes by UV explains the difference in tumorigenesis. The strong reduction after chronic UV irradiation is much more important - this should be clarified.
- 5.) Page 13, second paragraph: Please mention why you expect an increase in IL-1b expression in fulvestrant-treated mice and cite a manuscript, which describes this regulation.
- 6.) Page 13, last paragraph: Please show that estrogen receptors are really degraded by fulvestrant in your experimental setting. Does estrogen treatment of cultured keratinocytes induce the expression of E2F1 and CDKN3? This would show a direct effect of estrogen on the expression of these genes.
- 7.) Page 14, second paragraph: I suggest replacing: "significant increase in CDKN3-positive cells..." by "we found significantly higher numbers of CDKN3-positive cells in..."
- 8.) The results shown with human skin explants from women is very interesting and consistent with the mouse data. However, it would be important to show that this down-regulation does not occur in skin explants from men.
- 9.) The differences between men and women in Fig. 6c and S9d are not convincing because of the high variability.

Point-by-point response to the reviewers' concerns

First, we would like to thank all the reviewers for their constructive feedback that allowed us to strengthen our manuscript.

Referee #1:

The study by Lukowicz et al investigates the signaling pathways underlying the sexual dimorphism in cutaneous squamous cell carcinoma (SCC) development. To this end, they analyzed acute and chronic UV-induced defects in hairless mice, acute UV effects on ex vivo skin explants, the growth of human squamous carcinoma cell lines, and publicly available datasets.

They report that UV-induced SCC development is enhanced in male mice compared to females, even though acute UV leads to similar DNA damage in mice of both sexes and repeated UV doses induce similar mutation patterns.

In addition, they show that acute UV increases epidermal proliferation in males but not in females, whereas differentiation is selectively enhanced in females.

Moreover, they provide evidence that E2F1/2 and their target gene CDKN3 are down regulated in acute UV-treated epidermis of female mice by estrogen receptors, and that chronic UV treatments enhances the number of CDKN3 expressing cells in SCC of males.

To extend their findings to humans, they analyzed human skin explants and publicly available data sets of normal skin, actinic keratosis (AK) and SCC which revealed increased CDKN3 and CDC25C in SCC of men. Moreover, CDKN3 depletion in human squamous carcinoma cell lines impaired their progression into the S phase of the cell cycle and colony-forming potential, as well as tumor growth in xenografted mice. Finally, they report that low CDKN3 and CDC25C levels are associated with longer survival of women, but not men, with head and neck squamous carcinoma.

The authors conclude that they unraveled a new mechanism protecting women from carcinogen-induced cancers, which could lead to better sex-targeted preventive and therapeutic strategies in SCC and other malignancies.

The manuscript is of interest, but additional experiments and proper statistical analyses should be performed to strengthen the conclusions.

We would like to thank the reviewer for her/his positive assessment of our manuscript and for all the constructive comments and questions.

General comments

1. Page 5: "Macroscopic observations revealed that the dorsal skin of males was far more affected than females: on average, 65% of the dorsal skin surface of males showed UV-induced lesions, compared to only 27% of the dorsal skin surface of females (Fig. 1a)". However, in Fig.S1, the difference in the number of lesions is lower. Indeed, 6.8 lesions are reported in males (1.8 AK and 5 SCC), and 5.5 in females (>3 AK and 2.5 SCC). The authors should comment this discrepancy.

We are glad to clarify our tumor sampling collection method.

Dorsal lesions observed in male and female mice after chronic UV exposure differ: in males, the entire dorsal skin is red (erythematous) and severely damaged, exhibiting poorly defined lesions. In contrast, females exhibit distinct lesions that are isolated from each another.

Following the advice of the dermatopathologist, who observes the same pattern in some men patients with multiple scalp lesions, a punch biopsy should be collected to represent an entire lesion when the lesion is poorly defined. Therefore, in males, we collected *i)* the best-defined lesions and *ii)* for the rest of the skin area, we collected punch biopsies representative of the damaged area. Indeed, it was not possible to collect and analyze all the dorsal skin surface in males.

In the end, we collected a total of 43 samples from males and 40 lesions from females.

A clarification regarding this lesion's sampling procedure has been added to the Methods section page 25: "Tumors were collected as follows: well-defined lesions were excised entirely, whereas for poorly defined lesions, which were observed only in males, representative punch biopsies of the affected areas were sampled."

2. Page 6: "tumor depth increased with the severity of SCC grade, similarly in both sexes (Fig. S1d)". As no statistical analysis was performed for females, this conclusion cannot be drawn.

We fully agree that, since not all statistical analyses were shown, no conclusions could be drawn. We initially thought that all figures would be less cluttered and therefore clearer by presenting only statistically significant data, but we realize that this is not the case.

All statistical analyses have now been added to Figure EV1D, with significant values in bold. The corresponding sentence in the manuscript has been revised as follows page 6: "tumor depth increased with the severity of SCC grade, only significantly in males (Fig. EV1D)".

We have explained our statistical analyses in the Methods section to page 35 as follows:

"To compare two independent variables in more than two groups, we used two-way ANOVA after verifying normality and homoscedasticity of the data. Post-hoc tests were selected based on the experimental design: Tukey's or Dunnett's for balanced pairwise comparisons, Holm-Šidák for multiple related comparisons with unbalanced samples, and uncorrected Fisher's LSD for a small number of samples, interpreted cautiously due to the lack of correction for multiple comparisons."

3. In figure 1d, 4-12 mice were analyzed by condition. With such a variability, the authors should specify the number of mice analyzed for each group to ensure the validity of the results.

We have added the numbers of mice analyzed for each time point and each sex in the Figure Legend 1D page 46: No UV: males (n = 5), females (n = 9); 24 hours post-UV: males (n = 15), females (n = 18); 72 hours post-UV: males (n = 6), females (n = 4).

4. "Page 6: "Epidermal thickness increased in both sexes in response to acute UV (Fig. 1e)". To make this statement, statistical tests should be performed between times after UV exposure but not between sexes.

Like for Fig. EV1D (Reviewer comment 2), we initially chose to show only the statistically significant data that compared males and females data. All statistics have now been added in figure 1E, which we believe now allows us to make this statement "Epidermal thickness increased in both sexes in response to acute UV (Fig. 1E)"

5. Page 7: The authors conclude from the first set of data that female mice develop more severe atypia than males after a single dose of UV exposure, whereas males develop more severe skin

lesions by chronic UV exposure.

Further characterization of the impact of acute UV exposure on skin revealed that epidermal proliferation and differentiation are mainly affected in males and females respectively, but the relationship with atypia is not discussed.

We thank the reviewer for raising this important aspect. We have added the following paragraph to the discussion section page 18-19:

“We observed more severe UV-induced epidermal atypia in female mice compared to males. This atypia was associated with altered epidermal differentiation, consistent with previous findings showing that defects in the differentiation process leads to epidermal disorganization and atypical architecture (Y. Li et al., 2020). In males, we observed that this effect was transient, as their differentiation capacity was restored 72 hours after UV exposure. In contrast, the prolonged disruption of the differentiation process in females likely impairs epidermal stratification, thereby contributing to the more pronounced epidermal atypia observed in female epidermis. Moreover, the higher proliferation rate observed in male epidermis following acute UV exposure likely supports effective epidermal stratification and, consequently, reduced atypia. Indeed, a previous study demonstrated that during epidermal development, higher proliferation rates, particularly in basal and early suprabasal layers are associated with the formation of a properly stratified epidermis (Damen et al., 2021). The combination of impaired epidermal differentiation in females and increased epidermal proliferation in males could therefore explain the sex-dependent differences in epidermal atypia observed in our study.”

6. Page 12: "we identified the transcription factor E2F, a major regulator of the cell cycle whose activity is repressed in response to UV specifically in female epidermis". It is not clear how the authors come to this conclusion. Indeed, they only provide a correlation between downregulated genes and binding of transcription factors (i.e. E2F4).

To address this question, we have added supplementary tables (Appendix Tables S1, S2, S3, S4) presenting, for each transcription factor identified, the list of its target genes detected in our dataset. Appendix Table S2 shows that for the E2F4 transcription factor, 60 target genes were identified, which is much more and with higher significance (adjusted p-value= 1.82×10^{-13}) than any of the other transcription factors. It is important to note that many of these genes are also shared with other E2F family members. Indeed, among these 60 E2F4 target genes, 43 are common with E2F7 and E2F8 (71.7%), 39 are common with E2F1 (65%), 38 are common with E2F2 (63.3%) and 22 are common with E2F3 (36.6%).

We have added the following paragraph to the results section page 13: “The list of target genes identified for each transcription factor — downregulated (Appendix Tables S1 and S2) and upregulated (Appendix Tables S3 and S4) in response to acute UV exposure — is presented separately for each sex. As shown in Appendix Table S2, the transcription factor E2F4 exhibits the highest number of identified deregulated target genes (n = 60), with the strongest statistical significance (adjusted p-value = 1.82×10^{-13}). Notably, many of these target genes overlap with those of other E2F family members: among the 60 E2F4 targets, 43 are shared with E2F7 and E2F8 (71.7%), 39 with E2F1 (65%), 38 with E2F2 (63.3%), and 22 with E2F3 (36.6%) (Keenan et al., 2019).”

7. Page 13: "we found that E2f1 and E2f2 gene expression was decreased in response to UV only in female epidermis (Fig. 5a and Fig. S7a), while the expression of E2f3, E2f4, E2f5, E2f6, E2f7 genes was not affected by sex (Fig. S7a)".

As no statistical analysis for E2F1 levels is provided in males (Fig 5a), as E2F7 levels are statistically increased by UV in males ($p= 0.0390$; Fig. S7a), but not in females, and as not statistical analysis was performed between males and females, this conclusion cannot be drawn.

Like for Fig. EV1D (Reviewer comment 2), we initially chose to show only the statistically significant data that compared male and female data. All statistics have now been added in figures Fig. 5A and Fig. EV3A as well.

8. Page 13: "... when estrogen receptors were degraded following fulvestrant treatment". No evidence is provided that the receptors are degraded in the experimental setting.

To answer this comment, we have added our analyses of the expression of Estrogen Receptor α by immunofluorescence in three types of skin samples: solvent-treated female mice, fulvestrant-treated female mice, and male mice that we had collected while optimizing our experiment. This staining, now included in Figure EV4A, shows that Estrogen Receptor α was indeed detected in female skin, but not detected in male skin, nor in female skin after treatment with fulvestrant at 150 mg/kg for 48 hours, demonstrating that Estrogen Receptor α was indeed degraded in female skin treated with fulvestrant.

We changed the text describing Fig. EV4 page 14 accordingly: "The effectiveness of Estrogen Receptors antagonism was demonstrated by the expected absence of expression of Estrogen Receptor α in the epidermis of fulvestrant-treated female mice on the one hand (Fig. EV4A), and by a reduction in uterine weight in these mice on the other hand (Fig. EV4B). The effectiveness of UV exposure was confirmed by an increase in epidermal Il1b gene expression (Ravindran et al., 2014) (Fig. EV4C)."

Moreover, the authors show that inhibition of estrogen receptors by fulvestrant impairs UV-induced downregulation of E2f1 and Cdkn3. However, they do not provide any mechanistic insight into how these receptors downregulate these genes, which of the two estrogen receptors is involved, and whether another anti-estrogen, Tamoxifen, which is used to treat breast cancer has similar effects. Such results would strengthen the clinical implications of the study.

We thank the reviewer for suggesting this interesting *in vivo* experiment, which could indeed have direct clinical implications. However, within the time frame allowed for revising our manuscript, and given the very strict veterinary constraints in Switzerland, we were unable to increase the number of animals in order to carry out these experiments. We had to prioritize with the publisher's agreement: we decided to perform another *in vivo* experiment, which was asked by Reviewer 2 point 2) (5 days post-UV exposure time point), and which results are now shown in Appendix Figure S2 and described on page 10.

9. Page 14: The authors analyze CDKN3 levels after chronic UV exposure and conclude that they increase only in males. However, according to Fig. 5c, the number of positive cells might also be increased, even though at lower levels. Statistical analyses between UV treated and non-treated females should be performed, and not between UV-treated males and females. Proper statistical analyses should also be performed in Fig. 5d, and E2F1, E2F2 and Cdc25c expression determined.

We thank the reviewer for raising this point, proper statistics have been added on Fig 5D.

The mouse tumors included in that part of our study were very small (2–5 mm) and so were included entirely in the paraffin blocks. This material was entirely used for histological (H&E) and CDKN3 staining shown in figure 1B and figure 5D, respectively. Therefore, we are unfortunately unable to perform further analyses of gene expression. We believe that CDKN3 expression regulation upon UV exposure is supported by the data presented in figures 1A, 1B, 1C and 1D, and we hope that the reviewer will consider this sufficient, given that E2F1, E2F2 and CDC25C are not directly involved in the main conclusions of our study.

We hope the reviewer will understand these technical limitations.

10. Page 15: The impact of UV exposure is analyzed in skin explants from healthy women. The authors conclude from their experiments "UV exposure led to a decrease in proliferation 24 hours and 72 hours after UV exposure (Fig. 6a), in line with the reduction in epidermal proliferation we also observed in female mouse epidermis (Fig. 3a)". However, in Fig.3a, the number of positive Ki67 cells does not seem to be decreased 72 hrs, after UV exposure, whereas it is increased in males. In female mice, it is only decreased at 24 hrs (Fig. S3a).

We thank the reviewer for pointing to this discrepancy. We apologize for this mistake, which we have corrected as follows page 16: "UV exposure led to a decrease in proliferation 24 hours and 72 hours after UV exposure (Fig. 6A), in line with the reduction in epidermal proliferation we also observed in female mouse epidermis 24 hours after UV exposure (Fig. EV2A). Unlike what we observed in male mice, there was no increase in proliferation in women human skin 72 hours after UV exposure (Fig. 3A)."

Moreover, proliferation is not determined in epidermis of UV treated skin explants of healthy men.

We fully agree that including skin explants from healthy men would strengthen our study. However, over the past two years, we have been able to obtain only one such sample. These samples are extremely rare and therefore difficult to access, and plastic surgeons collaborators have confirmed that no male abdominoplasty procedures are scheduled at the University hospital of Lausanne within a timeframe compatible with the revision of our manuscript. A recent study revealed that this is not a specificity of the University Hospital of Lausanne but is also relevant in other hospitals and other countries, including the United States. Over a period of seven years, data were collected from 55,596 abdominoplasties: 96.9% were performed on women and 3.1% on men (Chaker et al., 2024).

(Chaker et al., 2024) Chaker, Sara C., Ya Ching Hung, Mariam Saad, Galen Perdakis, James C. Grotting, and K. Kye Higdon. 2024. 'Complications and Risks Associated With the Different Types of Abdominoplasties: An Analysis of 55,956 Patients'. *Aesthetic Surgery Journal* 44(9):965–75. doi:10.1093/ASJ/SJAE060.

Nevertheless, we have included the data we obtained with the single man sample, which we have analyzed and report here.

Figure 1: *Ex vivo* man skin explant analyses. **a.** Left: Ki67 (red) immunofluorescence in human *ex vivo* skin explant cultures from man subject, 24 hours and 72 hours after exposure to a single dose of UV (720mJ/cm²), compared to non-UV-exposed control explant. Scale bars: 20µm. Right: Quantification of the percentage of Ki67 positive epidermal cells. n(field)= 4 per subject. **b.** Quantification of *IL1b*, *E2F1*, *CDKN3*, *E2F2* and *CDC25C* transcripts levels in epidermis collected from *ex vivo* human skin explant cultures from a man subject, 24 hours after exposure to UV with increasing intensities (240, 480 and 720 mJ/cm²), compared to non-UV-exposed control explant, quantified by RT-qPCR.

These data show that UV exposure appears to gradually reduce cell proliferation by 72 hours. Gene expression analysis indicates that *E2F1*, *E2F2*, *CDKN3*, and *CDC25C* expression levels tend to decrease in a UV dose-dependent manner, similar to what was observed in female samples. However, these results should be interpreted with caution, as they are based on a single sample. This suggests that while the *ex vivo* human skin explant model effectively reproduces UV-induced effects and their temporal dynamics, consistent with those previously observed in mouse epidermis, it may not necessarily recapitulate sex-dependent differences.

11. Page 15: "the increased expression of *CDKN3* and *CDC25C*, was more prominent in men SCC lesions compared to women (Fig.6c and Fig. S9d)".

As proper statistical analyses were not performed, this conclusion cannot be drawn.

We have modified and added proper statistical analyses to the figures Fig. 6C and Fig. EV5D and edited the text as follows (page 16): "Of much interest, the increased expression of *CDKN3* and *CDC25C*, was significant in men AK and SCC lesions compared to healthy tissue (Fig. 6C and Fig. EV5D)."

12. Page 16: "the increase in *CDKN3*-positive cells in SCC compared to AK was more prominent in men than in women tissues sections". Same comment as above.

In that case, correct statistical analysis cannot be performed, because we only have n=2 samples. According to the editor request, these statistics were therefore removed from Figure 6D and only means are show on the graph.

13. Page 16: The relevance of analyzing head and neck squamous cell carcinomas (HNSC) is not clear, as it is not induced by UV.

We thank the reviewer for highlighting this point. Because the survival rate in cutaneous squamous cell carcinoma (SCC) is very high (typically exceeding 90% at five years for early or locally treated cases when detected and managed appropriately (Alam et al., 2018)), survival analysis is not a relevant indicator for this type of cancer. Therefore, we referred to what we believed is the most relevant and comparable cancer model: head and neck squamous cell carcinomas (HNSC), which is also an epithelial-derived cancer model that is well documented and referenced in multiple databases. Another important point is that among the different SCC subtypes, HNSC exhibits the closest molecular signature with cutaneous SCC. Indeed, a previous study demonstrates that the transcriptional profile of human cutaneous SCC is most similar to that of HNSC (Chitsazzadeh et al., 2016b).

We have added the following paragraph to the results section page 17: "To do so, because cutaneous SCC has a very high survival rate (Alam et al., 2018), survival analysis is not a relevant indicator for this type of cancer. We therefore turned to another form of carcinoma, highly aggressive, head and neck squamous cell carcinomas (HNSC), the SCC subtype with the closest molecular profile to cutaneous SCC (Chitsazzadeh et al., 2016b), as the most relevant comparable epithelial cancer model."

14. Page 16: It should be mentioned in the text that A431 and SCC13 are derived from skin carcinoma of a female and male, respectively, and that nude females mice were xenografted.

These precisions were added in the main text of our manuscript page 17-18:

- "We therefore evaluated the impact of CDKN3 depletion, which is recognized as proto-oncogene in some cancers (Gao et al., 2023) on the tumorigenicity of two human squamous carcinoma cell lines, A431 and SCC13, derived from female and male skin carcinoma, respectively."

- "Using an *in vivo* xenograft model of A431 and SCC13 cells in nude female mice, we showed that stable shRNA-mediated depletion of CDKN3 significantly inhibited tumor growth of both cell lines (Fig. 7E-G)."

15. Page 16: "while weight of these tumors was reduced by 0.53cm³ and 0.21cm³" should be: "while size of these tumors was reduced by 0.53 cm³ and 0.21 cm³

We thank the reviewer for pointing this mistake, which we have corrected (page 18).

16. Page 17: The conclusion of the result section "Collectively, these data indicate that decreased CDKN3 levels have a protective effect against SCC development and mechanistically, that CDKN3 promotes SCC progression by regulating G1/S phase transition of the cell cycle" is not very strong.

We thank the reviewer for her/his comment. We suggest changing the sentence (page 18) by "Collectively, these data indicate that decreased CDKN3 levels have a protective effect against SCC development, most likely through blocking G1/S phase transition and thus blocking cell cycle progression (C. Li et al., 2014; J. Wang et al., 2017)." We also added two additional references showing that inducing cell cycle arrest in the G1 phase decreases tumor development in two other cancer types: esophageal squamous cell carcinoma (J. Wang et al., 2017) and colorectal cancer (C. Li et al., 2014).

Minor points:

Page 10, line 13: "Fig. S3c" should be "compare Fig. S3c and 3b", and proper statistical analyses should be performed.

We have taken this point into account and amended the manuscript accordingly (page 11).

Page 20: Antibodies table : Anti-CDKN3 is indicated twice, but K10 and gH2AX antibodies are not listed. Their references are given page 29.

We have taken this point into account and amended the manuscript accordingly (page 22).

Chemicals table : Trizol LS (Page 20) and TRIzol (Page 21) are listed, but I did not find the use of TRIzol in the methods section.

We thank the reviewer. Upon checking, we found the TRIzol LS usage described on page 33, and we have removed the mention of TRIzol on page 23 from the chemicals table.

Page 21: "B6.129-SKH1-Hrhr mouse" should be "B6.129-SKH1-Hrhr mice".

We have taken this point into account and amended the manuscript accordingly (page 24).

Page 22: "Mice were raised, housed, and experienced" should be "Mice were raised, housed, and subjected to experimental protocols/characterized..., or : and exposed to experimental conditions as described below...

We have taken this point into account and amended the manuscript accordingly (page 24).

Page 24: The sentence "DNA was extracted in the same manner from liver samples from the same animal to act as a germline control." is not clear.

We hope we have clarified this aspect by changing this sentence (page 27) by "To analyze germline mutations, DNA was extracted from liver tissue for each animal."

"pooled together in an equimolar fashion" should be : "Pooled in equimolar ratios".

We have taken this point into account and amended the manuscript accordingly (page 27).

Page 28: "50ul PBS, and 5x10⁶ SCC13 cells in 200ul..." should be : "50 µl PBS, and 5x10⁶ SCC13 cells in 200 µl..."

We have taken this point into account and amended the manuscript accordingly (page 31).

"The maximum volume authorized by the authorities was fixed at 1 cm³, therefore mice were euthanized when tumors exceeded this size, and tumors were recovered for further analysis" ... If this was the case, the authors did not comply with the guidelines.

We are grateful to the reviewer for this comment, which helps us understand that our explanations on this important aspect were unclear.

We have added more precise information in Materials and Methods section page 31: "The maximum tumor volume authorized by the authorities was fixed at 1 cm^3 , as estimated once per week in live animals using the following formula: $(4/3) \times 3,14159 \times (\text{Length}/2) \times (\text{Width}/2)^2$. Mice were euthanized when tumors measured on the live animal reached this size, and tumors were recovered for further analyses. "

Page 30: "Primer sequences are accessible in the supplemental Table 1".
This table was not present in the supplementary data file.

We apologize for this oversight. We have added the primer sequence table (Appendix Table S5).

Figures and Figure legends

- as mentioned above, proper statistical analyses should be performed. Moreover, as the number of samples is relatively low, statistical analyses should be rounded at best to three decimal places.

We have taken this point into account and amended the figure accordingly.

Figure 2: "a. Left: CPD (red) immunofluorescence in male and female dorsal epidermis collected..." should be : "a. Left: CPD (red) immunofluorescence in male and female dorsal skin collected...". Same comment for 2b and Fig. 3; Fig. S3.

We have taken this point into account and amended legends of Fig. 2A, 2B and Fig. 3, Fig. EV2 (pages 46-52).

Figure 6c: "Expression of CDKN3 mRNA expression" should be "CDKN3 transcript level"...idem for CDC25C in Fig. S9c and d.

Fig 6c: "The left graph shows CDKN3 expression with sex separation" should be "The right graph shows CDKN3 expression with sex separation..

Fig 6d: "mmunofluorescence" should be "immunofluorescence".

We have taken this point into account and amended legends of Fig. 6C, Fig. 6D and Fig EV5 (pages 50-54).

Fig S1: "b-c. Left: Ki67 (red) immunofluorescence in AK (panel c) or SCC (panel d) lesions" should be "b-c. Left: Ki67 (red) immunofluorescence in AK (b) or SCC (c) lesions.

"d. Tumor depth from different stages SCC" should be : "d. Tumor depth from indicated SCC stages".

We have taken this point into account and amended legends of Fig. EV1 page 52.

Fig S2: Please clarify: in a "mice exposed to UV for 8 weeks and aged for 14 weeks".

In d: "Differential selection between male and female mice".

We have changed the Appendix figure legend S1A by "UV mutational signatures (SBS7a and SBS7b) on epidermis from mice exposed to UV for 2 weeks, 8 weeks or 12 weeks. n= 3 to 5 mice." and in Appendix Fig. S1D by "Differential selection of mutant genes across sex. Differential selection between male and female mice at 2, 8, and 12 weeks shows no significant sex-specific signal. Each

dot represents a gene. Dots that deviate strongly from the diagonal would indicate genes under differential selection; however, no significant sex-specific differences were observed."

Supplementary Fig. S4. "Cytokine production in male and female mice following acute UV exposure" should be : "Cytokine transcript levels in male and female mice following acute UV exposure."

We have taken this point into account and amended legends of Appendix Figure S3.

Fig S5: "n= 4 mice per sex" should be : " n= 4 male mice"

Fig S6: "n= 4 mice per sex" should be : " n= 4 female mice"

We have taken this point into account and amended legends of Appendix Fig. S4 and S5.

Fig S7 : "a. Quantification of the relative gene expression of E2f2, E2F3, E2f4, E2f5, E2f6 and E2f7 at mRNA level by RT-qPCR" should be " a. Quantification of the relative E2f2, E2F3, E2f4, E2f5, E2f6 and E2f7 transcripts by RT-qPCR".

We have taken this point into account and amended legends of Fig. EV3A (page 53).

"b. Quantification of the relative gene expression of E2f2 and Cdc25c at mRNA level by RT-qPCR..."

should be : "b. Quantification of the relative Cdc25c transcript level by RT-qPCR..."

We have taken this point into account and amended legends of Fig. EV3B (page 53).

Fig S8: "a. Left: Representative image of the female uterus following treatment with either solvent or fulvestrant. Right: Relative uterus weight after treatment with fulvestrant (150 mg/kg) for 48 hours, or with solvent (control). n = 5-8 mice",

Should be : "a. Left: Representative image of the uterus following treatment with either solvent or fulvestrant. Right: Relative uterus weight after treatment with fulvestrant (150 mg/kg) for 48 hours, or with solvent (control). n = 5-8 female mice,..."

We have taken this point into account and amended legends of Fig. EV4B (page 53).

Fig. S9: "a-b Relative mRNA expression of IL1b, E2F2 and CDC25C..." should be :

"a-b Relative IL1b (a), and E2F2 and CDC25C (b) transcript levels..."

"The two right graphs represent gene expression with sex separation" should be "The right graphs represent gene expression with sex separation".

We have taken this point into account and amended legends of Fig. EV5 (page 54).

Fig. S10: "Quantification of the relative gene expression of CDKN3at mRNA level by RT-qPCR" should be : "Quantification of the relative CDKN3 transcripts by RT-qPCR".

We have taken this point into account and amended legends of Appendix Fig. S7.

Referee #2:

It has long been known that the incidence of cutaneous squamous cell carcinoma (SCC) is significantly

higher in male compared to female individuals, but the underlying mechanisms remained speculative. Here, the authors recapitulate this finding in mice. Most importantly, they show that the effect is dependent on estrogen receptors and involve an E2F1-CDKN3 signaling axis. These data are highly relevant and important for the field. The results paper is very well written, and the data are nicely presented. Overall, this is an important study from a highly experienced team of investigators.

We would like to thank the reviewer for her/his positive assessment of our manuscript and for all her/his constructive comments and questions.

I have a few suggestions and comments:

1.) Fig. 1c shows epidermal atypia after acute UV irradiation. It would be useful to show the individual parameters (nuclear size, shape, texture, nucleolar prominence) separately. It is not fully clear how the overall score was calculated.

We thank the reviewer for this comment. As highlighted in this article (Pambuccian, 2015), cellular atypia is not readily quantifiable by evaluating individual morphological parameters such as nuclear size, shape, texture, or nucleolar prominence in isolation. Instead, the assessment of atypia relies on an integrative evaluation performed by dermatopathologists, who combine these histological features based on their expertise and experience to estimate the overall degree of cellular abnormality. Consequently, atypia represents a semi-quantitative and observer-dependent parameter rather than an objective measurement derived from discrete morphometric variables.

To address this concern and clarify the methodology, we have revised the description in the Methods section as follows page 26:

“Epidermal atypia was graded based on cytological features, including keratinocyte nuclei larger than the average size within each specific epidermal layer, abnormal nuclear shape and texture, nucleolar prominence in more than 10% of keratinocytes, as well as architectural changes such as acanthosis, hyperkeratosis, altered epidermal thickness, and disorganization of cell alignment (Katayama et al., 2022; Pambuccian, 2015; Pellacani et al., 2015; Sanfrancesco et al., 2013; Smoller, 2006; Tucker et al., 2002). The degree of atypia was based on the number of epidermal layers affected: Grade 0, no atypia; Grade 1, discrete atypia; Grade 2, intermediate atypia; and Grade 3, severe atypia. Histological evaluation was performed blindly by a dermatopathologist.”

2.) The result in Fig. 1e suggests that the response of keratinocytes to UV irradiation may simply be faster in female compared male mice (rather than generally stronger). A 5-day time point would be useful to check if there are still differences between females and males at a later stage. Ongoing

We thank the reviewer for suggesting this experiment. We have added the data to the Appendix Figure S2.

These new data show that 5 days after a single UV exposure, the epidermis returns to its original thickness (similar to the thickness of the epidermis non exposed to UV) in both sexes (Appendix Fig. S2A). Proliferation quantified using Ki67 staining, similar to epidermal thickness, 5 days post-UV proliferation returns to its basal levels (No UV) and no difference between males and females was observed (Appendix Fig. S2B). These data show that acute UV exposure induces transient differences between males and females in terms of epidermal thickness and proliferation, which is visible at day

two and three after UV exposure. At a later stage, differences are no longer observed, which suggests a difference in amplitude rather than a difference in the kinetics of the response to UV radiation in males and females, based on the time points we analyzed (please refer to graphs below).

Figure 2: Epidermal thickness measurements (a) and proliferation (b) in male and female mice following a single dose of acute UV exposure (120mJ/cm²) at 24 hours, 72 hours and 120 hours time-points, compared to control mice (No UV). n=4 to 12 mice, mean ±SEM, two-way ANOVA followed by Holm–Šidák post hoc test.

We have added the following paragraph to the results section page 10: “Five days after a single UV exposure, epidermal thickness (Appendix Fig. S2A) and epidermal proliferation (Appendix Fig. S2B) returned to baseline in both sexes, with no difference observed between sexes. These findings indicate that acute UV exposure induces transient sex-dependent differences detectable at day two and three, reflecting a difference in response amplitude rather than in response kinetics, based on the time points we analyzed.”

3.) Please mention the time point in the legend to Fig. S3b.

The time point was added in the legend of the Fig. EV2B page 52.

4.) Fig. 5a nicely shows that E2f1 and Cdkn3 expression are only significantly down-regulated by UV irradiation in female mice but not in male mice. However, the absolute expression of both genes is still higher in female than in male mice - prior to and after UVB irradiation (at least at the mRNA level). Therefore, it is unlikely that the altered (early) down-regulation of these genes by UV explains the difference in tumorigenesis. The strong reduction after chronic UV irradiation is much more important - this should be clarified.

We thank the reviewer for pointing this out. We hope to have clarified this point by modifying the Discussion section as follows (page 20):

“In response to acute UV exposure, we observed a downregulation in CDKN3 expression in the epidermis of female mice only, with CDKN3 mRNA levels remaining higher than in male epidermis. We believe that the key difference between males and females lies in the ability to dynamically regulate the expression of CDKN3, rather than in its level of expression, thereby limiting its overexpression during repeated (chronic) UV exposures. The pronounced reduction of CDKN3 expression after chronic UV exposure, observed only in female epidermis further supports the

hypothesis that the ability to downregulate Cdkn3 under repeated UV exposure provides a protective effect.”

5.) Page 13, second paragraph: Please mention why you expect an increase in IL-1b expression in fulvestrant-treated mice and cite a manuscript, which describes this regulation.

We thank the reviewer for this question, which allows us to clarify this point. It is well-documented that, independently of fulvestrant treatment, UV exposure induces the production of epidermal cytokines, such as Il1b and Tgfb1. Therefore, we routinely use the increase in *Il1b* expression as a positive control to ensure that the epidermis has responded as expected to UV exposure (Ravindran et al., 2014). We edited our manuscript to clarify this point, and we added a reference (page 14): “The effectiveness of UV exposure was confirmed by an increase in epidermal Il1b gene expression (Ravindran et al., 2014) (Fig. EV4C).”

6.) Page 13, last paragraph: Please show that estrogen receptors are really degraded by fulvestrant in your experimental setting.

To answer this important comment, we have included the analyses of the expression of Estrogen Receptor α using immunofluorescent staining in skin samples of solvent-treated females, fulvestrant-treated females, and males that we had collected while optimizing our experiment. This staining, now included in Figure EV4A, shows that Estrogen Receptor α was detected in the skin of solvent-treated females, but was not detected in male skin, nor in female skin treated with fulvestrant (150 mg/kg for 48 hours). This demonstrates that Estrogen Receptor α was indeed degraded in fulvestrant-treated female skin.

We edited the text as follows (page 14): “The effectiveness of Estrogen Receptors antagonism was demonstrated by the expected absence of expression of Estrogen Receptor α in the epidermis of fulvestrant-treated female mice on the one hand (Fig. EV4A), and by a reduction in uterine weight in these mice on the other hand (Fig. EV4B).”

Does estrogen treatment of cultured keratinocytes induce the expression of E2F1 and CDKN3? This would show a direct effect of estrogen on the expression of these genes.

We thank the reviewer for suggesting this experiment. We treated human immortalized keratinocyte cell lines (HaCaT) with increasing concentrations of β -estradiol (E2). Our results show that only the more physiological concentration of estradiol (0.1 μ M) induced the expression of a well-known Estrogen Receptors target gene (*TFF1*), as well as the transcription factor *E2F1*. This suggests that Estrogen *via* estradiol (the main biologically active estrogen) may have a direct effect on the expression of *E2F1*. Since we observed no direct effect of estrogen on the expression of CDKN3, and *CDKN3* is a known target of *E2F1*, we can hypothesize that its expression is regulated by estrogen indirectly through *E2F1* in these human keratinocyte cell lines.

Results were added to Appendix Figure S6 and a paragraph was added on page 14-15.

“In order to test whether the regulation of the expression of these genes by estrogen is direct or not, we treated human immortalized keratinocyte cell lines (HaCaT) with increasing concentrations of β -estradiol (E2). Our results show that only the more physiological concentration of estradiol (0.1 μ M) induced the expression of a well-known Estrogen Receptors target gene (*TFF1*), as well as the transcription factor *E2F1* (Appendix Fig. S6). These findings indicate that the most physiological levels

of estradiol selectively activate the *E2F1* pathway, suggesting a subsequent indirect regulation of *CDKN3* in human keratinocyte cells."

7.) Page 14, second paragraph: I suggest replacing: "significant increase in CDKN3-positive cells..." by "we found significantly higher numbers of CDKN3-positive cells in..."

We have edited the text according to the reviewer's suggestion page 15.

8.) The results shown with human skin explants from women is very interesting and consistent with the mouse data. However, it would be important to show that this down-regulation does not occur in skin explants from men.

We fully agree that including skin explants from healthy men would strengthen our study. However, over the past two years, we have been able to obtain only one such sample. These samples are extremely rare and therefore difficult to access, and plastic surgeons collaborators have confirmed that no male abdominoplasty procedures are scheduled at the University of Lausanne hospital within a timeframe compatible with the revision of our manuscript. A recent study revealed that this is not a specificity of the University Hospital of Lausanne but is also relevant in other hospitals and other countries, including the United States. Over a period of seven years, data were collected from 55,596 abdominoplasties: 96.9% were performed on women and 3.1% on men (Chaker et al., 2024).

(Chaker et al., 2024) Chaker, Sara C., Ya Ching Hung, Mariam Saad, Galen Perdakis, James C. Grotting, and K. Kye Higdon. 2024. 'Complications and Risks Associated With the Different Types of Abdominoplasties: An Analysis of 55,956 Patients'. *Aesthetic Surgery Journal* 44(9):965–75. doi:10.1093/ASJ/SJAE060.

Nevertheless, we have included the data we obtained with the single man sample, which we have analyzed and report here.

Figure 1: *Ex vivo* man skin explant analyses. **a.** Left: Ki67 (red) immunofluorescence in human *ex vivo* skin explant cultures from man subject, 24 hours and 72 hours after exposure to a single dose of UV (720mJ/cm²), compared to non-UV-exposed control explant. Scale bars: 20µm. Right: Quantification of the percentage of Ki67 positive epidermal cells. n(field)= 4 per subject. **b.** Quantification of *IL1b*, *E2F1*, *CDKN3*, *E2F2* and *CDC25C* transcripts levels in epidermis collected from *ex vivo* human skin explant cultures from a man subject, 24 hours after exposure to UV with increasing intensities (240, 480 and 720 mJ/cm²), compared to non-UV-exposed control explant, quantified by RT-qPCR.

These data show that UV exposure appears to gradually reduce cell proliferation by 72 hours. Gene expression analysis indicates that *E2F1*, *E2F2*, *CDKN3*, and *CDC25C* expression levels tend to decrease in a UV dose-dependent manner, similar to what was observed in female samples. However, these results should be interpreted with caution, as they are based on a single sample. This suggests that while the *ex vivo* human skin explant model effectively reproduces UV-induced effects and their temporal dynamics, consistent with those previously observed in mouse epidermis, it may not necessarily recapitulate sex-dependent differences.

9.) The differences between men and women in Fig. 6c abd S11d are not convincing because of the high variability.

We fully agree that there is high variability. Moreover, the number of women samples is smaller (n= 1 for NS, n=3 for AK and n=3 for SCC) than men samples (n=6 for NS, n=7 for AK and n=6 for SCC). Despite these limitations, we believe that these data are interesting to examine, because it is the only publicly available study reporting gene expression in normal skin, AK and SCC and that includes sex as a variable. The fact that we observe a significant difference in the expression of *CDKN3* only in men SCC (Fig. 6C) reinforces what we observed in our own experiments (Fig. 6D).

To increase the statistical power of our analysis and to overcome the smaller sample size for women normal skin samples, we combined normal skin samples, AK and SCC from both sexes (men and women samples remain clearly identified with a color code). This new presentation and analysis revealed a strong significant increase in *CDKN3* and *CDC25C* in men AK and SCC compared to normal skin. Such difference was not observed in women. We modified the figures 6C and EV5D and edited the text as follows (page 16): "Of much interest, the increased expression of *CDKN3* and *CDC25C*, was significant in men AK and SCC lesions compared to healthy tissue (Fig. 6C and Fig. EV5D)."

Dear Dr. Lukowicz,

Thank you for the submission of your revised manuscript. We have now received the enclosed reports from the referees and I am happy to say that both support its publication now. Referee 1 still has several suggestions though that I would like you to address and incorporate before we can proceed with the official acceptance of your manuscript. Please submit a point-by-point response with your final ms.

A few editorial requests will also need to be addressed:

- Please add up to 5 keywords to your ms file.
- Please add the corresponding authors' emails to the title page. There is one name discrepancy: Yun-Tsan Chang in the ms vs. Yun-Tsu Chang in our online submission system, please correct.
- The author credits need to be removed from the ms file. All credits need to be entered during online ms submission.
- The REFERENCE format is not correct: et al needs to be used after 10 author names; DOIs should only be used for preprints and datasets that have not been published yet. Please correct.
- A callout for Appendix Table S4 is missing, please add. Table S5 is an incorrect callout, please correct. You can convert the Appendix tables to EV tables, which integrates them directly into the ms text in the online version. The last Appendix table can be part of the Reagents & Tools table instead (which will also be easier for the reader). If you keep any tables in the Appendix file, then the figures and tables need to be provided as one single PDF Appendix file.
- The Reagents & Tools table needs to be removed from the ms file and uploaded as a separate file.
- Each Source Data folder needs to be uploaded as a separate folder, as one folder per main ms figure.
- Materials and methods should be just Methods.
- During our routine image integrity checks, we observed that the cell images within the EV files and the Appendix file appear pixelated under analysis. This is often a result of converting original 16-bit TIFF files to RGB format for publication. While this is not inherently problematic, it can give the impression of image alteration to critical readers. To address this, please upload the EV files and the Appendix file in a higher resolution. If it is not possible to reproduce the Appendix at higher resolution, we recommend instead uploading the original blot source data with your online submission or depositing the raw files on BioStudies <https://www.ebi.ac.uk/biostudies/sourcedata/studies> and including the archive accession number in your Data Availability section. This will enable us to confirm the integrity of the complete figure set and enhance transparency for readers.

* Figure Legends - Comments *

- Please note that the exact p values are not provided in the legends of figures 1D, 7A, C; EV2 D, EV3 B. Please provide exact values as reasonable.
- Please note that the error bars are not defined in the legends of figures 7A, C, E, G

I would like to suggest some minor changes to the ms title and abstract that needs to be written in present tense. Do you agree with this:

An Estrogen Receptor/E2F1/CDKN3 Axis Protects from UV-induced Skin Cancers in Females

Men have a higher risk of developing cutaneous squamous cell carcinoma (SCC) compared with women, but models and comprehensive analyses of signaling pathways highlighting this sexual dimorphism are missing. Here, we use a UV-induced SCC model in hairless mice recapitulating this sex difference, with enhanced SCC development in males. While UV-induced DNA damage is similar between sexes, we uncover sex-specific responses in epidermal proliferation and differentiation. Global transcriptional profiling identifies E2F transcription factors as key sex-specific markers of the proliferative response to UV. E2F1/2 and their target gene CDKN3 are selectively downregulated in female mouse and human epidermis following UV exposure, and this is mediated by estrogen receptors. CDKN3 depletion impairs SCC cell progression into S-phase and reduces tumor growth in xenograft models. Consistently, low CDKN3 expression in head and neck SCC occurs exclusively in female patients and correlates with better prognosis. We thus reveal a mechanism protecting women from carcinogen-induced cancer

formation, which could lead to better sex-targeted preventive and therapeutic strategies in SCC.

- The SYNOPSIS IMAGE needs to be removed (together with the legend) from the ms file and uploaded separately without a legend:

EMBO press papers are accompanied online by A) a short (1-2 sentences) summary of the findings and their significance, B) 2-3 bullet points highlighting key results and C) a synopsis image that is exactly 550 pixels wide and 200-600 pixels high (the height is variable). The synopsis image should provide a sketch of the major findings, like a graphical abstract. Please note that text needs to be readable at the final size. Please send us this information along with the final manuscript.

I think the synopsis image you sent is too busy for its final size. It would probably be better to simplify it.

I look forward to seeing a final version of your manuscript as soon as possible. Please use this link to submit your revision: <https://embor.msubmit.net/cgi-bin/main.plex>

Referee #1:

The revised version of the manuscript by Lukowicz et al. implements a number of clarifications, even though not all proposed experiments were performed, apparently in agreement with the editor. To further strengthen the manuscript, the following points should be addressed.

General comments:

- Page 6 : "Female mice presented on average, higher number of benign AK lesions, but fewer malignant SCC lesions compared to male mice (Fig. EV1A)."

According to the figure, the number of SCC is not statistically lower in females than in males as $p = 0.07$.

- Page 10 "In agreement with this observation, expression of the K10 marker was decreased after UV exposure only in females (Fig. EV2D)."

This sentence is misleading: Indeed, it is the % of K10+ and K14- cells that is decreased.

-Page 14: the sentence : "while the expression of E2f3, E2f4, E2f5, E2f6, E2f7 genes was not affected by sex (Fig. EV3A)." is wrong : $p < 0.05$ for E2f4 and E2f5 between UV-treated males and females.

Fig. EV4A: The authors should comment on the exclusively cytoplasmic staining of ER α in solvent conditions. Does this indicate that non-genomic effects are involved or that the ER activity might depend on the estrous cycle ?

Page 14: "Significantly, when Estrogen Receptors were degraded following fulvestrant Treatment..." There are 2 ER isotypes. The authors do not demonstrate that ER β is also degraded.

Page 14: "Our results show that only the more physiological concentration of estradiol (0.1 μ M) induced the expression of a well-known Estrogen Receptors target gene (TFF1), as well as the transcription factor E2F1 (Appendix Fig. S6). These findings indicate that the most

physiological levels of estradiol..."

It is not clear what "more/most physiological" means. The sentence should be rephrased.

Page 16: "Of much interest, the increased expression of CDKN3 and CDC25C, was significant in men AK and SCC lesions compared to healthy tissue (Fig. 6C and Fig. EV5D)."

The authors should recall that CDKN3 is also increased in AK and SCC male mice (see Fig. 5D), and determine CDC25C expression in such samples to further strengthen their data.

Minor comments:

- references should be properly cited in the text (e.g Page 4 : B Tan et al., 2022 should be Tan et al., 2022 ; L. Li et al., 2023 should be Li et al., 2023 etc...)

- Fig. 2C represents the number of mutations per mm² or per 2 mm² as indicated in the legend and text Page 8? Moreover, according to the text "the epidermal mutation burden, quantified as the number of synonymous substitutions per megabase, increased 3-fold between the groups exposed for 2 versus 12 weeks (Fig. 2C, middle)", but according to the figure, the increase is much higher. In addition, the authors mention: "However, we observed no difference either in the total number of mutations or the mutation burden between male and female epidermis at any time point except for the total number of mutations 12 weeks after repeated UV exposure (Fig. 2C, left and middle)." The increase at 12 weeks should be discussed.

- The indicated skin thickness is fuzzy in Appendix Fig. S2A, but the thickness of UV-treated skin looks increased. Is the magnification really similar or are the pictures not representative?

- The white square in Fig. 5C (male +UV) should be removed.

- Fig 5D is not representative for females.

- Page 16: "Unlike what we observed in male mice, there was no increase in proliferation in women human skin 72 hours after UV exposure (Fig. 3A)." Why do the authors compare to males, and not mention that it was similar that in female mice?

- Page 16: The description "The expression of E2F1, E2F2, CDKN3, CDC25C mRNA was also reduced in a UV dose-dependent manner,..." is incorrect. Indeed for CDC25C, $p > 0.05$.

- Fig. EV5A is not described.

- Fig. 6D: The CDKN3 immunofluorescent pictures in AK are not representative, according to the quantification.

Page 17: The sentence "Available public databases, show that the expression of CDKN3 and CDC25C genes in patients with HNSC, is negatively correlated with patient survival rate only in women, and not in men (Fig. 6E and Fig. EV5E) (Nagy et al., 2021)." should be: "Available public databases show that the expression of CDKN3 and CDC25C genes in patients with HNSC is negatively correlated with patient survival rate in women, and not in men (Fig. 6E and Fig. EV5E) (Nagy et al., 2021)."

Page 18-19: the new section: "We observed more severe UV-induced epidermal atypia in female explain the sex-dependent differences in epidermal atypia observed in our study." is not well placed.

Figure legends:

Figure 1 - Chronic and acute UV exposure induce distinct, sex-specific skin responses in male and female mice.

Should be: Chronic and acute UV exposure induce distinct, sex-specific skin responses in mice.

Figure 2 - UV exposure induces similar level of epidermal DNA damage and mutations in male and female mice epidermis.

If "except for the total number of mutations 12 weeks after repeated UV exposure (Fig. 2C, left and middle)" is true, the title should be modified accordingly.

Figure 6: E Kaplan-Meier survival curves for men (left) or women (right) patients with Head and Neck Squamous Cell Carcinoma (HNSC), based on CDKN3 gene expression by Kaplan Meier plotter.

Should be: E Kaplan-Meier survival curves for men (left) or women (right) patients with Head and Neck Squamous Cell Carcinoma (HNSC), based on CDKN3 gene expression.

!dem for Fig. EV5E.

Figure 7 F: Representative images of tumors collected 8 or 14 weeks after injection of human CDKN3- depleted (shCDKN3) A431 and SCC13 cells, compared with respective controls (shNC).

It should be specified to what "or" refers to.

Idem for G.

Expanded View Figure Legends

Figure EV1 - Characterization of dorsal lesions following chronic UV exposure in males and in females.

Should be: Characterization of dorsal lesions following chronic UV exposure in

male and in female mice.
Idem Fig. EV2 and EV3.

Fig. EV3. In A and B, it should be added: ...collected 24 hours after a single dose of acute UV exposure (120mJ/cm²), compared to control skin (No UV).

Referee #2:

The authors performed additional experiments and added further explanations to address the reviewer comments. All my comments have been addressed, and the manuscript is further improved.

Point-by-point response to the reviewers' concerns

Referee #1:

The revised version of the manuscript by Lukowicz et al. implements a number of clarifications, even though not all proposed experiments were performed, apparently in agreement with the editor. To further strengthen the manuscript, the following points should be addressed.

We would like to thank reviewer 1 for their constructive feedback that allowed us to strengthen our manuscript.

General comments:

- Page 6 : "Female mice presented on average, higher number of benign AK lesions, but fewer malignant SCC lesions compared to male mice (Fig. EV1A)."

According to the figure, the number of SCC is not statistically lower in females than in males as $p=0.07$.

We thank the reviewer for highlighting this point and we have changed the sentence page 5 by : "Female mice presented a statistically higher number of benign AK lesions, and tended to have fewer malignant SCC lesions compared to male mice (Fig. EV1A)."

- Page 10 "In agreement with this observation, expression of the K10 marker was decreased after UV exposure only in females (Fig. EV2D)."

This sentence is misleading: Indeed, it is the % of K10+ and K14- cells that is decreased.

We have changed the sentence page 10 by "In agreement with this observation, the percentage of K10 positive and K14 negative epidermal cells was decreased after UV exposure only in females (Fig. EV2D)."

-Page 14: the sentence : "while the expression of E2f3, E2f4, E2f5, E2f6, E2f7 genes was not affected by sex (Fig. EV3A)." is wrong : $p<0.05$ for E2f4 and E2f5 between UV-treated males and females.

We thank the reviewer for highlighting this discrepancy and we have changed the sentence page 13 for clarification "...while the expression of E2f3, E2f6 and E2f7 genes was not affected by sex (Fig. EV3A). As for E2f4 and E2f5 expression, we observed that it was significantly increased in UV-treated females compared to males, but no difference was observed in their basal levels. "

Fig. EV4A: The authors should comment on the exclusively cytoplasmic staining of ER α in solvent conditions. Does this indicate that non-genomic effects are involved or that the ER activity might depend on the estrous cycle ?

We thank the reviewer for this comment. The lack of detectable nuclear ER α under solvent conditions is likely due to the transient nature and rapid proteasomal degradation of transcriptionally active nuclear ER α , which limits its steady-state detectability (Nawaz *et al*, 1999). For example, treatment of human breast cancer cell lines with estradiol decreases nuclear ER α protein levels (Kocanova *et al*, 2010). Therefore, we cannot exclude the involvement of genomic ER α signaling.

Additionally, ER activity is dynamically regulated across the estrous cycle, with fluctuations in endogenous estradiol driving changes in ER expression, localization, and transcriptional activity (Ciana

et al, 2006). Together, the rapid nuclear turnover of ER α and the hormonal fluctuations during the estrous cycle can influence its nuclear localization and genomic activity, highlighting that ER α signaling is tightly controlled both temporally and physiologically.

Ciana P, Scarlatti F, Biserni A, Ottobrini L, Brena A, Lana A, Zagari F, Lucignani G & Maggi A (2006) The dynamics of estrogen receptor activity. *Maturitas* 54: 315–320

Kocanova S, Mazaheri M, Caze-Subra S & Bystricky K (2010) Ligands specify estrogen receptor alpha nuclear localization and degradation. *BMC Cell Biol* 11: 98-

Nawaz Z, Lonard DM, Dennis AP, Smith CL & O'Malley BW (1999) Proteasome-dependent degradation of the human estrogen receptor. *Proc Natl Acad Sci U S A* 96: 1858–1862

Page 14: "Significantly, when Estrogen Receptors were degraded following fulvestrant Treatment..." There are 2 ER isotypes. The authors do not demonstrate that ER β is also degraded.

We thank the reviewer for highlighting this point and we have changed the sentence page 14 by "Significantly, when Estrogen Receptor α was degraded following fulvestrant treatment..."

Page 14: "Our results show that only the more physiological concentration of estradiol (0.1 μ M) induced the expression of a well-known Estrogen Receptors target gene (TFF1), as well as the transcription factor E2F1 (Appendix Fig. S6). These findings indicate that the most physiological levels of estradiol..."

It is not clear what "more/most physiological" means. The sentence should be rephrased.

We have rephrased the sentence in page 14 by "Our results show that only the lowest concentration of estradiol (0.1 μ M) we used induced the expression of a well-known Estrogen Receptors target gene (TFF1), as well as the transcription factor E2F1 (Appendix Fig. S6). These findings indicate that the lowest levels of estradiol we used..."

Page 16: "Of much interest, the increased expression of CDKN3 and CDC25C, was significant in men AK and SCC lesions compared to healthy tissue (Fig. 6C and Fig. EV5D)."

The authors should recall that CDKN3 is also increased in AK and SCC male mice (see Fig. 5D), and determine CDC25C expression in such samples to further strengthen their data.

We have rephrased the sentence page 16 by "Of much interest, the increased expression of CDKN3 and CDC25C was significant in men AK and SCC lesions compared to healthy tissue (Fig. 6C and Fig. EV5D), which is consistent with what we previously observed in AK and SCC male mice (Fig. 5D)".

It will unfortunately not be possible to determine CDC25C protein expression. The mouse tumors included in that part of our study were very small (2–5 mm) and so were included entirely in the paraffin blocks. This material was entirely used for histological (H&E) and CDKN3 staining shown in figure 1B and figure 5D, respectively. Therefore, we are unfortunately unable to perform further analyses of gene expression. We believe that CDKN3 expression regulation upon UV exposure is supported by the data presented in figures 1A, 1B, 1C and 1D, and we hope that the reviewer will consider this sufficient, given that E2F1, E2F2 and CDC25C are not directly involved in the main conclusions of our study.

Minor comments:

- references should be properly cited in the text (e.g Page 4 : B Tan et al., 2022 should be Tan et al., 2022 ; L. Li et al., 2023 should be Li et al., 2023 etc...)

We thank the reviewer for highlighting this point, which was also raised by the editor. We have adapted the citations format according to the editor request.

- Fig. 2C represents the number of mutations per mm² or per 2 mm² as indicated in the legend and text Page 8?

We thank the reviewer for highlighting this discrepancy and we have changed mm² by 2mm² in Figure 2C.

Moreover, according to the text "the epidermal mutation burden, quantified as the number of synonymous substitutions per megabase, increased 3-fold between the groups exposed for 2 versus 12 weeks (Fig. 2C, middle)", but according to the figure, the increase is much higher.

We thank the reviewer for highlighting this discrepancy and we have corrected by "the epidermal mutation burden, quantified as the number of synonymous substitutions per megabase, increased 30-fold between the groups exposed for 2 versus 12 weeks (Fig. 2C, middle)".

In addition, the authors mention: " However, we observed no difference either in the total number of mutations or the mutation burden between male and female epidermis at any time point except for the total number of mutations 12 weeks after repeated UV exposure (Fig. 2C, left and middle)." The increase at 12 weeks should be discussed.

We added a sentence to discuss this point page 7 : "Despite a higher number of mutations observed in the female epidermis compared with males 12 weeks after repeated UV exposure, neither the mutational burden nor the proportion of CC>TT UV-signature mutations was increased, suggesting that the additional mutations are attributable to the accumulation of low-level background mutations rather than to enhanced UV-induced mutagenesis."

- The indicated skin thickness is fuzzy in Appendix Fig. S2A, but the thickness of UV-treated skin looks increased. Is the magnification really similar or are the pictures not representative?

We have increased the resolution of the H&E-stained skin image presented in Appendix Fig. S2A, which now allows the thickness measurement to be clearly read, and we have checked that the magnification is indeed similar.

- The white square in Fig. 5C (male +UV) should be removed.

We would like to clarify that this is not a white square but an axis break, which was included to allow visualization of lower values.

- Fig 5D is not representative for females.

We reanalyzed the samples and selected more representative images. Figure 5D has been modified.

- Page 16: "Unlike what we observed in male mice, there was no increase in proliferation in women human skin 72 hours after UV exposure (Fig. 3A)." Why do the authors compare to males, and not mention that it was similar that in female mice?

We thank the reviewer for this comment, we have now clarified this point in the manuscript page 16 by changed the sentence as following: "Unlike what we observed in male mice, but consistent with our observations in female mice, there was no increase in proliferation in women human skin 72 hours after UV exposure (Fig. 3A). "

- Page 16: The description "The expression of E2F1, E2F2, CDKN3, CDC25C mRNA was also reduced in a UV dose-dependent manner,..." is incorrect. Indeed for CDC25C, $p > 0.05$.

We thank the reviewer for this comment, we have changed the description page 15 by this sentence "The expression of E2F1, E2F2, and CDKN3 mRNAs was also significantly reduced, and CDC25C mRNA showed a tendency to decrease in a UV dose-dependent manner, ... "

- Fig. EV5A is not described.

We thank the reviewer for pointing this out, and we have now added the following sentence on pages 15-16: 'The effectiveness of UV exposure on the different ex vivo explants was confirmed by an increase in epidermal Il1b gene expression, used as a marker of UV-induced inflammation (Ravindran et al., 2014) (Fig. EV5A).

- Fig. 6D: The CDKN3 immunofluorescent pictures in AK are not representative, according to the quantification.

In Fig. 6D, we show two men AK samples with two different percentages of CDKN3-positive cells (3.99% and 13.66%). We selected a representative image corresponding to the 3.99% value in the figure and acknowledge that it does not reflect the meaning of the two samples (8.89%) presented in the quantification graph.

- Page 17: The sentence "Available public databases, show that the expression of CDKN3 and CDC25C genes in patients with HNSC, is negatively correlated with patient survival rate only in women, and not in men (Fig. 6E and Fig. EV5E) (Nagy et al., 2021)." should be : "Available public databases show that the expression of CDKN3 and CDC25C genes in patients with HNSC is negatively correlated with patient survival rate in women, and not in men (Fig. 6E and Fig. EV5E) (Nagy et al., 2021)."

We have edited the text according to the reviewer's suggestion page 17.

Page 18-19: the new section: "We observed more severe UV-induced epidermal atypia in female explain the sex-dependent differences in epidermal atypia observed in our study." is not well placed.

We thank the reviewer for pointing this out, and we have moved the new section to pages 18-19, immediately after the sentence: 'Our study highlights that, in the long term, sex-dependent proliferative response to UV is likely crucial for sex-specific SCC development.'

Figure legends:

Figure 1 - Chronic and acute UV exposure induce distinct, sex-specific skin responses in male and female mice.

Should be : Chronic and acute UV exposure induce distinct, sex-specific skin responses in mice.

We have edited the text according to the reviewer's suggestion.

Figure 2 - UV exposure induces similar level of epidermal DNA damage and mutations in male and female mice epidermis.

If "except for the total number of mutations 12 weeks after repeated UV exposure (Fig. 2C, left and middle)" is true, the title should be modified accordingly.

With reference to the above comment regarding Figure 2C, we suggest changing the title to clarify that this refers to the UV-signature mutation: "UV exposure induces similar level of epidermal DNA damage and UV-signature mutations in male and female mice epidermis. "

Figure 6: E Kaplan-Meier survival curves for men (left) or women (right) patients with Head and Neck Squamous Cell Carcinoma (HNSC), based on CDKN3 gene expression by Kaplan Meier plotter.

Should be : E Kaplan-Meier survival curves for men (left) or women (right) patients with Head and Neck Squamous Cell Carcinoma (HNSC), based on CDKN3 gene expression.

Idem for Fig. EV5E.

We have edited the text according to the reviewer's suggestion.

Figure 7 F: Representative images of tumors collected 8 or 14 weeks after injection of human CDKN3-depleted (shCDKN3) A431 and SCC13 cells, compared with respective controls (shNC).

It should be specified to what "or" refers to.

Idem for G.

We have changed the Figure 7F legend by "Representative images of tumors collected 8 weeks after injection of human CDKN3- depleted (shCDKN3) A431 cells (top panel) and 14 weeks after injection of human CDKN3- depleted (shCDKN3) SCC13 cells (bottom panel), compared with respective controls (shNC)."

We have changed the Figure 7G legend by "Tumor weights collected 8 weeks after injection of human CDKN3-depleted (shCDKN3) A431 cells (left panel) or 14 weeks after injection of human CDKN3-depleted (shCDKN3) SCC13 cells (right panel), compared with their respective control (shNC)."

Expanded View Figure Legends

Figure EV1 - Characterization of dorsal lesions following chronic UV exposure in males and in females.

Should be : Characterization of dorsal lesions following chronic UV exposure in male and in female mice.

Idem Fig. EV2 and EV3.

We have edited the text according to the reviewer's suggestion.

Fig. EV3. In A and B, it should be added: ...collected 24 hours after a single dose of acute UV exposure (120mJ/cm²), compared to control skin (No UV).

We have edited the text according to the reviewer's suggestion.

Point-by-point response to the editor

A few editorial requests will also need to be addressed:

- Please add up to 5 keywords to your ms file.

We propose these 5 keywords (that were added to the ms): UV-induced squamous cell carcinoma – Skin – Proliferation - Estrogen receptors - E2F transcription factors

- Please add the corresponding authors' emails to the title page. There is one name discrepancy: Yun-Tsan Chang in the ms vs. Yun-Tsu Chang in our online submission system, please correct.

Thank you for pointing out this discrepancy; we have now corrected it. The corresponding authors' emails were added.

- The author credits need to be removed from the ms file. All credits need to be entered during online ms submission.

We have removed author credits from the ms file.

- The REFERENCE format is not correct: et al needs to be used after 10 author names; DOIs should only be used for preprints and datasets that have not been published yet. Please correct.

We have changed the reference format from the ms file.

- A callout for Appendix Table S4 is missing, please add. Table S5 is an incorrect callout, please correct. You can convert the Appendix tables to EV tables, which integrates them directly into the ms text in the online version. The last Appendix table can be part of the Reagents & Tools table instead (which will also be easier for the reader). If you keep any tables in the Appendix file, then the figures and tables need to be provided as one single PDF Appendix file.

We have changed Appendix tables according to editor suggestion.

- The Reagents & Tools table needs to be removed from the ms file and uploaded as a separate file.

We have removed this table from the ms file and uploaded as a separate file.

- Each Source Data folder needs to be uploaded as a separate folder, as one folder per main ms figure.

We have uploaded Source data as a separate folder.

- Materials and methods should be just Methods.

We have changed the title section.

- During our routine image integrity checks, we observed that the cell images within the EV files and the Appendix file appear pixelated under analysis. This is often a result of converting original 16-bit TIFF files to RGB format for publication. While this is not inherently problematic, it can give the impression of image alteration to critical readers.

To address this, please upload the EV files and the Appendix file in a higher resolution.

We have changed the resolution of these files.

* Figure Legends - Comments *

- Please note that the exact p values are not provided in the legends of figures 1D, 7A, C; EV2 D, EV3 B. Please provide exact values as reasonable.

We have added exact values in the legends of figures 1E, 7A, C; EV2 D, EV3 B. No statistical p-value is shown in Figure 1D.

- Please note that the error bars are not defined in the legends of figures 7A, C, E, G

We have defined error bars in the legends of figures 7A, C, E, G.

I would like to suggest some minor changes to the ms title and abstract that needs to be written in present tense. Do you agree with this:

An Estrogen Receptor/E2F1/CDKN3 Axis Protects from UV-induced Skin Cancers in Females

Men have a higher risk of developing cutaneous squamous cell carcinoma (SCC) compared with women, but models and comprehensive analyses of signaling pathways highlighting this sexual dimorphism are missing. Here, we use a UV-induced SCC model in hairless mice recapitulating this sex difference, with enhanced SCC development in males. While UV-induced DNA damage is similar between sexes, we uncover sex-specific responses in epidermal proliferation and differentiation. Global transcriptional profiling identifies E2F transcription factors as key sex-specific markers of the proliferative response to UV. E2F1/2 and their target gene CDKN3 are selectively downregulated in female mouse and human epidermis following UV exposure, and this is mediated by estrogen receptors. CDKN3 depletion impairs SCC cell progression into S-phase and reduces tumor growth in xenograft models. Consistently, low CDKN3 expression in head and neck SCC occurs exclusively in female patients and correlates with better prognosis. We thus reveal a mechanism protecting women from carcinogen-induced cancer formation, which could lead to better sex-targeted preventive and therapeutic strategies in SCC.

Thank you for these corrections; we fully agree with them. We have edited the text according to the editor's suggestion.

- The SYNOPSIS IMAGE needs to be removed (together with the legend) from the ms file and uploaded separately without a legend:

EMBO press papers are accompanied online by A) a short (1-2 sentences) summary of the findings and their significance, B) 2-3 bullet points highlighting key results and C) a synopsis image that is exactly 550 pixels wide and 200-600 pixels high (the height is variable). The synopsis image should provide a sketch of the major findings, like a graphical abstract. Please note that text needs to be readable at the final size. Please send us this information along with the final manuscript.

I think the synopsis image you sent is too busy for its final size. It would probably be better to simplify it.

Thank you for this suggestion, we have prepared a new simplify synopsis image and a short summary and key results were added to the ms.

A) Short summary:

Females are protected from UV-induced SCC through estrogen receptor-mediated downregulation of E2F1/2 and CDKN3, limiting epidermal proliferation and tumor growth.

B) Key results:

- Males exhibit higher incidence of UV-induced SCC
- UV exposure reduces epidermal proliferation in females, with downregulation of the E2F/CDKN3 axis *via* estrogen receptors.
- CDKN3 depletion protects from SCC progression

Dr. Celine Lukowicz
University of Lausanne
Center for Integrative Genomics
Switzerland

Dear Dr. Lukowicz,

I am very pleased to accept your manuscript for publication in the next available issue of EMBO reports. Thank you for your contribution to our journal.

You may qualify for financial assistance for your publication charges - either via a Springer Nature fully open access agreement or an EMBO initiative. Check your eligibility: <https://link.springer.com/journal/44319/how-to-publish-with-us>

Yours sincerely,

>>> Please note that it is EMBO Reports policy for the transcript of the editorial process (containing referee reports and your response letter) to be published as an online supplement to each paper. If you do NOT want this, you will need to inform the Editorial Office via email immediately. More information is available here: <https://link.springer.com/partners/embo-press/editorial-policies#Peer%20review>